# The astrocytic ensemble acts as a multiday trace to stabilize memory

Ken-ichi Dewa[1,22], Kodai Kaseda[1,2,22], Aoi Kuwahara[1,2], Hideaki Kubotera[1], Ayato Yamasaki[3], Natsumi Awata[4], Atsuko Komori[1], Mika A. Holtz[1], Atsushi Kasai[5,6,7], Henrik Skibbe[8,9], Norio Takata[10], Tatsushi Yokoyama[11,12,13,14], Makoto Tsuda[15,16], Genri Numata[17,18], Shun Nakamura[19], Eiki Takimoto[18,20], Masayuki Sakamoto[11,12,13,21], Minako Ito[4,7], Takahiro Masuda[3] & Jun Nagai[1,2,7 ✉]

Recalled memories become transiently labile and require stabilization[1–3]. The mechanism for stabilizing memories of survival-critical experiences, which are often emotionally salient and repeated, remains unclear[4]. Here we identify an astrocytic ensemble that is transcriptionally primed by emotional experience and functionally triggered by repeated experience to stabilize labile memory. Using a novel brain-wide *Fos* tagging and imaging method, we found that astrocytic *Fos* ensembles were preferentially recruited in regions with neuronal engrams[5] and were more widespread during fear recall than during conditioning. We established the induction mechanism of the astrocytic ensemble, which involves two steps: (1) an initial fear experience that induces day-long, slow astrocytic state changes with noradrenaline receptor upregulation; and (2) enhanced noradrenaline responses during recall, a repeated experience, enabling astrocytes to integrate coincident signals from local engrams and long-range noradrenergic projections, which induce secondary astrocytic state changes, including the upregulation of *Fos* and the neuromodulatory molecule IGFBP2. Pharmacological and genetic perturbation of the astrocytic ensemble signalling modulate engrams, and memory stability and precision. The astrocytic ensemble thus acts as a multiday trace in a subset of astrocytes after experience-dependent neural activity, which are eligible to capture future repeated experiences for stabilizing memories.

Recall transiently destabilizes memories, which require re-stabilization to become long-lasting[1–3]. Despite its importance in human cognition and neuropsychiatric disorders[6,7], the mechanisms that specifically stabilize memories of critical experiences—those that are essential for survival and frequently marked by emotional salience and repetition—remain incompletely understood. Memory traces are linked to specialized neuronal ensembles (neuronal engrams), which include neuronal populations that become *Fos*[+] during both initial and repeated experiences[8]. However, neuronal *Fos* alone, a plasticity marker that indicates circuit remodelling[9], is insufficient for memory stabilization, implying the need for additional cellular substrates.

By tiling the entire mammalian central nervous system (CNS), astrocytes structurally and functionally interact with neurons and other glia. Their functions contribute to information processing and animal behaviour, and their dysfunctions are widely implicated in CNS disorders[10,11]. Astrocytes are known to be developmentally diverse[12–14] and can adaptively change their molecular programmes in response to physiology[11,15–18] and pathology[19–21]—hereafter referred to as state—in specific ways that suggest flexibly altered functions in a context-dependent manner. This adaptability is akin to experience-dependent neuronal responses (such as *Fos* upregulation)[22] and the subsequent formation of inter-regional neuronal engram network[8]. However, the manifold nature of astrocyte responses through an extensive array of cell-surface

[1]Laboratory for Glia–Neuron Circuit Dynamics, RIKEN Center for Brain Science, Wako, Japan. [2]Department of Life Science and Medical Bioscience, Graduate School of Advanced Science and Engineering, TWIns, Waseda University, Tokyo, Japan. [3]Division of Molecular Neuroimmunology, Medical Institute of Bioregulation, Kyushu University, Fukuoka, Japan. [4]Division of Allergy and Immunology, Medical Institute of Bioregulation, Kyushu University, Fukuoka, Japan. [5]Systems Neuropharmacology, Research Institute of Environmental Medicine, Nagoya University, Nagoya, Japan. [6]Laboratory of Molecular Neuropharmacology, Graduate School of Pharamceutical Sciences, Osaka University, Suita, Japan. [7]Japan Science and Technology Agency (JST), Fusion Oriented REsearch for disruptive Science and Technology (FOREST), Kawaguchi, Japan. [8]Brain Image Analysis Unit, RIKEN Center for Brain Science, Wako, Japan. [9]Department of Informatics, Faculty of Informatics, Matsuyama University, Ehime, Japan. [10]Division of Brain Sciences, Institute for Advanced Medical Research, Keio University School of Medicine, Tokyo, Japan. [11]Department of Optical Neural and Molecular Physiology, Graduate School of Biostudies, Kyoto University, Kyoto, Japan. [12]Center for Living Systems Information Science, Graduate School of Biostudies, Kyoto University, Kyoto, Japan. [13]Department of Brain Development and Regeneration, Graduate School of Biostudies, Kyoto University, Kyoto, Japan. [14]Laboratory of Deconstruction of Stem Cells, Institute for Frontier Life and Medical Sciences, Kyoto University, Kyoto, Japan. [15]Department of Molecular and System Pharmacology, Graduate School of Pharmaceutical Sciences, Kyushu University, Fukuoka, Japan. [16]Kyushu University Institute for Advanced Study, Fukuoka, Japan. [17]Isotope Science Center, The University of Tokyo, Tokyo, Japan. [18]Department of Cardiovascular Medicine, The University of Tokyo Hospital, Tokyo, Japan. [19]Department of Cardiovascular Medicine, Institute of Science Tokyo, Tokyo, Japan. [20]Division of Cardiology, Department of Medicine, The Johns Hopkins Medical Institutions, Baltimore, MD, USA. [21]Precursory Research for Embryonic Science and Technology (PRESTO), Japan Science and Technology Agency, Kawaguchi, Japan. [22]These authors contributed equally: Ken-ichi Dewa, Kodai Kaseda. ✉e-mail: jun.nagai@riken.jp

receptors for neuroactive biomolecules[19,23,24] have made it challenging to determine whether, when and how multiple inputs are integrated by a subset of astrocytes to generate specific states that form astrocyte ensembles that are responsible for particular cellular outputs[15,18,24,25]. This is because the field has lacked adequate tools to explore behaviourally relevant astrocyte ensembles (BAEs) in an unbiased manner at the single-cell level as well as in a brain-wide manner, which would enable ensemble-specific molecular dissection and functional perturbation.

## Brain-wide BAE tagging

To agnostically and holistically identify BAEs, we developed a brain-wide astrocyte *Fos* tagging tool that enables permanent genetic access to BAEs that express *Fos* due to a specific experience. Although behaviourally relevant stimuli-induced astrocyte *Fos* mRNA[22] and c-Fos protein[18,26] have been observed in selected brain regions, its brain-wide profiling remains largely unexplored. We systematically validated a novel genetic and imaging strategy for whole-brain BAE tagging, demonstrating astrocyte selectivity, *Fos* sensitivity and temporal regulation (Extended Data Figs. 1–3). First, using a CNS-wide-transducing adenovirus (AAV) vector (AAV-PHP.eB) with an astrocyte-specific GfaABC1D promoter[27], we achieved widespread, astrocyte-selective (97.3–99.5%) expression (Extended Data Fig. 1). Second, we tested whether using AAV-PHP.eB to express Cre-dependent mNeonGreen (mNG) can serve as a marker of transient astrocytic *Fos* induction in Fos-iCre[ERT2] (TRAP2) mice[28]. This intersectional approach aimed to drive astrocytic mNG expression only when the *Fos* promoter was active during fast-acting 4-hydroxytamoxifen (4-OHT) administration, enabling behaviour epoch-specific astrocyte *Fos* tagging across the CNS. As a positive control, we virally expressed the astrocytic $G\alpha_q$-coupled designer receptor exclusively activated by designer drugs (DREADD) hM3Dq, a strong inducer of astrocyte c-Fos[26,29,30], combined with 4-OHT and the hM3Dq agonist clozapine *N*-oxide (CNO) intraperitoneal injection. hM3Dq stimulation yielded approximately 282 mNG+ astrocytes per mm², and significantly lower tagging in controls (Extended Data Fig. 2). Third, to probe the temporal dynamics of *Fos*-mNG tagging, dark-adapted mice were exposed to light at various intervals relative to 4-OHT administration[28]. Tagging peaked within a 6-h window around 4-OHT administration, with an approximately 22-fold increase in astrocytes in visual cortex compared with those in somatosensory cortex, indicating regional specificity and dependence on local neuronal activity (Extended Data Fig. 3). Collectively, these experiments validated our behaviour epoch-specific, brain-wide BAE tagging strategy.

## Recall induces astrocyte ensembles

To broadly explore BAEs in a learning paradigm, we used a contextual fear conditioning and recall task[5,28] (Fig. 1a). TRAP2::AAV-PHP. eB-GfaABC1D-DIO-mNG mice underwent three days of habituation to minimize contextual novelty and isolate associative components of fear learning and recall[28,31]. On day 4, one group received 3 foot shocks with immediate 4-OHT (FC), and another received 4-OHT immediately after recall 24 h post-FC (FR). Controls were non-shocked mice (NoFC) or fear-conditioned mice kept in the home cage for 24 h and given 4-OHT without recall (NoFR). Shocked groups exhibited robust freezing in the conditioned chamber, indicating learning and recall (Fig. 1b). One week later, whole-brain serial two-photon tomography (STPT)[32] enabled the detection of individual mNG+ astrocytes (BAEs) (Supplementary Videos 1 and 2). After excluding ventricular niches for bona fide astrocyte counts (Methods), BAE counts across 677 regions covered approximately 80.7% of the brain (Supplementary Table 2).

Whereas the number of *Fos*+ neurons increased during both FC and FR, forming networks causally linked to fear memory traces—commonly referred to as engram neurons[5,8,28], the density of *Fos*+ astrocytes during FC (FC-BAE; Fig. 1c) was seldom distinguishable from

the NoFC control (NoFC-BAE). However, we found that the number of BAEs during recall (FR-BAE) was augmented across multiple regions (Fig. 1c). This was not simply reflecting delayed *Fos* upregulation after FC, because *Fos* induction was significantly higher in FR than in NoFR. Normalized comparisons across engram regions[5] (Fig. 1d) showed a significantly greater number of FR-BAEs compared with FC-BAEs, particularly in the amygdala (Fig. 1e), the region long implicated in fear memory across species[3,4,33–35]. Consistently, in the lateral and basolateral amygdala (LA/B), astrocytic c-Fos protein was unchanged after FC or in NoFR versus NoFC, but FR induced a marked increase at 1.5 h with partial decay by 6 h (Extended Data Fig. 4). Consistently, the number of of FR-BAEs in the LA/B (FR-BAE[LA/B]) was robust whether 4-OHT was given immediately or 3 h post-FR (Extended Data Fig. 5). These data indicate that astrocyte *Fos* promoter activity, and the associated surge in *Fos* mRNA and c-Fos protein expression, peaks within a biologically plausible window of 0–3 h following FR, but not following FC.

A recent study[18] reported an increase in hippocampal astrocytic *Fos* after FC without habituation, contrasting with our findings (Fig. 1c–f). We hypothesized that this reflects contextual novelty: without prior habituation, novelty strongly drives astrocytic *Fos*. Supporting this, our 'FC + novelty' group (FC in novel context with no habituation) showed widespread astrocytic *Fos*, including in CA1 (Extended Data Fig. 6). Thus, the CA1 astrocytic *Fos* is likely to signal novelty, whereas our habituated FC protocol reveals astrocytic ensembles that are more selectively engaged in learning and recall.

To holistically compare the anatomical positioning of BAEs with previously reported brain-wide *Fos*+ neuron distribution during FC and FR[5], we analysed subregions from previously defined regions in which *Fos*+ neuron counts of FC and FR were significantly higher than in non-shocked controls, referred to as neuronal 'engram significant brain regions'[5] (Fig. 1e–g and Methods). The number of *Fos*+ neurons increased during FC versus non-shocked controls, whereas there was no apparent change in astrocytic *Fos* counts (Fig. 1f). By contrast, FR elicited a strong correlation between neuronal and astrocytic *Fos* induction (*r* = 0.56; Fig. 1g,h), suggesting that local astrocyte–neuron interactions may underlie FR-BAE formation[22]. Notably, amygdala subregions that displayed selective astrocytic *Fos* increases during FR (Fig. 1e) also showed parallel neuronal *Fos* increases (Fig. 1h, yellow dots).

To test whether astrocytic and neuronal ensembles are co-engaged within the same regions, we performed dual-colour *Fos* tagging in the amygdala of TRAP2 mice (Fig. 1i–k). Neuron-specific DIO-mCherry and astrocyte-specific DIO-mNG AAVs were co-injected, and four groups were analysed: NoFC, FC, NoFR and FR. Neuronal *Fos* increased after FC or FR, consistent with past reports[5,8,22]. By contrast, astrocytic tagging was minimal following FC and NoFR but was robustly induced after FR (Fig. 1i–k), replicating our brain-wide observations (Fig. 1c–h). These results confirm that astrocyte *Fos* induction in the amygdala is recall-specific and distinct from neuronal ensemble dynamics.

## Convergence induces astrocyte *Fos*

To identify mechanisms underlying astrocyte *Fos* induction, we combined pharmacological screening, circuit genetics, in vivo imaging and transcriptomics (Figs. 2–4). Growth factors that activate c-Fos in neurons[36] did not trigger c-Fos signals in primary astrocytes (Extended Data Fig. 7), indicating that the induction mechanisms of c-Fos in neurons and astrocytes are distinct. Given the accumulating evidence that astrocyte G-protein-coupled receptor (GPCR) activation upregulates c-Fos expression[26,29,30], we then stimulated primary astrocytes with a variety of neurotransmitters that bind to GPCRs expressed in astrocytes[11,22,23], observing that noradrenaline (NA) increased c-Fos signals (Fig. 2b,c). The astrocyte c-Fos signals induced by NA at 0.1 μM and 1 μM were both abolished by propranolol (Fig. 2d), which antagonizes $G\alpha_s$-coupled β-adrenergic receptors (β-receptors), but not by prazosin or atipamezole, which antagonize $G\alpha_q$-coupled α1-adrenoreceptors

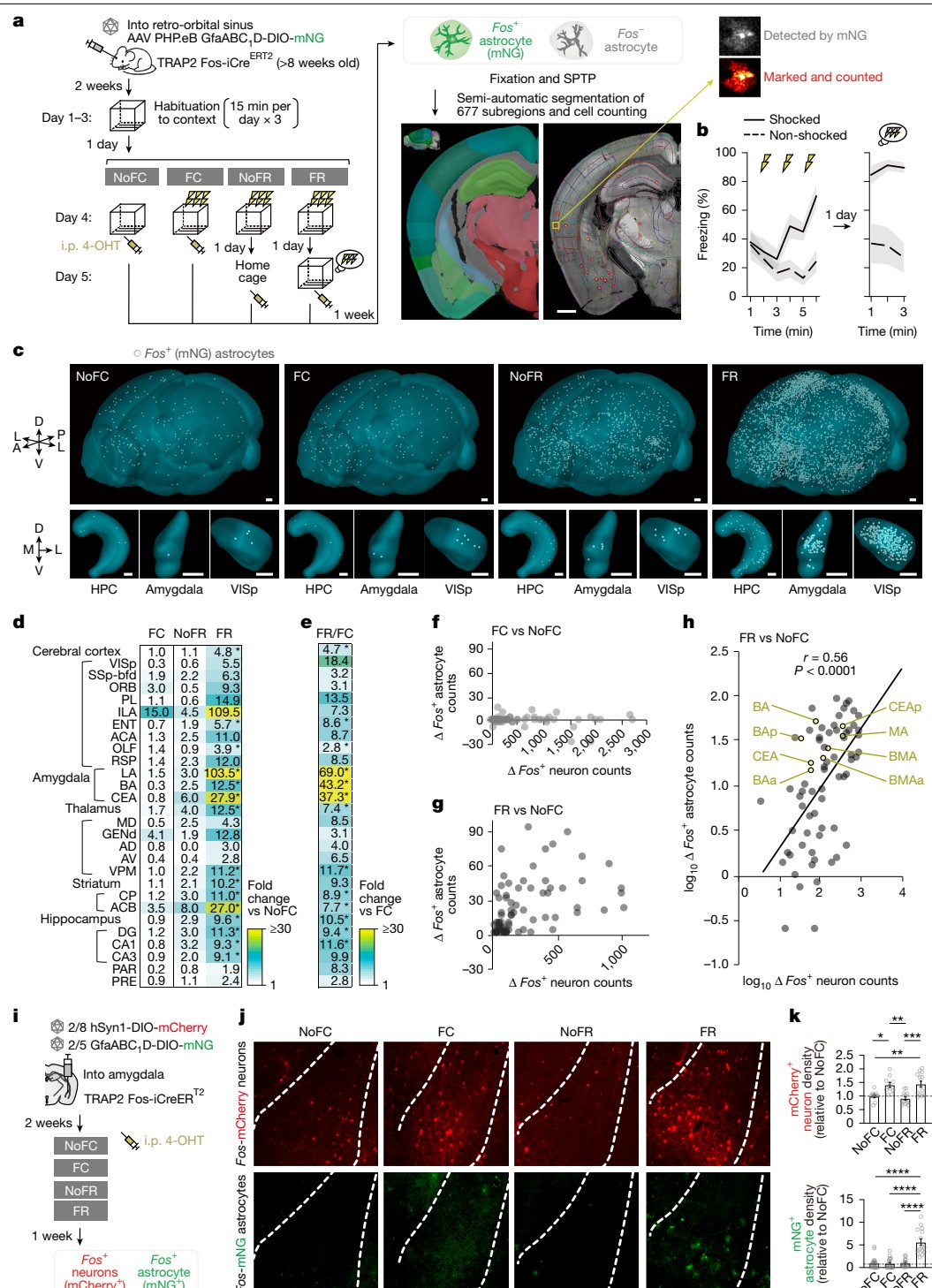

**Fig. 1 | Behaviour epoch-specific, brain-wide astrocyte *Fos* ensemble mapping. a**, *Fos*⁺ astrocytes were tagged with Cre-dependent mNG AAV in four cohorts (NoFC, FC, NoFR and FR), followed by STPT, 3D reconstruction, segmentation and single-cell counting. i.p., intraperitoneal injection. Scale bar, 1 mm. **b**, Freezing of shocked (FC, FR and NoFR: $n = 12$) and non-shocked (NoFC: $n = 6$) mice (left), and recall freezing shocked (FC and FR: $n = 8$) and non-shocked (NoFC: $n = 6$ mice) (right). **c**, 3D rendering of whole brain, the hippocampus (HPC), amygdala and primary visual cortex (VISp) with *Fos*-mNG astrocytes. A, anterior; D, dorsal; L, lateral; M medial; P, posterior; V, ventral. Scale bars, 500 μm. **d,e**, Heat map of fold changes in BAE count (versus NoFC (**d**)) and FR versus FC (**e**); NoFC: $n = 6$; FC: $n = 4$; NoFR: $n = 4$; FR: $n = 4$ mice). Abbreviations for brain regions are listed in Extended Data Fig. 6. *False discovery rate (FDR) < 0.05. Levene's test followed by one-way analysis of variance (ANOVA) or Kruskal–Wallis; post hoc Tukey or Dunn test with Bonferroni correction. FDR

controlled by Benjamini–Hochberg procedure. Full data in Supplementary Table 2. **f–h**, FC- and FR-induced *Fos* counts in neurons versus astrocytes across engram areas for FC versus NoFC (**f**), FR versus NoFC (**g**) and FR versus NoFC with log scales, showing correlation Pearson's correlation coefficient (*r*) (**h**). **h**, Amygdalar subregions are indicated with yellow dots. BMA, basomedial amygdala; MA, medial amygdala. **i**, Schematic of AAV tagging of *Fos*⁺ neurons and BAEs. **j**, Images of tagged *Fos*⁺ neurons and BAEs (representative of 10–22 sections from 3–5 mice per group). Scale bar, 200 μm. **k**, Quantification of *Fos*⁺ neurons (NoFC: $n = 11$ slices from 4 mice; FC: $n = 10$ slices from 3 mice; NoFR: $n = 12$ slices from 4 mice; FR: $n = 13$ slices from 4 mice) and BAE density (NoFC: $n = 22$ slices from 4 mice; FC: $n = 18$ slices from 3 mice; NoFR: $n = 17$ slices from 3 mice; FR: $n = 15$ slices from 5 mice). One-way ANOVA, Tukey's test. Data are mean ± s.e.m. *$P < 0.05$, **$P < 0.01$, ***$P < 0.001$, ****$P < 0.0001$; NS, not significantly different. Sample sizes and replicates in Supplementary Table 1.

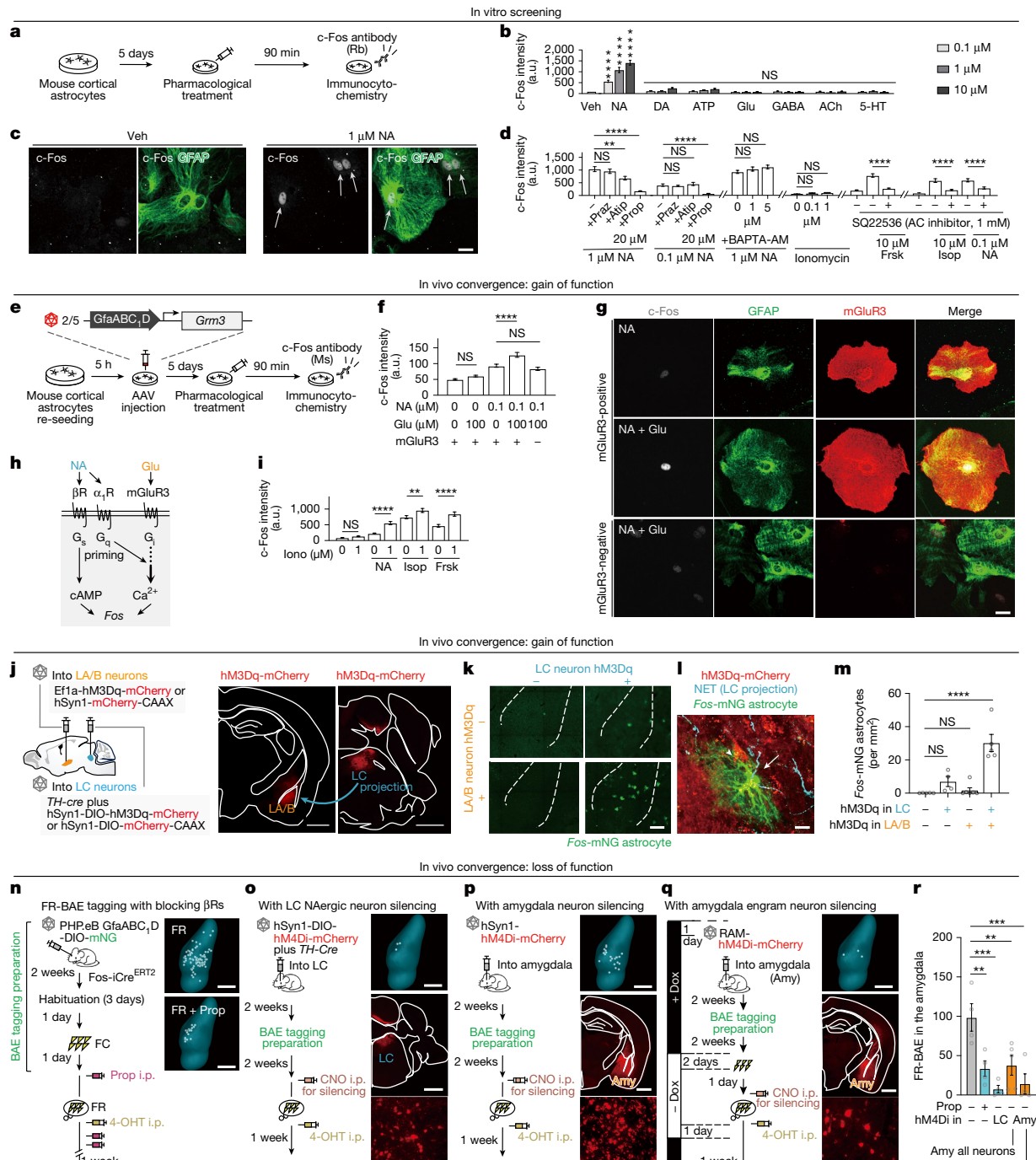

**Fig. 2 | Convergence of neural signals evokes astrocyte *Fos*. a**, Schematic of pharmacological screening. **b**, NA increased c-Fos intensity in GFAP⁺ astrocytes (*n* = 27–50 cells per condition). One-way ANOVA, Dunnett's test versus vehicle (Veh). 5-HT, serotonin; a.u., arbitrary units; ACh, acetylcholine; DA, dopamine; GABA, γ-aminobutyric acid. **c**, Image showing NA (1 μM)-induced c-Fos increase (vehicle: *n* = 40; NA 1 μM: *n* = 41 cells). Scale bar, 10 μm. **d**, NA-induced c-Fos was blocked by inhibition of β-receptors or cAMP but not by Ca²⁺ mobilization (*n* = 28–55 cells per condition). One-way ANOVA, Dunnett's or Tukey's test. AC, adenylate cyclase; Atip, atipamezole; Frsk, forskolin; Isop, isoproterenol; Praz, prazosin; Prop, propranolol. **e**, Schematic of AAV-mediated *Grm3* (encoding mGluR3) expression. **f,g**, Convergent NA and glutamate stimulation enhanced c-Fos in mGluR3⁺ astrocytes, but not in mGluR3⁻ astrocytes (*n* = 98–131 cells per condition). One-way ANOVA, Tukey's test. Scale bar, 20 μm. **h**, Schematic of signalling cascade. α₁R, α₁-receptor; βR, β-receptor. **i**, Ionomycin (Iono)-induced increase in Ca²⁺ increased c-Fos when combined with NA, isoproterenol (β-receptor agonist) or forskolin (*n* = 41–59 cells per condition). One-way ANOVA,

Tukey's test. **j**, Schematic of AAV injection into amygdala (LA/B) and LC in TRAP2 mice. Right, hM3Dq expression. Scale bars, 1 mm. **k–m**, Dual stimulation of LA/B and LC increased astrocyte density (LC–/LA/B–: *n* = 5; LC+/LA/B–: *n* = 4; LC–/LA/B+: *n* = 6; LC+/LA/B+: *n* = 5 mice). **k**, Images of hM3Dq-expressing cells. Scale bar, 200 μm. **l**, *Fos* co-expression in hM3Dq⁺ cells. Scale bar, 10 μm. **m**, Quantification of *Fos*-tagged astrocytes (mNG⁺) localized near hM3Dq⁺ LA/B neurons and NA fibres in **l** (representative of five biological replicates from the LC+/LA+ group). One-way ANOVA, Dunnett's test. **n–q**, FR-BAE tagging with β-receptor blockade (propranolol, 10 mg kg⁻¹) (**n**), LC silencing (**o**), amygdala (Amy) silencing (**p**) or RAM-based engram silencing (**q**) reduced BAE density. Scale bars (**n**,**o–q**, top and middle), 1 mm. **o–q**, Bottom image, magnified view showing local hM4Di-mCherry in neurons. Scale bars, 40 μm. Dox, doxycycline. **r**, FR-BAE density was reduced in all perturbation groups (Prop–/hM4Di–: *n* = 4; Prop+/hM4Di–: *n* = 4; LC: *n* = 4; amygdala (Amy): *n* = 5; engram: *n* = 4 mice). One-way ANOVA, Dunnett's test. Data are mean ± s.e.m. Sample sizes and replicates in Supplementary Table 1.

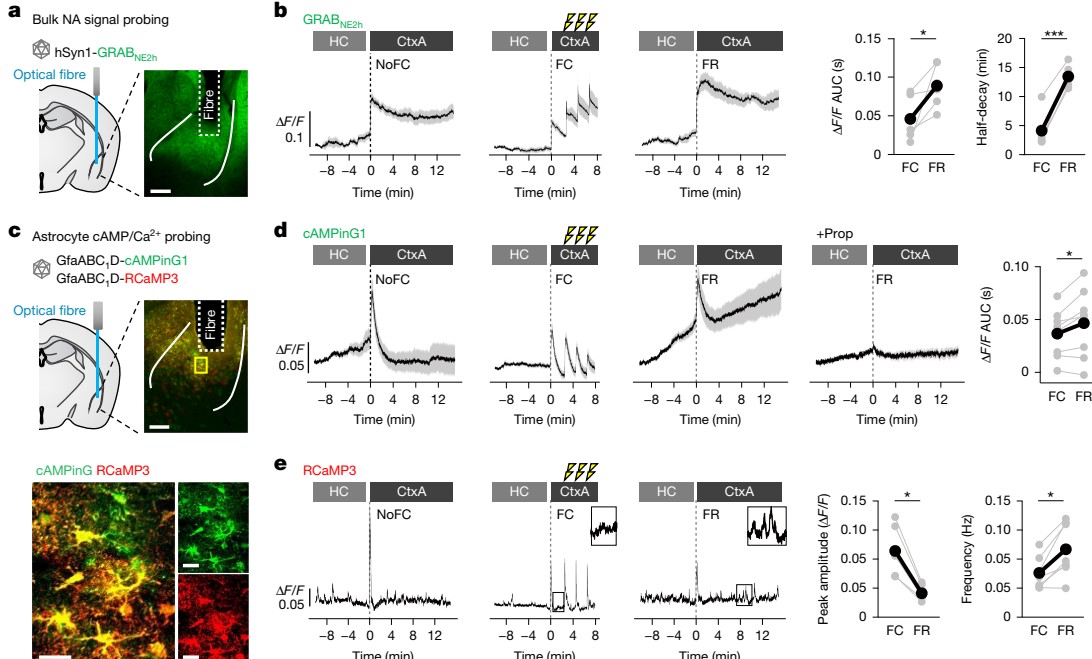

**Fig. 3 | NA, astrocyte cAMP and Ca²⁺ signals before, during and after fear experiences. a**, AAV injection of GRAB$_{NE2h}$ with optical fibre above LA/B to monitor bulk NA response (representative image from 6 mice; scale bar, 200 μm). **b**, ΔF/F traces showing NA signals in home cage (HC) and CtxA on fear conditioning (FC) day. NA signal increased with CtxA exposure and foot shocks, and showed larger, longer responses during recall (FR). Right, summary plots ($n = 5$–$6$ mice). AUC, area under the curve. **c**, Top, AAV injection of astrocyte-selective cAMPinG1 and RCaMP3 with optical fibre above LA/B to monitor astrocyte cAMP and Ca²⁺. Bottom, co-expression of cAMPinG1 and RCaMP3 in LA/B astrocytes (representative image from 8 mice; scale bars: 200 μm

(main image), 20 μm (magnified views, bottom)). **d**, Astrocyte cAMP signals increased with context exposure and shocks, and during FR they increased gradually from HC baseline and were further enhanced in CtxA. Propranolol (10 mg kg⁻¹, β-receptor antagonist) abolished these effects. Right, summary plot ($n = 8$ mice). **e**, Astrocyte Ca²⁺ signals also increased with context exposure and shocks. During FR, Ca²⁺ amplitude was reduced but event frequency was significantly higher. Right, summary plots ($n = 7$ mice). Wilcoxon signed-rank test. Data are mean ± s.e.m; in some cases, error bars are smaller than symbols. Sample sizes and number of replicates in Supplementary Table 1.

(α₁-receptors) or Gα$_i$-coupled α₂-adrenoreceptors, respectively (Fig. 2d). The astrocyte c-Fos induction was resistant to the Ca²⁺ chelator 1,2-bis(2-aminophenoxy)ethane-N,N,N′,N′-tetraacetic acid (BAPTA-AM)[37]. The membrane-permeable Ca²⁺ ionophore ionomycin did not increase astrocyte c-Fos signals (Fig. 2d). However, the inhibition of cAMP production by the adenylate cyclase inhibitor SQ22536 attenuated astrocyte c-Fos signals induced by the adenylate cyclase–cAMP activator forskolin, the β-receptor agonist isoproterenol or NA (Fig. 2d). Together, these data show that NA induces c-Fos signals in primary astrocytes mainly through Gα$_s$-coupled β-receptors and subsequent increases in cAMP.

Astrocytes in vivo respond to both local circuits and long-range neuromodulators[15]. Our data (Fig. 1g and Extended Data Fig. 3d) suggest that local activity also contributes to Fos induction in vivo. Given the transcriptional differences between in vitro and in vivo astrocytes, notably the reduced expression of metabotropic glutamate receptor 3 (mGluR3, encoded by *Grm3*) in cultured astrocytes[17,38], which is abundantly expressed in adult in vivo astrocytes[39], we heterologously expressed mGluR3 in primary astrocytes via AAVs to mimic the in vivo context (Fig. 2e). In mGluR3-expressing astrocytes, glutamate alone was ineffective, but in combination with NA it strongly enhanced c-Fos intensity (Fig. 2f,g). No such synergy was seen in mGluR3-negative astrocytes. Because Gα$_i$-coupled GPCRs such as mGluR3 can enhance Gβγ–Ca²⁺ signalling when Gα$_q$–phospholipase Cβ is co-activated[29,40], NA may recruit α₁-receptors (Gα$_q$) and mGluR3 (Gα$_i$) to cooperate with β-receptor–cAMP pathways (Fig. 2h). Supporting this, ionomycin-induced Ca²⁺ signals enhanced c-Fos when combined with NA, isoproterenol or forskolin (Fig. 2i). Thus, astrocytic c-Fos is driven primarily by β-receptor–cAMP and is further potentiated by convergent glutamate and NA signalling.

To test this convergence in vivo, we expressed neuron-specific hM3Dq in LA/B and locus coeruleus (LC), which mainly consists of noradrenergic neurons to stimulate them with CNO (Fig. 2j–m). The number of *Fos*-mNG⁺ astrocytes remained low when only LA/B or LC neurons were activated, but increased markedly when both were stimulated (Fig. 2j–m). This provides in vivo evidence that coincident local and noradrenergic inputs induce astrocytic *Fos*.

We next tested the necessity of local and far-reaching NA neural signals during FR-BAE tagging. TRAP2::AAV-PHP.eB-GfaABC₁D-DIO-mNG mice received one of four perturbations at the time of tagging: β-receptor antagonism with propranolol (Fig. 2n) or hM4Di-mediated silencing of LC neurons (Fig. 2o), amygdalar neurons (Fig. 2p) or amygdalar engram neurons tagged with the robust activity marking (RAM) system[41,42] whose hM4Di-based silencing (Extended Data Fig. 8) is known to reduce fear memory recall[41] (Fig. 2q). In all conditions, FR-BAE density decreased significantly (Fig. 2r), demonstrating dependence on both LC-NA and local engram activity.

To further test whether FR-BAEs reside in anatomical contexts where they could receive convergent inputs from both noradrenergic projections and local engram neurons, we performed dual-colour *Fos* tagging with neuron-specific DIO-mCherry and astrocyte-specific DIO-mNG AAVs in TRAP2 mice, combined with noradrenaline transporter (NET) immunostaining to visualize noradrenergic projections (Extended Data Fig. 9). Territories of FR-BAEs showed greater overlap with noradrenergic projections and FR engram neurons than those of non-FR-BAEs, with no change in astrocyte territory size. These findings indicate that FR-BAEs are positioned to integrate convergent inputs from noradrenergic projections and local engram neurons, supporting the model that their coincidence drives FR-BAE induction in the amygdala.

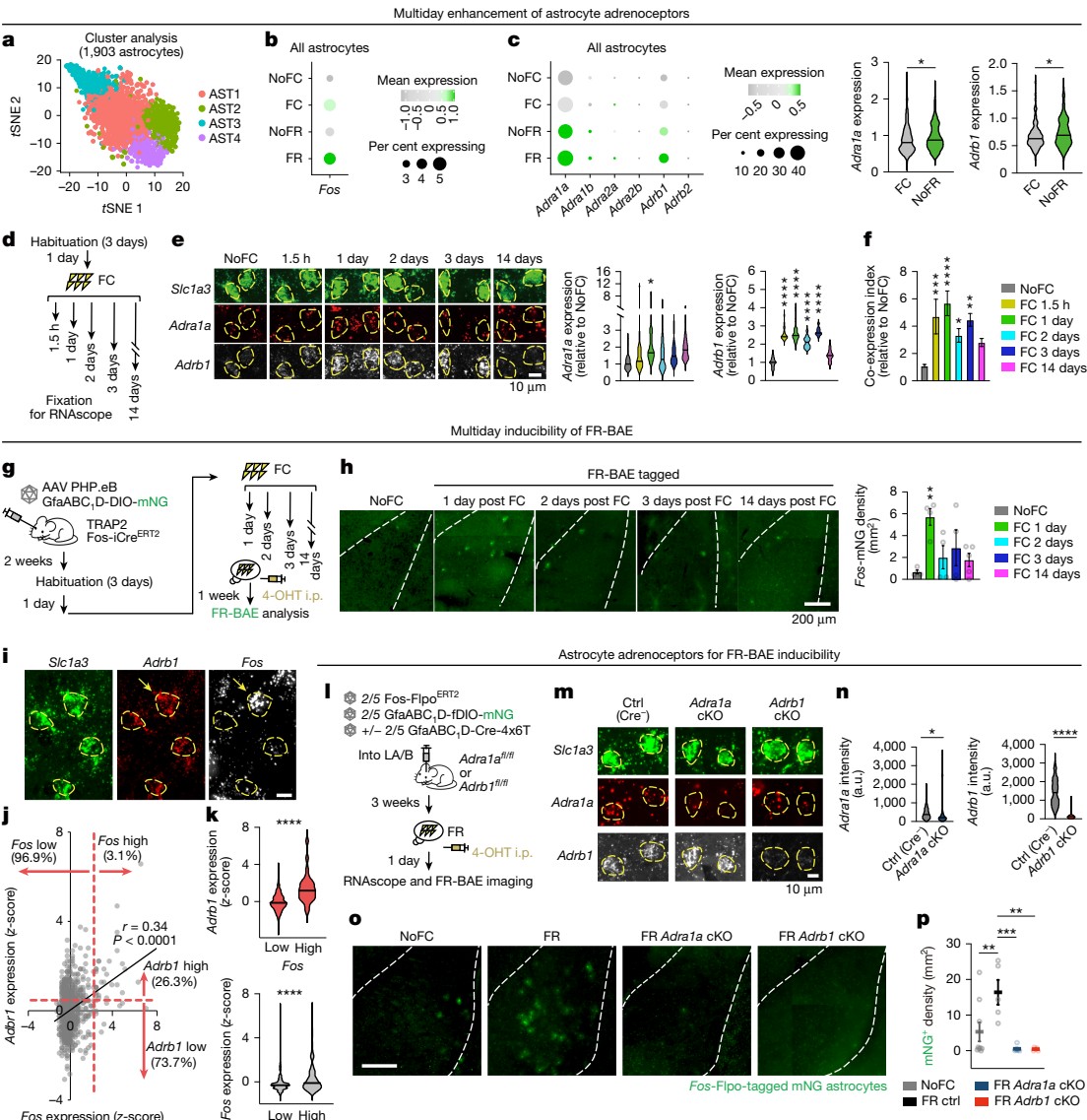

Multiday enhancement of astrocyte adrenoceptors

Multiday inducibility of FR-BAE

Astrocyte adrenoceptors for FR-BAE inducibility

**Fig. 4 | Adrenoreceptor upregulation after fear conditioning enables FR-BAE induction. a**, *t*-Distributed stochastic neighbour embedding (*t*SNE) clustering of scRNA-seq data from amygdalar astrocytes across NoFC, FC, NoFR and FR conditions (*n* = 4 mice per group). **b**, Dot plots showing *Fos* expression. Dot colour indicates *z*-score (grey, depletion; green, enrichment). **c**, Left, dot plots showing adrenoreceptor gene expression. Undetected receptors (*Adra1d*, *Adra2c* and *Adrb3*) are not shown. Right, *Adra1a* and *Adrb1* upregulation in NoFR versus FC. Horizontal lines represent median. Mann–Whitney test. **d**, RNAscope timeline. **e**, RNAscope images and violin plots of *Adra1a* and *Adrb1* in *Slc1a3*⁺ astrocytes (18–28 cells from 5 mice per group). One-way ANOVA, Dunnett's test versus NoFC. **f**, Co-expression index for *Adra1a* and *Adrb1* (Methods) peaks at 1 day post-FC. One-way ANOVA, Dunnett's test versus NoFC. *n* = 18–28 cells, 5 mice per group. **g**, BAE tagging timeline. **h**, Representative images and quantification of tagged FR-BAE, peaking at 1 day post-FC. *n* = 4–5 mice per group. One-way ANOVA, Dunnett's test versus NoFC.

**i**, RNAscope of *Adrb1* and *Fos* in *Slc1a3*⁺ astrocytes (FR). Arrows indicate a *Fos* and *Adrb1*-high cell (representative images from 18 sections, 6 mice). Scale bar, 10 μm. **j**, Pearson's correlation coefficient (*r*) for *Fos* and *Adrb1* expression (*z*-scores). *Fos*-high: *z* > 2 (3.1% cells); *Adrb1*-high: *z* > 0.5 (26.3%). Rationale in Methods. *n* = 814 cells from 6 mice. **k**, *Adrb1* expression is increased in *Fos*-high astrocytes (25 *Fos*-high cells, 789 *Fos*-low cells) and *Fos* is enriched in *Adrb1*-high astrocytes (215 *Adrb1*-high cells, 600 *Adrb1*-low cells). Mann–Whitney test. **l**, Schematic of *Adra1a* or *Adrb1* conditional knockout (cKO). **m**, RNAscope of *Adra1a* and *Adrb1* in *Slc1a3*⁺ astrocytes. Ctrl, control. **n**, Violin plots showing reduced targeted receptor after conditional knockout (77–117 cells, 3 mice per group). Horizontal lines represent the median. **o**,**p**, Images (**o**) and quantification (**p**) of FR-BAEs, showing decreased density following *Adra1a* or *Adrb1* conditional knockout. Scale bar, 200 μm. *n* = 5–9 mice, One-way ANOVA, Dunnett's test versus FR controls. Data are mean ± s.e.m. Sample sizes and replicates in Supplementary Table 1.

## Recall-evoked NA–astrocyte signalling

Our data (Fig. 2) support a convergence model in which NA and engram activity induce astrocyte *Fos*. However, since both activities occur during FC and FR[8,43], the stronger astrocyte *Fos* induction during FR than FC (Fig. 1) raises the question of how astrocytes generate this selective response. To address this, we examined NA release and astrocyte cAMP–Ca²⁺ signalling dynamics across fear learning and recall using

in vivo fibre photometry (Fig. 3 and Extended Data Fig. 8). We deployed AAV2/9-hSyn1-GRAB_NE2h[44] to monitor bulk NA signals in the LA/B of freely behaving mice (Fig. 3a,b). During habituation 24 h prior to FC, NA signals were observed when the mice were placed into context A (CtxA; foot shock chamber) (peak $\Delta F/F$ = 0.142 ± 0.017), probably reflecting mild stressors and/or locomotion[44] (NoFC; Fig. 3b). On top of the contextual NA signals, single foot shocks evoked enhanced and relatively fast (peak $\Delta F/F$ = 0.206 ± 0.04; half-decay = 4.3 min) NA responses in LA/B

(FC; Fig. 3b). In addition, further enhanced (peak $\Delta F/F = 0.211 \pm 0.022$) and prolonged (half-decay = 13.5 min) NA signals were observed during the contextual FR (FR; Fig. 3b). The prolonged NA signals were absent in context B (CtxB) (Extended Data Fig. 10c,d), a neutral, non-conditioned context where mice show no freezing (Extended Data Fig. 10b), indicating that recall of contextual fear drives sustained NA signals.

We next used dual-colour fibre photometry (AAV2/5-GfaABC$_1$D-cAMPinG1 and AAV2/5-GfaABC$_1$D-RCaMP3[45]) to measure astrocyte cAMP and Ca$^{2+}$ signals (Fig. 3c). Consistent with the NA signals (Fig. 3a,b), cAMPinG1 signals increased during habituation and foot shocks, which fully decayed within about 2 min (NoFC and FC; Fig. 3d). On the FR day, however, astrocytes displayed a progressive increase in cAMP (FR; Fig. 3d), which was absent in CtxB (Extended Data Fig. 10e,f). β-receptor antagonism (propranolol) in vivo abolished the astrocytic cAMP signalling during FR (Fig. 3d), which suppressed FR-BAE induction (Fig. 2n,r) and freezing (Extended Data Fig. 10b′). Foot shock elicited astrocyte Ca$^{2+}$ signals as reported previously[46], and Ca$^{2+}$ event frequency increased during FR (Fig. 3e). These data collectively indicate that fear experiences induce the enhancement of NA signals within the amygdala, which strongly trigger astrocyte cAMP signals through β-receptors during FR.

## Multiday astrocyte state changes

The increased astrocyte cAMP–Ca$^{2+}$ signalling during FR raised the question of how astrocytes mount stronger responses during recall than during conditioning, despite NA release and engram neural activity occurring in both fear experiences. To address this, we examined experience-dependent molecular changes in astrocytes under NoFC, FC, NoFR and FR conditions. We dissected amygdala tissue 90 min after behavioural testing and isolated astrocytes by fluorescence-activated cell sorting (FACS) using the ATPase Na$^+$/K$^+$ transporting subunit β2 (ATP1B2)[47] antibody ACSA2 (Extended Data Fig. 11a) and performed single-cell RNA sequencing (scRNA-seq). After quality control, we obtained profiles of 1,903 astrocytes (Fig. 4a and Extended Data Fig. 11b,c) and identified 4 transcriptionally distinct astrocyte clusters (AST1–4; Fig. 4a, Extended Data Fig. 11d,e and Supplementary Table 3). We did not observe any noticeable shifts in the subcluster composition across conditions, suggesting that FC and FR do not alter cluster identity. Fos-positive astrocytes comprised 2.9–4.2% of total astrocytes (Fig. 4b), consistent with earlier data (Extended Data Fig. 4). Of note, FR astrocytes showed the highest Fos levels (Fig. 4b), in line with BAE tagging (Fig. 1). Whereas scRNA-seq provides a snapshot at a single time point (approximately twofold increase in FR versus FC; Fig. 4b), the Fos-TRAP system integrates promoter activity over hours, which is likely to account for the greater fold differences observed with tagging (more than 30-fold in FR versus FC; Fig. 1d,e).

Subsequent analyses revealed that Adra1a (α$_1$-receptor) and Adrb1 (β$_1$-adrenoreceptor) as the two dominant adrenoreceptor genes expressed in amygdalar astrocytes. Notably, both receptors were upregulated in the FC group (1.5 h post-FC) and further elevated in the NoFR group (24 h post-FC without recall), suggesting continued priming of astrocytes over time (Fig. 4c). This data suggests a model in which FC induces a molecular state change in astrocytes, with a latency, increasing adrenoreceptor expression to potentiate NA signalling and enable future repeated fear-evoked Fos induction during FR. The enhancement in astrocytic adrenoreceptor expression is likely to underlie the NA–β-receptor-driven cAMP signals during FR (Fig. 3c,d) and Fos induction in response to recall of strong fear memory a day after conditioning (Fig. 1). To test this model, we explored temporal changes in adrenoreceptor expression, their link to FR-BAE induction, and causal roles in Fos induction.

We examined the time course of Adra1a and Adrb1 expression following FC, using single-cell RNAscope on amygdalar sections collected at 1.5 h, 1 day, 2 days, 3 days and 14 days post-conditioning (Fig. 4d–f). Adrb1 expression was significantly increased from 1.5 h to 3 days post-conditioning (approximately twofold versus NoFC; Fig. 4e), but returned to baseline by 14 days. By contrast, Adra1a expression was increased at 1 day post-conditioning (Fig. 4e), showed partial decay by 2 to 3 days, and peaked again at 14 days. We calculated a co-expression index for each cell (relative Adrb1 expression × relative Adra1a expression; Methods) and found a peak at 1 day post-FC (Fig. 4f), followed by gradual decline. To test whether this molecular profile predicts the degree of astrocyte Fos induction during recall, we performed FR-BAE tagging at multiple time points post-FC (1 day, 2 days, 3 days and 14 days; Fig. 4g,h). Consistent with the adrenoreceptor co-expression time course (Fig. 4f), FR-BAE density peaked at 1 day and declined thereafter (Fig. 4h), suggesting a defined window for FR-BAE induction. Moreover, RNAscope further revealed strong Fos–Adrb1 correlation in FR astrocytes (Fig. 4i–k).

To determine whether these adrenoreceptors are causally required for FR-BAE induction, we performed astrocyte-specific knockdown of Adra1a or Adrb1 in the amygdala (Fig. 4l) using floxed mouse lines[48–50], AAV2/5-GfaABC$_1$D-Cre-4x6T[51] and Flp-based Fos-tagging AAVs (AAV2/5-Fos-Flpo$^{ERT2}$ and AAV2/5-GfaABC$_1$D-fDIO-mNG). Knockdown of either receptor (Fig. 4m,n) significantly reduced FR-BAE density (Fig. 4o,p), indicating that both Gα$_q$-coupled α$_1$-receptors and Gα$_s$-coupled β$_1$-receptors are necessary for FR-BAE induction in vivo. Our earlier finding that the α$_1$-receptor antagonist prazosin did not block NA-induced c-Fos induction in cultured astrocytes (Fig. 2d) probably reflects the downregulation of Adra1a in vitro[17], highlighting the importance of in vivo validation. Collectively, our data support a model in which initial fear experience primes astrocytes to upregulate adrenoreceptor expression over time, creating a multiday molecular trace that allows future NA release upon repeated experiences (that is, recall) to selectively recruit the astrocyte ensemble.

## The astrocyte ensemble stabilizes memory

Transcriptomics enabled us to probe molecular changes of FR-BAEs linked to functional outputs. We mined differentially expressed genes between Adrb1-positive and Adrb1-negative astrocytes (Extended Data Fig. 12a) and identified Igfbp2 as highly upregulated in Adrb1-positive astrocytes (Extended Data Fig. 12a,b). Our single-cell RNAscope revealed highly correlated expression of Adrb1 and Igfbp2 (Extended Data Fig. 12c–e), suggesting that astrocytic Adrb1 upregulation could promote Igfbp2 induction. Igfbp2 encodes the secreted protein IGFBP2, which was recently reported to be enriched in 'peri-neuronal engram' Fos$^+$ astrocytes of the amygdala after fear recall[22].

Given that IGFBP2 and its binding partners, insulin-like growth factors (IGFs), are crucial for synaptic plasticity[52,53] and that chronic deletion of IGFBP2 disrupts synaptic transmission, circuit maturation and memory[22,52,54], we hypothesized that IGFBP2 expression in FR-BAE$^{LA/B}$ contributes to circuit de-stabilization and re-stabilization upon recall[2,3]. We tested this hypothesis with a spatiotemporally specific in vivo pharmacological perturbation of IGFBP2 (Extended Data Fig. 12f–h). We bilaterally infused an IGFBP2-neutralizing antibody[52] (IGFBP2-Ab; 1 mg ml$^{-1}$) immediately after FR in CtxA—that is, within the reconsolidation window during which memories become labile to be weakened or strengthened[2]. Mice received repeated daily infusions and FR sessions for five days (Extended Data Fig. 12f). IGFBP2-Ab treatment reduced freezing compared with controls (Extended Data Fig. 12g,h), indicating impaired memory stabilization; controls infused with IgG showed no change. Combined, our transcriptomics, RNAscope and pharmacological perturbation data indicate that a subset of astrocytes increases adrenoreceptor expression upon fear experiences, enhancing NA responsiveness and recall-driven induction of Fos and Igfbp2 to stabilize fear memory.

To directly examine the causal role of FR-BAE signalling in memory stabilization, we performed a loss-of-function experiment selectively

targeting FR-BAEs in LA/B. We used iβARK, a genetically encoded 122-amino acid peptide that binds $G\alpha_q$–GTP to suppress GPCR signalling[26] (Extended Data Fig. 13a). In ex vivo $Ca^{2+}$ imaging in amygdalar astrocytes, iβARK reduced $\alpha_1$-receptor-evoked $Ca^{2+}$ responses relative to its mutant control, although some residual oscillatory activity persisted (Extended Data Fig. 13c,f,g), consistent with our prior findings in the striatum[26]. To improve efficacy, we developed iβARK2 by fusing the original peptide to a membrane-anchoring Lck domain, enabling more effective proximity-based inhibition (Extended Data Fig. 13a,b). iβARK2 showed robust membrane localization across astrocytic compartments, including fine processes (Extended Data Fig. 13d). Astrocytes expressing iβARK2 displayed significantly attenuated $Ca^{2+}$ responses relative to both iβARK and the control iβARK2mut (Extended Data Fig. 13e–g). We next generated Cre-dependent AAV versions of iβARK2 and iβARK2mut for use in Fos-iCre[ERT2] TRAP mice to selectively target FR-BAEs (Fig. 5a,b). Remarkably, silencing FR-BAEs with iβARK2 impaired memory stabilization upon repeated recall (Fig. 5c), consistent with the behavioural effects observed upon neutralization of astrocyte-derived IGFBP2 (Extended Data Fig. 12f–h). Moreover, astrocytic IGFBP2 levels were significantly reduced in iβARK2-expressing mice compared with iβARK2mut controls (Fig. 5d), suggesting that suppression of GPCR signalling in FR-BAEs disrupts the recall-dependent transcriptional induction of IGFBP2 and compromises memory stabilization.

## Astrocyte–neuron engrams modulate memory

To further explore the causal impact of BAE signalling on neuronal engram and memory formation, we performed a gain-of-function experiment targeting astrocytic $\beta_1$-receptors in LA/B. Astrocytes transduced with AAV2/5-GfaABC1D-*Adrb1* for *Adrb1* overexpression (*Adrb1*-OE) showed significantly increased *Igfbp2* expression compared with controls (Fig. 5f). *Adrb1* overexpression also increased the density of FR-BAEs (*Fos*-mNG⁺ astrocytes) by around threefold relative to controls (Fig. 5g,h), indicating that astrocyte $\beta_1$-receptor overexpression amplifies the recall-induced astrocyte ensemble.

We next functionally assessed the behavioural effect of the augmented astrocyte ensembles (Fig. 5i–l and Extended Data Fig. 14). We detected no differences in ambulation in an open arena or time spent in the centre zone during habituation between control and *Adrb1*-OE mice (Extended Data Fig. 14a,b). We then tested whether *Adrb1* overexpression affected memory stabilization. With three-shock conditioning, both groups showed equivalent freezing during initial recall and second recall after reconsolidation (Fig. 5j), potentially owing to a ceiling effect. To create parametric space to detect enhancements in memory, we trained mice to a weak (one-shock) conditioning paradigm (Fig. 5i,k). *Adrb1*-OE mice did not show altered freezing during FR one day after FC; however, *Adrb1*-OE mice exhibited significantly increased freezing during the second recall (Fig. 5k), indicating enhanced memory stabilization following the reconsolidation after initial recall. In another cohort with three-shock conditioning, initial recall showed no significant difference in freezing between the groups (Extended Data Fig. 14i,j). However, we observed significantly enhanced freezing in an altered context (CtxB) in mice with *Adrb1* overexpression compared with controls (Fig. 5l). These data suggest that overstabilization of the initial fear memory leads to generalization or a loss of memory precision[41,55–57].

Finally, we explored whether fear generalization is accompanied by activity of engram neurons. We bilaterally injected AAV2/8-hSyn1-DIO-mCherry with and without AAV2/5-GfaABC1D-*Adrb1*, and tagged engram neurons during FR. Mice were euthanized 1.5 h after the CtxB freezing test and the reactivation of FR engram neurons was assessed by c-Fos immunoreactivity (Fig. 5m–o). The overall number of mCherry⁺ neurons (FR engram neurons) was not significantly different between groups, but the c-Fos levels in FR engram neurons in the *Adrb1*-OE mice, which showed fear generalization, were significantly higher than those in control mice (Fig. 5o). These findings suggest that the larger astrocyte ensembles contribute to enhance subsequent reactivation of engram neurons, consistent with the view that memory retrievability is not determined during memory formation, but may be regulated after memory retrieval[8], and demonstrating that astrocytes are involved in that crucial process.

## Discussion

These data demonstrate that astrocytes can link discrete fear experiences across at least a day-long timescale. As postulated more than a century ago[58] and recently implicated in brain computation[11,25,59,60], astrocytes may integrate information on slower timescales than neurons. Whereas neurons and astrocytes have been traditionally compared over milliseconds versus seconds-to-minutes, our findings reveal a functionally significant mismatch on the scale of days. Unlike neurons that express the immediate early genes during both FC and FR[5,8,28], astrocytes respond to fear conditioning (FC) with rapid $Ca^{2+}$ and cAMP increases[61], but undergo slower transcriptional state changes[62]. Our single-cell transcriptomics and RNAscope revealed that multiple days post-FC, without recall, astrocytes upregulated *Adra1a* and *Adrb1*, marking a learning-dependent state that facilitates heightened responses during repeated experiences (FR). This in turn drives expression of *Fos* and *Igfbp2*, supporting memory stabilization. The astrocyte ensembles identified in this study may partially meet the criteria for engram cells[8] that undergo enduring biochemical changes upon experiences. However, our findings suggest that the astrocytic ensemble may not serve as a direct memory trace encoding experience-specific information like neurons because its perturbation did not alter initial fear recall. Instead, it may function as an eligibility trace for memory stabilization over days, creating a permissive microenvironment within circuits that supports the stabilization of memories related to repeated experiences. From a computational and bioenergetic perspective, astrocytes may underlie retrospective linking[63,64], which benefits from spaced intervals over days between learning sessions[65] by providing a slow modulatory trace that complements fast neuronal activity[64], thereby offloading the energetically costly task of maintaining firing patterns in neurons across days and enabling more stable circuit transformations[60].

Mounting evidence from animal[3,4,57] and human imaging[6] studies has highlighted the amygdala and NA in modulating long-term memory. Early pharmacological studies targeting protein synthesis[66], β-receptors[67], PKA[35] or cAMP–CREB[34] in the amygdala disrupted memory re-stabilization, which our findings indicate could partially be mediated by astrocytes. We show that astrocytes undergo learning-dependent increases in adrenoreceptor expressions, and manipulating these receptors alters memory stability, precision and neuronal engram representations. Notably, both NA input and reactivated neuronal engram activity are required to recruit astrocytic ensembles that stabilize labile memories. These findings imply that astrocyte ensembles may not simply be passive bulk substrates of NA signals, but instead exert behaviourally relevant, circuit-specific regulations through astrocyte-defined microcircuitry[15,68]. Disrupting the precise NA–astrocyte signalling through β1R overexpression led to fear generalization, suggesting physiological astrocytic roles in selective memory stability that constrain maladaptive updating. In this regard, targeting these astrocyte–neuron interaction mechanisms could be beneficial in treating cognitive and neuropsychiatric dysfunctions, including post-traumatic stress disorder, in which hyperactivity of the NA system is a prominent feature[4,69].

Deciphering how diverse astrocyte responses link to multiple circuit outputs has long been a challenge[15]. Considerable effort has been devoted to probe[24] or perturb[11] astrocyte $Ca^{2+}$ signals since they were first reported[70], yet causal relationships are usually inferred from separate subjects, leaving input–output links unclear. Astrocyte

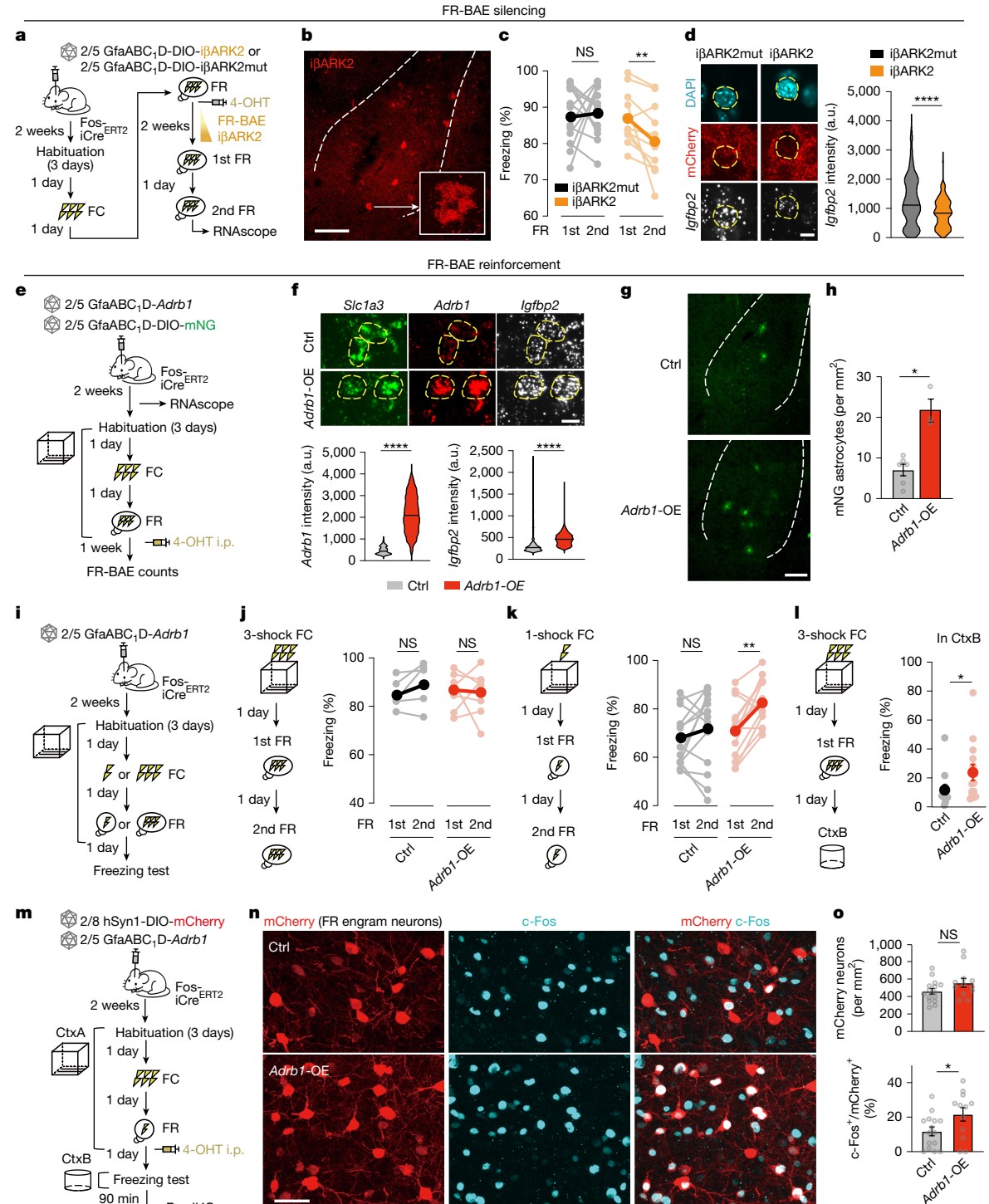

**Fig. 5 | FR-BAE perturbations bidirectionally regulate memory stability and precision. a**, Schematic of behavioural and RNAscope analysis. **b**, *Fos*-iβARK2[+] astrocytes in LA/B. Inset, enlarged view of an iβARK2[+] astrocyte (representative of 12 mice). Scale bar, 200 μm. **c**, Reduced freezing at second recall in iβARK2 versus sustained freezing in iβARK2mut control (iβARK2mut: *n* = 13; iβARK2: *n* = 11 mice). Wilcoxon matched-pairs signed rank test. **d**, RNAscope images of *Igfbp2* with mCherry–iβARK2 or mCherry–iβARK2mut and violin plots showing lower *Igfbp2* expression in the presence of iβARK2 (iβARK2mut: *n* = 23; iβARK2: *n* = 24 cells; 4 mice per group). Mann–Whitney test. Horizontal lines represent the median. Scale bar, 10 μm. **e**, FR-BAE tagging with astrocytic *Adrb1* overexpression. **f**, RNAscope images of *Adrb1* and *Igfbp2* in *Slc1a3*[+] astrocytes with or without *Adrb1* overexpression. Graphs showing increased expression with *Adrb1*-OE (ctrl: 501; *Adrb1*-OE: 1,810 cells; 18 sections from 6 mice per group). Mann–Whitney test. Scale bar, 10 μm. **g**,**h**, Images (**g**) and quantification (**h**)

showing higher FR-BAE density with *Adrb1* overexpression. Mann–Whitney test. Scale bar, 200 μm. **i**, Memory tests under weak (one-shock) versus strong (three-shock) conditioning. **j**, *Adrb1* overexpression has no effect on freezing behaviour under strong conditioning (ctrl: *n* = 5; *Adrb1*-OE: *n* = 7 mice). Wilcoxon matched-pairs signed rank test. **k**, Under weak conditioning, *Adrb1* overexpression enhanced freezing at second recall (*n* = 13 mice per group). Wilcoxon matched-pairs signed rank test. **l**, After strong conditioning, *Adrb1* overexpression caused greater freezing in CtxB versus control (ctrl: *n* = 13; *Adrb1*-OE: *n* = 14 mice). Mann–Whitney test. **m**, FR engram labelling with mCherry and c-Fos immunohistochemistry 90 min after freezing test with or without *Adrb1* overexpression. **n**,**o**, FR engram neurons exhibit similar density but higher reactivation with *Adrb1* overexpression (ctrl: *n* = 15; *Adrb1*-OE: *n* = 12 slices; 3 mice per group). Scale bar, 50 μm. Mann–Whitney test. Data are mean ± s.e.m. Sample sizes and replicates in Supplementary Table 1.

perturbation in a bulk manner is also incongruent with the heterogeneity of astrocyte identity[49,71], state[62,72] and response[59,73]. Our approaches bridge this gap: by exploiting context-specific astrocyte state changes[18,20,22], the brain-wide BAE tagging method enables broad exploration of ensembles causal for neural processing and behaviour. This framework facilitates research into astrocytic ensembles across diverse behaviours and disease conditions. Expanding such efforts will accelerate discoveries in computation and pathology that require understanding of multicellular interactions as a core attribute of brain biology.

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

## Methods

### Mouse models

Young adult (8- to 12-week-old) wild-type C57BL/6NCrSlc mice obtained from the Japan SLC, Fos2A-iCre[ERT2] (TRAP2[28], JAX 030323) mice obtained from the Jackson Laboratory, *Adra1a* flox mice[49,74] obtained from RIKEN (RBRC11837) and *Adrb1* flox mice[50] (accession number CDB1068K, generated by RIKEN Center for Biosystems Dynamics Research) were used for in vivo experiments. TRAP2 mice were bred and maintained in our animal facility, backcrossed with C57BL/6NCrSlc at least eight times before use for experiments. Wild-type C57BL/6NCrSlc female mice were obtained from the Japan SLC to obtain postnatal day 0–1 (P0–P1) pups. Experiments were done using both male and female mice. Number of mice used in each experiment is listed in the main text and/or figure legends accordingly. Mice were housed in groups of 2–5 in a temperature- and humidity-controlled room ($23 \pm 3$ °C, $45 \pm 5\%$ humidity) under a 12-h light:dark cycle (lights on from 07:00 to 19:00) and given ad libitum access to water and laboratory mouse diet at all times. After fibre implantation, the mice were single housed under a 12-h light:dark cycle (lights on from 07:00 to 19:00) and given ad libitum access to water and laboratory mouse diet at all times. After fibre implantation, the mice were singly housed. All experiments were approved by the RIKEN Animal Care and Use Committee.

### Cell lines

AAVpro 293T cells (Clontech, 632273) were maintained in Dulbecco's Modified Eagle Medium (DMEM; Gibco, 11995073) with fetal bovine serum (FBS; Cosmo Bio, CCP-FBS-BR-500) and 0.1% antibiotic-antimycotic (Gibco, 15240062). Cells were grown in a humidified cell culture incubation with 95% air, 5% $CO_2$ at 37 °C.

### Viral vector construction

All plasmid constructs were generated using standard molecular biology techniques and the In-Fusion HD Cloning Kit (Clontech). The pZac2.1 GfaABC$_1$D-Rpl22-HA plasmids (Addgene plasmid #111811) were digested with XhoI and NotI, which removed the Rpl22-HA sequence. Multiple cloning sequences (MCS) comprised of restriction enzyme sites that were absent from the backbone pZac2.1 plasmid were generated via gene synthesis (Eurofins Genomics) and incorporated to produce pZac2.1-GfaABC$_1$D-MCS plasmids. The PCR-amplified DIO-GFP sequence from Addgene plasmid #28304 was cloned into the mammalian expression vector pCAG-tdTomato (Addgene #83029) between EcoRI sites to generate pCAG-DIO-EGFP-tdTomato. The PCR-amplified DIO-EGFP-tdTomato was inserted into the XbaI and NotI sites of the pZac2.1-GfaABC$_1$D-MCS plasmid to generate pZac2.1-GfaBC$_1$D-DIO-EGFP-tdTomato plasmids. The PCR-amplified WPRE sequence from pFBAAV-CAG-XCaMP-R-WPRE (a gift from H. Bito[12]) was inserted into the NotI site of the resulting plasmid to generate pZac2.1-GfaABC$_1$D-DIO-EGFP-tdTomato-WPRE plasmids. The plasmids were digested with NotI and AgeI, which removed the tdTomato sequence, and a PCR-amplified mScarlet-I sequence (Addgene #98816) was incorporated, and the resulting plasmids were digested with XhoI and KpnI, which removed the EGFP sequence, and the mNG sequence from Addgene #99135 was ligated using DNA ligation kit (Takara) to generate pZac2.1-GfaABC$_1$D-DIO-mNG-mScarlet-I-WPRE plasmids. The mScarlet-I sequence was removed with restriction enzymatic digestion with PspOMI and NotI to generate pZac2.1-GfaABC$_1$D-DIO-mNG-WPRE plasmids. The PCR-amplified hM4Di-mCherry sequence from Addgene #92286 was cloned into the PmeI site of pAAV-RAM-d2TTA::TRE-MCS-WPRE (Addgene #63931) to generate pAAV-RAM-d2TTA::TRE-hM4Di-mCherry-WPRE plasmids. pAAV-GfaABC$_1$D-Flag-mAdrb1-WPRE, pAAV-GfaABC$_1$D-mGrm3-mCherry-WPRE, pAAV-GfaABC$_1$D-Lck-mCherry-iβARK(D110A), pAAV-GfaABC$_1$D-Lck-mCherry-

iβARK, pAAV-GfaABC$_1$D-DIO-Lck-mCherry-iβARK(D110A), pAAV-GfaABC$_1$D-DIO-Lck-mCherry-iβARK pAAV-Fos-Flpo[ERT2] and pAAV-GfaABC$_1$D-FRT-mNG plasmids were designed by the authors and constructed by VectorBuilder. The PCR-amplified DIO-Lck-mCherry-iβARK(D110A) and DIO-Lck-mcherry-iβARK sequence from pAAV-GfaABC$_1$D-DIO-Lck-mCherry-iβARK(D110A) and pAAV-GfaABC$_1$D-DIO-Lck-mCherry-iβARK was clone into the EcoRI site of pAAV-GfaABC$_1$D-MCS-4x6T-WPRE (Addgene #196417) to generate pAAV-GfaABC$_1$D-DIO-Lck-mCherry-iβARK(D110A)-4x6T and pAAV-GfaABC$_1$D-DIO-Lck-mCherry-iβARK-4x6T. pAAV-GfaABC$_1$D-mGrm3-mCherry-WPRE was digested with EcoNI and AccI to remove mGrm3-mCherry, and PCR-amplified mGrm3 was cloned back to the backbone to generate AAV-GfaABC$_1$D-mGrm3-WPRE plasmids. The PCR-amplified XCaMP-Gf-WPRE from the pFBAAV-CAG-XCaMP-Gf-WPRE plasmid (a gift from H. Bito[75]) was cloned into the NcoI and HindIII site of pAAV-hSyn1-cyto-mRuby3-iATPSnFR1.0 (Addgene #102557) to generate pAAV-hSyn1-XCaMP-Gf-WPRE plasmids. The mCherry-CAAX sequence digested with EcoRI of the pZac2.1-GfaABC$_1$D-mCherry-CAAX plasmid (a gift from T. Takano[76]) was ligated into the EcoRI and EcoRV the resultant plasmids to generate pAAV-hSyn1-mCherry-CAAX-WPRE plasmids. All constructs were sequenced before use. All our constructs have been deposited to Addgene in the Nagai Lab repository for unrestricted distribution (http://www.addgene.org/Jun_Nagai, Supplementary Table 4).

### Viral production

Recombinant AAVs were generated by triple transfection of AAVpro 293T cells (Clontech, 632273) using polyethylenimine (PEI; Polysciences inc., 24765-100). Cells were cultured up to 50–70% confluency in DMEM (Gibco, 11995073) with FBS (Cosmo Bio, CCP-FBS-BR-500) and 0.1% antibiotic-antimycotic (Gibco, 15240062). The plasmids and PEI were mixed with Opti-MEM (31985-088) and kept for 5 min at room temperature. The mixture was combined with a DNA mix which contained pAAV Capsids, pHelper (Agilent Technologies, 240071-54) and pAAV transgenes. All these plasmids had a concentration of 4.2 μg per dish. The DNA mix was then applied with PEI/Opti-MEM, and incubated for 20 min at room temperature. Then, this solution was added to the cells and incubated for 24 h. After incubation, the medium was changed to DMEM with 2% FBS and 0.1% antibiotic-antimycotic. Viral particles were collected from the medium at 120 h post-transfection. Cell pellets were centrifuged (3,500 rpm, 30 min, 4 °C). The supernatant was filtered by Setricup (Merc, S2HVU02RE) and pellets were resuspended in FBS-free DMEM and freeze-thawed three times. In the case of PHP.eB capsid virus production, the supernatant was resuspended in DMEM and fractured by harshly pipetting. Thawed pellets were centrifuged (3,500 rpm, 10 min, 4 °C) again and the supernatant was filtered by Setricup (Merc, S2HVU02RE). Media containing viral particles were centrifuged (22,000 rpm, 2 h, 4 °C) by Optima XE-100 (Beckman Coulter, A94516). The pellets were collected with 10–50 μl PBS and clarified by centrifugation (4,000 rpm, 10 min, 4 °C). The supernatant-containing viral particles were stored at 4 °C. AAV titre was analysed by quantitative PCR (QuantStudio 12 K; Applied Biosystems) using primers designed to selectively bind AAV2 inverted terminal repeats (forward: 5′-AACATGCTACGCAGAGAGGGAGTGG-3′, reverse: 5′-CATGAGACAAGGAACCCCTAGTGATGGAG-3′).

### Viral injections

All surgical procedures were conducted under general anaesthesia using continuous isoflurane (induction at 5%, maintenance at 1.5% to 2% vol/vol). Depth of anaesthesia was monitored continuously and adjusted when necessary. Intravenous administration of AAV-PHP.eB (50 μl to deliver $2.5 \times 10^{12}$ genome copies (GCs) per mouse) was performed by injection into the retro-orbital sinus of TRAP2 mice. Two weeks after AAV-PHP.eB GfaABC$_1$D-DIO-mNG-WPRE or AAV-PHP.

eBGfaABC$_1$D-DIO-mNG-mScaret-I-WPRE injections, mice were administered 4-OHT for inducing Cre recombination under the *Fos* promoter (*Fos* tagging) and then returned to their home cage until they were euthanized 7 days after tagging. In some cases, *Fos* tagging was performed using the FLP–FRT system (fDIO), and mice were euthanized one day after tagging. For stereotaxic microinjection of AAVs, mice were fitted into a stereotaxic frame (David Kopf Instruments) with their heads secured by blunt ear bars and their noses placed into a veterinary grade anaesthesia and ventilation system (VetEquip 911103). The surgical incision site was then cleaned with 70% ethanol (vol/vol). Craniotomies (1–2 mm in diameter) were made powered by a high-speed drill at specific coordinates based on the bregma position. Bilateral viral injections were conducted using a stereotaxic apparatus (David Kopf Instruments) to guide the placement of beveled glass pipettes (World Precision Instruments, 1B100-4) into the regions of interest: the basolateral amygdala (anterior–posterior (AP): −1.48 mm, medial–lateral (ML): 3.3 mm, dorsal–ventral (DV): 5.00 mm to bregma) and LC (AP: −5.4 mm, ML: 0.85 mm, DV: 3.7 mm to bregma). Total of 0.3 µl of AAV was delivered at 0.1 µl min$^{-1}$ using a syringe pump (SmartTouch Pump, World Precision Instruments). The glass pipette was withdrawn 5–10 min after the infusion to prevent the backflow. Mice were then sutured, returned to their home cage, and allowed to rest for at least two to three weeks before any assessments were conducted. AAV titres (GCs per ml) were adjusted with sterile saline to deliver the indicated genome copies into the regions of interest: AAV2/5-GfaABC$_1$D-DIO-mNG-WPRE ($3.0 \times 10^{12}$), AAV2/5-GfaABC$_1$D-hM3Dq-mCherry ($3.0 \times 10^{12}$), AAV2/DJ8-RAM-d2TTA::TRE-hM4Di-WPRE ($3.0 \times 10^{13}$), AAV2/5-GfaABC$_1$D-Flag-mAdrb1-WPRE ($3.0 \times 10^{12}$), AAV2/9-rTH-Cre ($3.0 \times 10^{12}$), AAV2/8-hSyn1-DIO-hM3Dq ($1.0 \times 10^{13}$), AAV2/8-hSyn1-DIO-hM4Di ($1.0 \times 10^{13}$), AAV2/8-hSyn1-DIO-mCherry ($3.0 \times 10^{12}$), AAV2/DJ8-Ef1a-hM3Dq ($4.8 \times 10^{11}$), AAV2/DJ8-Ef1a-hM4Di ($4.0 \times 10^{12}$), AAV2/5-hSyn1-mCherry-CAXX ($3.0 \times 10^{12}$) AAV2/9-hSyn1-GRAB$_{NE2h}$-WPRE ($2.7 \times 10^{12}$), AAV2/1 GfaABC$_1$D-cAMPinG1-NE ($1.3 \times 10^{13}$), and AAV2/1 GfaABC$_1$D-RCaMP3 ($1.5 \times 10^{13}$), GfaABC$_1$D-GCaMP6f ($1.1 \times 10^{13}$), AAV2/5-GfaABC$_1$D-iβARK-mCherry ($1.3 \times 10^{12}$), AAV2/5-GfaABC$_1$D-iβARK(D110A)-mCherry ($1.3 \times 10^{12}$), AAV2/5-GfaABC$_1$D-Lck-mCherry-iβARK ($1.3 \times 10^{12}$), AAV2/5-GfaABC$_1$D-Lck-mCherry-iβARK(D110A) ($1.1 \times 10^{13}$), AAV2/5-GfaABC$_1$D-DIO-Lck-mCherry-iβARK-4x6T ($3.0 \times 10^{13}$), AAV2/5-GfaABC$_1$D-Lck-mCherry-iβARK(D110A) ($3.0 \times 10^{13}$), AAV2/5-Fos-Flpo$^{ERT2}$ ($3.0 \times 10^{12}$), AAV2/5-GfaABC$_1$D-fDIO-mNG ($1.0 \times 10^{12}$), AAV2/5-GfaABC$_1$D-Cre-4x6T ($1.2 \times 10^{13}$).

## Drug administration in vivo

4-OHT (Sigma, H6278) was dissolved at 25 mg ml$^{-1}$ in 100% ethanol with agitation at 37 °C for 15 min and was then aliquoted and stored at −20 °C for up to several weeks until use. Before use, 4-OHT was redissolved in ethanol by shaking at 37 °C for 15 min and corn oil (Sigma, 259853 and S5007) was added to give a final concentration of 3 mg ml$^{-1}$ 4-OHT. The ethanol was evaporated by incubation at 50 °C. The final 3 mg ml$^{-1}$ 4-OHT solutions were always used on the day they were prepared. All 4-OHT injections were delivered intraperitoneally at 50 mg kg$^{-1}$ 2–4 weeks after AAV injection into TRAP2 mice for inducing Cre recombination under the *Fos* promoter.

CNO (Tocris, 4936) was administered intraperitoneally 0.5 h before the initiation of behavioural assessments. CNO was administered intraperitoneally 20 min before 4-OHT intraperitoneal injection. Propranolol (MedChemExpress, HY-B0573) was dissolved in dH$_2$O at 100 mM and stored at −20 °C. The final solutions were always used on the day they were prepared.

Mice were placed on doxycycline (40 mg kg$^{-1}$, Bio-Serv, F4159) chow 24 h prior to and for at least 1 week after viral injection, switched to doxycycline-free chow 48 h before recall, and euthanized 24 h later. IGFBP2 antibody was infused bilaterally through implanted guide cannulas targeting basolateral amygdala. Infusions were performed using an internal cannula connected to a 10-µl Hamilton syringe. One microgram of IGFBP2 antibody (1 mg ml$^{-1}$, 1 µl per site) was infused into mice immediately after a fear recall paradigm at 400 nl min$^{-1}$ using a syringe pump (SmartTouch Pump, World Precision Instruments).

## Immunohistochemical analysis

Mice were deeply anaesthetized with isoflurane and transcardially perfused with PBS, followed by 4% paraformaldehyde (PFA) in PBS for fixation. The dissected brains were post-fixed overnight at 4 °C in 4% PFA in PBS, and cryoprotected in 30% sucrose in PBS. The brains were embedded with OCT compound (Sakura Finetek Japan, 4583) and sliced into coronal sections with a thickness of 50 µm at −20 °C using cryostat (CryoStar NX70, Thermo Fisher Scientific, 956960). The sections were washed with PBS 3 times for 5 min each and stored in 50% glycerol (Fujifilm, 075-00616) in PBS at −20 °C. For staining with c-Fos antibody, the sections were permeabilized with PBS containing 1% Triton X-100 (Nacalai tesque, 35501), incubated at room temperature for 3 h, and then incubated with 0.3% Triton X-100 in PBS containing 1% normal donkey serum (NDS, Sigma-Aldrich, D9663) at room temperature for 1 h for blocking, and then incubated at 4 °C overnight in blocking buffer. With the following primary antibodies, Triton X-100 was used at 0.1% for permeabilization and PBS containing 1% NDS was used for blocking and antibody reactions: rat anti-GFAP (1:500, Thermo Fisher Scientific, 13-0300), mouse anti-S100β (1:500, Sigma-Aldrich, S2532), goat-Sox9 (1:100, R&D systems, AF3075) mouse anti-RFP (1:500, MBL, M155-3), rabbit anti-RFP (1:500, Rockland, 600-401-379), mouse anti-NET (1:2,000, MAb technologies, NET05-2) and rabbit anti-c-Fos (1:5,000, Synaptic Systems, 226003). After washing with PBS 3 times for 10 min each, the sections were incubated at room temperature for 2 h in blocking buffer with the following secondary antibodies: Alexa Fluor 488 donkey anti-rat, Alexa Fluor 488 donkey anti-chicken, Alexa Fluor 568 donkey anti-mouse, Alexa Fluor 568 donkey anti-rabbit, Alexa Fluor 568 donkey anti-goat, Alexa Fluor 647 donkey anti-rabbit, Alexa Fluor 647 donkey anti-mouse, Alexa Fluor 647 donkey anti-chicken (1:1,000, Abcam, ThermoFisher and Jackson ImmunoResearch). The sections were incubated at 4 °C overnight in blocking buffer with secondary antibodies when staining with anti-c-Fos antibody. After washing with PBS 3 times for 10 min each, the sections were mounted on slides (Fluoromount-G, Southern bio, 0100-35) and visualized under a confocal laser microscope (Olympus, FV3000) and a fluorescent microscope (Olympus, VS200). Immunoreactivities were analysed using ImageJ software (NIH). The numbers of c-Fos$^+$ neurons (Fig. 5o and Extended Data Fig. 8b) and astrocytes (Extended Data Fig. 4b) were quantified based on a consistent c-Fos intensity threshold for each experiment. This threshold was defined such that the number of c-Fos$^+$ neurons or astrocytes corresponded to the values previously reported[18,42,77]. Number of cells, field of views, sections and mice used in each analysis is listed in the main text and/or figure legends accordingly.

## Visual stimulation

Mice were singly housed in a light-proof chamber for 48 h. On the day of tagging, mice were exposed to 1 h of light inside the chamber and injected intraperitoneally with 50 mg kg$^{-1}$ 4-OHT either 24 h, 12 h, 6 h or 3 h before or 0 h, 3 h, 6 h, 12 h or 24 h after the termination of light exposure. Mice were returned to the dark chamber for an additional two days and then returned to a regular light-dark cycle until the time of euthanasia seven days after tagging.

## Contextual fear conditioning

Behavioural tests were performed during the light cycle between 07:00 and 19:00. Temperature and humidity of the experimental rooms were maintained at 23 ± 2 °C and 55 ± 5%, respectively. Background noise (55 dB) was generated by a white noise generator (Ohara).

All the experimental mice were transferred to the behaviour testing room at least 1 h before the tests to acclimatize to the environment and to reduce stress. Mice were habituated by freely exploring the conditioning chamber (CtxA) that consisted of a square partition (30 × 30 × 15 cm; 100 lux and no odour) with a grid floor wired to a shock generator (Ohara) for 15 min per day for 3 days. On the fourth day, they were either fear conditioned (1 time or 3 times shock; 2 s, 0.3 mA) (FC group) or presented with no shocks (NoFC group) in the conditioning chamber. One day after the conditioning, mice were either returned to CtxA for 3 min for fear memory recall (FR group) or kept in their home cages (NoFR group). Freezing durations exceeding 70% in the conditioned chamber 1 day post-conditioning with 3 foot shocks or exceeding 50% 1 day post-conditioning with a single foot shock, are considered indicative of sufficient conditioning. All the experimental mice received intraperitoneal injections of 4-OHT (50 mg kg$^{-1}$) immediately after the task for *Fos* tagging. Freezing was automatically measured throughout the testing trial by the TimeFZ4 software (version 2021_7_29, Ohara) and subsequently processed by a blinded investigator. The chambers were cleaned with 70% ethanol between tests. In some cases, 10 mg kg$^{-1}$ Propranolol (MedChem-Express, HY-B0573) was administered by intraperitoneal injection to mice 0.5 h before, and 2.5 h and 5.5 h after 4-OHT-based *Fos* tagging in order to block β-adrenergic signalling throughout the tagging timeframe. In some experiments using DREADDs, mice were injected with CNO (1 mg kg$^{-1}$) 10 min before the task in CtxA. For experiments in Fig. 2n in which CNO was used for silencing neuronal activity or propranolol was used for antagonizing β-adrenoreceptor signalling to test their roles in astrocyte *Fos* induction during fear memory recall, both were administered with consideration of their pharmacokinetics. Propranolol was given both before and after the behavioural session to ensure effective β-receptor blockade throughout the several-hour *Fos*-tagging window (Extended Data Fig. 3), given its short half-life[78] (~1.5–2 h), whereas CNO was given once before the session owing to its prolonged effect[79] (>6 h). Freezing was measured for 3 min in the CtxB chamber, which consisted of a rhombic partition (30 cm × 60 cm × 15 cm, 10 lux and 1% acetic acid odour) with a flat floor (Ohara).

## Whole-brain imaging

Mice were deeply anaesthetized with isoflurane and transcardially perfused with PBS, followed by 4% PFA in PBS for fixation of brain tissues. The dissected brains were post-fixed overnight at 4 °C in 4% PFA in PBS and stored in PBS at 4 °C until embedding. To oxidize agarose, 2.25 g agarose (A6013, Sigma-Aldrich) was added to 100 ml of 10 mM sodium periodate (S1878, Sigma-Aldrich)/50 mM phosphate buffer and stirred gently for 2.5 h at room temperature in the dark. The oxidized agarose was filtered on a bottle-top vacuum filter (0.2 µm, 569-0020, Thermo Scientific) washed 3 times with phosphate buffer and resuspended in 50 ml of phosphate buffer to make 4.5% oxidized agarose solution. The oxidized agarose solution was boiled with a microwave oven to melt the agarose and kept at 60–65 °C. The brains were pat-dried and embedded in oxidized agarose solution with a plastic mould (18646A, Polysciences). For covalent crosslinking between oxidized agarose and the surface of the brain, borate/borohydride solution was prepared the day before use by adding 0.2 g sodium borohydride (452882, Sigma-Aldrich) in 100 ml of prewarmed (40 °C) borate buffer (50 mM borax, 221732, Sigma-Aldrich; 50 mM boric acid, B6768, Sigma-Aldrich; pH 9.0–9.5) in the chemical fume hood. The agarose block was removed from the mould, soaked in borate/borohydride solution to initiate crosslinking and incubated at 4 °C overnight. The block was washed and kept in phosphate buffer at 4 °C until imaging.

The agarose block was mounted on a glass slide using Loctite 401 glue immediately before imaging. STPT was performed with the TissueCyte 1400FC system (TissueVision) equipped with Coherent

Chameleon Ultra II laser (Coherent) or Spectra-Physics Insight X3+ laser (MKS Spectra-Physics) and a Nikon 16× water-immersion objective (MRP07220, NA: 0.8). The laser wavelength was tuned at 1,000 nm. Brains were coronally sectioned with a built-in vibrating microtome at 50-µm thickness with a frequency of 60 Hz and a speed of 0.5 mm s$^{-1}$. Four optical *z*-planes were taken at 50 µm, 62.5 µm, 75 µm and 87.5 µm below the cut surface of the tissue block. To scan an entire coronal plane, individual tile images were acquired (832 × 832 pixels per frame, resolution at 1.382 µm per pixel) by moving the stage between each tile. Following the full scanning of a coronal plane, a 50-µm section was cut away by vibrating microtome and imaging of a new cut surface repeated to acquire optical sections covering the entire brain from the caudal to the rostral side (~320 coronal sections). The raw image tiles were first trimmed and subjected to flat field correction, then stitched into 2D mosaic coronal section images using the Autostitcher software (TissueVision). The signals from the green channel (500–550 nm, mNG signal) and from the far-red channel (>600 nm, background) were simultaneously acquired, and the latter was used to subtract background from the green channel to enhance the signal to noise ratio. Each region of the whole brain was aligned and segmented according to the Mouse Brain Atlas (NeuroInfo, MBL Bioscience, BN-200). Owing to the high quality preservation of tissue architecture during imaging, we semi-automatically performed alignment to a standard mouse brain atlas, segmentation of 839 brain regions, and detection of single Fos-tagged mNG astrocytes (BAEs), defined by their territories that typically displayed diameters of approximately 40 to 60 µm with ~10-µm-wide cell bodies[68] (Fig. 1a and Supplementary Videos 1 and 2). Consistent with our intersectional genetic approach, we rarely observed neuron-like mNG-positive cells, with larger cell bodies (>~20-µm diameter) and elongated processes (>~50-µm). We only found these cells in regions nearby ventricles and the midline where well-characterized GFAP-positive neural progenitor cells reside[48,80], and they were excluded in subsequent analyses. As a result, using this brain-wide analysis approach we successfully quantified the density of BAEs in 677 brain regions that accounted for 80.7% of the whole brain.

## Correlational analysis of *Fos*+ cell counts

To observe correlation between *Fos*+ astrocytes and neurons, we used the same TRAP2 (Fos-iCre$^{ERT2}$) system[28] used by Roy et al.[5], but instead with a Cre-dependent mNG reporter expressed in *Fos*+ astrocytes, allowing comparison of *Fos*+ cell counts from anatomically matched regions between studies. The regions of interest were first narrowed to the 117 engram-signifiant regions defined by Roy et al. Then, to ensure analysis of bona fide astrocyte-tagged regions, areas with possible neuronal leakage (mNG+ neuron-like cells), particularly near ventricles were excluded (see above). In addition, high-level hierarchical structure areas (such as midbrain or thalamus) with extremely high total counts that skew correlation analysis were also excluded. In the end, this selection yielded 75 regions for subsequent correlational analyses. Changes in *Fos*+ astrocyte numbers (FC−NoFC and FR−NoFR) were calculated using the non-foot shocked control group as the baseline, consistent with Roy et al. Supplementary table 1 from Roy et al. provided region-by-region counts of *Fos*+ neurons in non-foot shocked controls, FC and FR groups, enabling us to extract the corresponding changes in *Fos*+ neuron numbers for each of the 75 regions. Pearson correlation analyses between these neuronal and astrocytic *Fos*+ cell count changes were performed.

## Whole-brain visualization

For image visualization in 3D, low-resolution image stacks were created across all four channels. These stacks were downsampled to 25 µm per pixel resolution in the *x* and *y* axes, while the inter-slice distance was maintained at 50 µm, resulting in a voxel resolution of 25 × 25 × 50 µm$^3$. To minimize high-frequency artifacts, median filtering and Gaussian

smoothing were applied to the coronal images prior to downscaling. The processed 3D image stacks were stored in the NIfTI format, and the world coordinates were adjusted to align roughly with the Allen Mouse Brain Common Coordinate Framework v.3 (CCFv3)[81]. For registration to the CCFv3, the Allen STP-1675 template (filename: P56_Atlas.nii.gz; available at https://scalablebrainatlas.incf.org/mouse/ABA_v3#downloads) was used as the reference image. This template, generated from STPT averaged over 1,675 specimens, was combined with corresponding atlas annotations (filename: P56_Annotation.nii.gz). Image alignment was performed using the Advanced Normalization Tools (ANTs)[82]. An affine registration was first applied at 3 scales (downscaling factors of 4, 2 and 1) with Gaussian smoothing of 2, 1 and 1 voxels and 250, 200 and 50 iterations, respectively. This was followed by deformable (SyN) registration at four scales (downscaling factors of 8, 4, 2 and 1) with Gaussian smoothing of 4, 2, 1 and 1 voxels and 1,000, 500, 100 and 50 iterations, respectively, using mutual information as the metric. After computing the transformation, individual image stacks and atlas labels were mapped to and from the CCFv3 image space. After aligning with the NIfTI world coordinates, cell positions were mapped to the Allen template using the antsApplyTransformsToPoints function from ANTs. Brain regions from the Allen atlas annotations were then assigned to individual cells based on their locations in the CCFv3 image space.

## Primary astrocyte culture

Astrocyte primary cultures were obtained as previously described with some modifications[83]. In brief, cerebral cortical tissues were isolated from P0–P1 neonatal mice and dissociated with a scalpel, gently agitated in 0.05% Trypsin-EDTA (Gibco, 25300-054), and rapidly strained with a Falcon 100 μm cell strainer (352360). Dissociated cells were seeded at a rate of four brains per bottle on poly-D-lysine-coated bottles. The cells were cultured in DMEM (Gibco, 11995073) supplemented with FBS (Cosmo Bio, CCP-FBS-BR-500) and penicillin-streptomycin. When the cells reached confluence after several weeks, the culture bottles were gently shaken at 120 rpm overnight at 37 °C to remove non-astrocytic cells. Astrocytes were then detached using Trypsin-EDTA, replated onto poly-D-lysine-coated bottles, and after 1–2 weeks, cultures were shaken again at 120 rpm for 4 h. Astrocytes were then re-seeded on coverslips in 12-well plates at a density of $1.0 \times 10^5$ cells per well, which were maintained in 5% $CO_2$ at 37 °C, with the medium being changed every 2 to 3 days.

## Primary astrocyte c-Fos analysis

Five days after re-seeding, primary astrocytes were incubated with tetrodotoxin (TTX at 1 μM, Fujifilm, 207-15901) to eliminate the potential effect of neuronal firing, for at least 5 min to allow adequate equilibration, together with the following inhibitors, blockers or chelators indicated for each experiment unless otherwise described: prazosin hydrochloride (MedChemExpress, HY-B0193A), atipamezole hydrochloride (Tokyo Chemical Industry, A2956), propranolol hydrochloride (MedChemExpress, HY-B0573), BAPTA-AM (SIGMA, A1076), and SQ 22536 (Tokyo chemical industry, Q0105). The cultures were then incubated with the following growth factors, neurotransmitters, neuromodulators or neuropeptides in 5% $CO_2$ at 37 °C for 1.5 h: L-glutamic acid monosodium salt hydrate (Sigma-Aldrich, G5889), GABA (Sigma-Aldrich, A2129), ATP disodium salt hydrate from yeast (Nacalai tesque, 01072-24), L-noradrenaline hydrochloride (Sigma-Aldrich, 74480), dopamine hydrochloride (Sigma-Aldrich, H8502), acetylcholine chloride (Sigma-Aldrich, A6625), serotonin hydrochloride (Alfa Aesar, B21263.03), brain derived neurotrophic factor (BDNF, Fujifilm, 028-16451), neurotrophin-3 (Fujifilm, 146-09231, 50 ng ml⁻¹), neurotrophin-4 (Fujifilm, 142-06634, 100 ng ml⁻¹), forskolin (Fujifilm, 067-02191, 10 μM), nerve growth factor-β (NGFβ, Fujifilm, 141-07601, 100 ng ml⁻¹), epidermal growth factor (EGF, Fujifilm, 059-07873, 100 ng ml⁻¹), PDGF-AA (Fujifilm, 163–19731, 100 ng ml⁻¹),

PDGF-BB (Fujifilm, 164–24031, 100 ng ml⁻¹), Ciliary Neurotrophic Factor (CNTF, Fujifilm, 032-23501, 100 ng ml⁻¹), IGF1 (Fujifilm, 099-04511, 100 ng ml⁻¹). The following agonist and ionophore were used: isoproterenol (MedChemExpress, HY-B0468) and ionomycin (LKT laboratories, I5752). Ninety minutes after incubation, primary astrocytes were fixed in 4% PFA (Nacalai Tesque) for 10 min, washed with PBS 3 times, permeabilized with 0.1% Triton X-100 (Nacalai Tesque) for 10 min and blocked with 1% NDS (Sigma-Aldrich, D9663) for 30 min. Primary astrocytes were then incubated overnight at 4 °C with the following primary antibodies: rabbit anti-c-Fos (1:1,000, Synaptic Systems, 226003), rat anti-GFAP (1:1,000, Thermo Fisher Scientific, 13-0300), rabbit anti-IBA1 (1:1,000, Fujifilm, 019-19741), chicken anti-TUJ1 (1:1,000 Novus Biologicals, NB100-1612), mouse anti-OLIG2 (1:100, Millipore, MABN50). The cultures were washed 3 times in PBS for 10 min each, then incubated with DAPI (1:10,000, Sigma-Aldrich, D9542) and the following secondary antibodies (1:1,000) in PBS containing 1% NDS for 1 h at room temperature: Alexa Fluor 488 donkey anti-rat (abcam, ab150153), Alexa Fluor 568 donkey anti-mouse (abcam, ab175700), Alexa Fluor 568 donkey anti-rabbit (Abcam, ab175692), Alexa Fluor 647 donkey anti-rabbit (abcam, ab150067), Alexa Fluor 647 donkey anti-chicken (Jackson ImmunoResearch, 703-605-155). Fluorescent images were taken using a 20× objective lens (UPlanXApo 20×, Olympus) with a confocal laser-scanning microscope (Olympus, FV3000 with Fluoview). Immunoreactivity quantification of c-Fos was performed using ImageJ. The c-Fos brightness of GFAP-positive astrocytes were measured by the intensity of ROIs that encompassed the cell nuclei. In some experiments involving a heterologous expression of mGluR3, we applied AAV2/5-GfaABC1D-Grm3 (800,000 multiplicity of infection) 5 h after re-seeding for expressing mGluR3 in primary astrocytes and incubated for 5 days before pharmacological experiments. Ninety minutes after stimulation with 0.1 μM NA and/or 100 μM glutamate in the presence of 1 μM TTX, primary astrocytes were fixed and stained as described above using primary antibodies: rabbit anti-mGluR3 (1:500, abcam, ab166608) and mouse anti-c-Fos (1:50, Santa Cruz Biotechnology, sc166940). It is noteworthy that the mouse anti-c-Fos antibody, different from the rabbit anti-c-Fos antibody used in other experiments, exhibited overall low immunoreactivity. However, this did not affect the quantification of c-Fos mobilization. Fluorescent imaging and quantification of c-Fos intensity were performed as described above.

## Fibre photometry

AAVs for expressing the noradrenaline biosensor GRAB_{NE2h} on the neuronal cell surface, or calcium indicator RCaMP3 in astrocytes with cAMP sensor cAMPinG1 in astrocytes were unilaterally injected into the basolateral amygdala. Optical fibres were implanted above the injection site two weeks post-injection to capture emission of fluorescence signals, with recordings conducted one week after fibre implantation. For fluorescence signal collection, mice were tethered to a patch cable, and a 465 nm LED light (Doric, 40–50 μW at the tip of patch cable) or a 560 nm LED light (Doric, 40–50 μW at the tip of patch cable) was used to excite the signal, with an isosbestic 405 nm LED light used to correct for movement artifacts. Mice tethered to the patch cable were habituated in a testing chamber for 5 min on the day of recording. A time-division multiplexing scheme was used, where the 465 nm, 560 nm and 405 nm LED lights were emitted alternately at 20 Hz (turned on for 24 ms and off for 26 ms), with the timing precisely controlled by a programmable pulse generator (Master-9, AMPI). Each excitation light was reflected by a dichroic mirror and coupled into an optical fibre cable (Doric, 200 μm in diameter, 2 m in length, NA 0.57) through a pinhole (200 μm in diameter). The fluorescence signals were detected by a photomultiplier tube with a GaAsP photocathode (H16722-40; Hamamatsu Photonics) and digitized by a data acquisition module (USB-6341, National Instruments). A custom-written LabVIEW program extracted signals at a sampling frequency of 1 kHz in real time.

Fluorescence signals collected during each light pulse were averaged, after dropping the first and last 2 samples (1 ms each) to eliminate potential noise caused during the alternation of LED lights. The data were smoothed using moving average and corrected for bleaching correction by fitting to least-squares linear. The fluorescence response was calculated (fitted 465 + 1)/(fitted 405 + 1) − baseline. The baselines of GRAB$_{NE2h}$ and astrocyte cAMPinG1 signals were identified during periods of no spontaneous signals in home cages for 1–2 min before the context exposure. Signal analyses were performed in Python (v.3.0.0) using JupyterLab (v.3.6.7, Project Jupyter). Numerical computations and data handling were conducted using NumPy[84] (v.1.26.4) and Pandas (v.2.1.4, NumFOCUS). The area under the curve of fluorescence traces was calculated with scikit-learn (sklearn, v.1.2.2, RRID:SCR_002577), and half-decay times of signal responses were extracted. Astrocytic RCaMP3 signals were bandpass filtered between 0.0003 and 1 Hz with SciPy[85] (v.1.11.4). Peaks of Ca$^{2+}$ signals were defined as data points exceeding 4× s.d. above the baseline mean, with baseline noise estimated from ≥20 s intervals lacking spontaneous activity in home cage recordings before context exposure. Signal traces were visualized with Matplotlib[86] (v.3.8.0).

## Acute brain slice preparation for astrocyte intracellular Ca$^{2+}$ imaging

Coronal amygdalar slices were prepared from adult mice with AAV injection[26,87]. In brief, animals were deeply anaesthetized with isoflurane and decapitated with sharp shears. The brains were placed and sliced in ice-cold modified artificial cerebrospinal fluid (ACSF) containing the following (in mM): 194 sucrose, 30 NaCl, 4.5 KCl, 1 MgCl$_2$, 26 NaHCO$_3$, 1.2 NaH$_2$PO$_4$ and 10 D-glucose, saturated with 95% O$_2$ and 5% CO$_2$. A vibratome (DSK NLS-MT) was used to cut 300-µm brain sections. The slices were allowed to equilibrate for 30 min at 32–34 °C in normal ACSF containing (in mM); 124 NaCl, 4.5 KCl, 2 CaCl$_2$, 1 MgCl$_2$, 26 NaHCO$_3$, 1.2 NaH$_2$PO$_4$ and 10 D-glucose continuously bubbled with 95% O$_2$ and 5% CO$_2$. Slices were then stored at 21–23 °C in the same buffer until use. All slices were used within 2–6 h of slicing. Cells for all the experiments were imaged with a 16× water-immersion objective lens (Nikon, N16XLWD-PF, MRP07220, NA: 0.8) with Nikon two-photon laser-scanning microscope (A1R MP) equipped with a 920 nm wavelength laser (Alcor, Spark Lasers). To image GCaMP6f signals, astrocytes located in the LA/B and typically ~20 to ~30 µm below the slice surface were selected for imaging. Images were acquired at 1 frame per second. Slices were maintained in ACSF (124 mM NaCl, 4.5 mM KCl, 1 mM MgCl$_2$, 1.2 mM NaH$_2$PO$_4$, 26 mM NaHCO$_3$, 10 mM D-glucose and 2.0 mM CaCl$_2$) through a perfusion system. Phenylephrine (PE, Tocris Bioscience 2838) was applied in the bath in the presence of 300 nM tetrodotoxin (Cayman Chemical 14964) were applied in the bath at least 5 min prior to recording to allow adequate equilibration. A constant flow of fresh buffer perfused the imaging chamber at all times. Spontaneous Ca$^{2+}$ activity was recorded for 3 min (0 µM PE), followed by a 2-minute bath application of 1 µM PE, a 5-min washout, a 2-min application of 3 µM PE, and a 5-min washout. Ca$^{2+}$ signals were processed in ImageJ (NIH) and presented as the relative change in fluorescence (ΔF/F).

## scRNA-seq and analysis

Mice were anaesthetized with isoflurane 90 min after the initiation of behavioural paradigms and transcardially perfused with PBS (30 ml for 2 min). The amygdala was freshly dissected out using a Brain Matrix (Ted Pella, 15067) and was placed into cold Hanks' balanced salt solution (HBSS) (14025092, Gibco) containing 10 mM HEPES (17514-15, Naclai Tesque), 0.54% glucose (16806-25, Naclai Tesque), 5.0 µg ml$^{-1}$ actinomycin D (A1410, Sigma-Aldrich), 10 µM triptolide (T3652, Sigma-Aldrich) and 27.1 µg ml$^{-1}$ anisomycin (A9789, Sigma-Aldrich), as previously described[88,89]. Then, the tissue block was incubated for 15 min at 37 °C in 3 ml enzyme solution (HBSS, 10 mM

HEPES, 0.54% glucose, 0.5 mM EDTA (15575020, Invitrogen), 1.0 mM L-cysteine (10309-41, Naclai Tesque), 5.0 µg ml$^{-1}$ actinomycin D, 10 µM triptolide and 27.1 µg ml$^{-1}$ anisomycin) containing 20 U papain per mouse (LS003127, Worthington Biochemical). The tissue was then homogenized by passing through a 18 G needle and was further incubated for 15 min at 37 °C. Following homogenization using a 20 G needle, the resulting cell suspension was centrifuged at 500g for 5 min at 4 °C. The homogenate was separated by gradient centrifugation with 30% Percoll (P1644, Sigma-Aldrich) in PBS at 500g for 25 min at 4 °C (no brake). The pellet containing astrocytes at the bottom of the tube was then collected and washed once with PBS containing 2% FBS and 10 mM EDTA before staining. Fc receptors were blocked with Fc block (2.4G2, BD Bioscience) for 10 min at 4 °C before incubation with primary antibodies. Cells were stained with antibodies directed against CD11b-BV786 (1:200, M1/70, BD Bioscience), CD45-APC-Cy7 (1:200, 30-F11, BioLegend), O1-eFluor660 (1:200, 50-6506-82, eBioscience) and ASCA2-PE-Cy7 (1:200, 130-116-246, Miltenyi Biotec) for 40 min at 4 °C. Additionally, cells were treated with hashtag antibodies to label source samples (TotalSeqB0301-B0310, B0312 and B0314, BioLegend). After washing, cells were sorted using a CytoFLEX SRT (Beckman Coulter). Data were acquired with CytExpert software (Beckman Coulter). Post-acquisition analysis was performed using FlowJo software, version 10.9.0. Single-cell libraries were generated and sequenced on the Illumina NovaSeq 6000 sequencer. Sequence reads were processed and aligned to the mouse genome (mm10, GENCODE vM23/Ensembl98; https://cf.10xgenomics.com/supp/cell-exp/refdata-gex-mm10-2020-A.tar.gz) using CellRanger 3.0. Initially, genes expressed in <3 cells and cells with <100 genes and mitochondrial reads <6% were excluded from the analysis. Pericytes were removed from amygdalar cells based on *Pdgfrb*. Hashtag oligos (HTOs) were demultiplexed using HTODemux and singlet cells were used for subsequent analysis. Principal component analysis was performed on the expression data matrix using the Seurat. Thirty principal components were used for generating *t*SNE plots. Transcriptomic clusters were identified from a total 1,903 amygdalar astrocytes with resolution set to 0.4. Differentially expressed gene candidates from a comparison of *Adrb1*-positive and *Adrb1*-negative astrocytes were estimated using Mann–Whitney–Wilcoxon test.

## RNAscope

Cryosections were prepared as described above and stored at −80 °C. RNAscope in situ hybridization was performed using RNAscope Multiplex Fluorescent Reagent Kit v.2 with TSA Vivid Dyes (ACDBio, 323270), according to the manufacturer's instructions. The sections were washed for 5 min with PBS, and then heated at 60 °C for 30 min before post-fix. After dehydration in an ethanol series, hydrogen peroxide was applied for 10 min at room temperature. The sections were washed with ddH$_2$O twice, and then incubated in Target Retrieval Reagents for 5 min at 90–100 °C. A wash with ddH$_2$O for 15 s was followed by a dehydration with 100% ethanol for 3 min and dried at room temperature. Then, the sections were incubated with Protease III for 30 min at 40 °C. The sections were washed with ddH$_2$O twice and then incubated with probes Mm-Fos (ACDBio, 316921), Mm-Adra1a (ACDBio, 408611), Mm-Adrb1-C2 (ACDBio, 449761-C2), Mm-Igfbp2 (ACDBio, 405951) and Mm-Slc1a3-C3 (ACDBio, 430781-C3) for 2 h at 40 °C. Thereafter, the sections were washed twice with wash buffer for 2 min between incubations at 40 °C. The sections were incubated in AMP1 for 30 min, AMP2 for 30 min and AMP3 for 15 min at 40 °C, then incubated in horseradish peroxidase (HRP) for 15 min, fluorophore for 30 min and HRP-blocker for 15 min at 40 °C for each channel. In the staining of mCherry, slices were washed in 0.1 M PBS 3 times for 10 min each, followed by immunohistochemistry that was performed as described above. Following primary and secondary antibodies used: rabbit anti-RFP (1:250, Rockland, 600-401-379), Alexa Fluor 568 donkey anti-rabbit (1:1,000, Abcam, ab175692). Finally, the sections

were incubated with DAPI at room temperature, removed DAPI, and mounted in Prolong Gold Antifade Reagent (Invitrogen, P36934). Images were obtained in the same way as immunohistochemistry described above except with a step size of 0.5 μm. Images were processed with ImageJ. Astrocyte somata were delineated using *Slc1a3* signals. The intensity of *Fos*, *Adra1a*, *Adrb1* and *Igfbp2* signals within the somata were measured. To account for differences in somata sizes, the signals were normalized using *Slc1a3*. Subsequently, the normalized signals were z-scored within each sample to evaluate gene expression across a heterogeneous cell population. This approach facilitated the robust identification of cells exhibiting high gene expression relative to the population mean. In Fig. 4e,f, adrenoreceptor-positive cells were identified based on scRNA-seq data, selecting the top 50% of cells ranked by *Adra1a* expression and the top 20% of cells ranked by *Adrb1* expression. Co-expression index of *Adra1a* and *Adrb1*, calculated by relative expression level of *Adra1a* to NoFC median multiplied by relative expression level of *Adrb1* for each cell. In Fig. 4j, the correlation analysis of *Fos* and *Adrb1*, expression levels were evaluated using z-scores of RNAscope signal intensity to assess gene expression across a heterogeneous population of cells, enabling robust identification of cells with high gene expression relative to the population mean. Specifically, we identified astrocytes with z-scores greater than 2 in RNAscope *Fos* signal intensity as *Fos*-high astrocytes, capturing 3% of the astrocyte population with significantly elevated *Fos* expression, consistent with scRNA-seq data in Fig. 4c where *Fos*-positive astrocytes occupied 4% of the total astrocytes analysed. Similarly, *Adrb1*-high astrocytes were identified using z-scores greater than 0.5, capturing 26% of astrocytes, also in line with scRNA-seq data in Fig. 4c where *Adrb1*-positive astrocytes occupied 20% of the total astrocytes analysed. In addition, *Igfbp2*-high astrocytes were identified using z-scores greater than 0.5, capturing 24% of all astrocytes in Extended Data Fig. 12d,e, which is consistent with scRNA-seq data indicating that *Igfbp2*-positive astrocytes occupy 24% of the total astrocytes analysed.

## Statistical analysis

Most statistical tests were run in Prism 10.1.2 (GraphPad). The specific tests used are indicated in the figure legends and Supplementary Table 1. All tests were two-sided unless otherwise noted. Astrocyte Fos count analyses across groups for each brain region were run with R v.4.3.2 on RStudio v.2023.12.1.402 and relied on the tidyverse, car, dunn.test and dplyr packages and base R functions aov(), TukeyHSD() and p.adjust(). The data were first analysed with Levene's test for homogeneity of variance. Brain regions that passed Levene's test ($P > 0.05$) were followed by a one-way ANOVA test with Tukey's honest significant difference test and regions that failed Levene's test ($P < 0.05$; no homogeneity of variances) were followed by the Kruskal–Wallis test with Dunn's post hoc test and Bonferroni adjustment. False discovery rates were calculated across groups and brain regions from post hoc $P$ values using the Benjamini–Hochberg method.

Mice were randomly assigned to experimental groups, and mice assigned to different experimental conditions were run in parallel. Mouse experiments and analyses were done blinded to group allocation. Tissue samples that yielded insufficient cell counts (less than 30 cells) during scRNA-seq quality control were excluded from the study. The results of statistical comparisons, $n$ numbers and $P$ values are shown in the figure panels or figure legends along with averages. $n$ is defined as the numbers of cells, sections or mice throughout on a case-by-case basis; the unit of analysis is stated in the text or in each figure legend. In the figures, summarized data are shown as mean ± s.e.m. with a scatter plot and $P$ values are indicated by asterisks: *$P < 0.05$; **$P < 0.01$; ***$P < 0.001$; ****$P < 0.0001$.

## Reporting summary

Further information on research design is available in the Nature Portfolio Reporting Summary linked to this article.

## Data availability

All the single-cell transcriptomics data are available at Gene Expression Omnibus with accession identifier GSE272414. All raw replicate data used to generate the figures and the associated statistical tests are provided in Supplementary Table 1. *Fos*-mNG astrocyte counts in 677 brain regions in NoFC, FC, NoFR and FR groups are provided in Supplementary Table 2. All raw data supporting the findings of this study are available on the RIKEN CBS data sharing platform (https://neurodata.riken.jp/id/20250831-001; https://doi.org/10.60178/cbs.20250831-001).

## Code availability

Custom code for 3D rendering of the brain-wide *Fos*-mNG astrocyte map is available on GitHub (https://github.com/BrainImageAnalysis/Astrocyte3DMapping/). Custom code for statistical analysis brain-wide *Fos*-mNG astrocyte counts, fibre photometry data analysis and single-cell transcriptomic analysis is available on GitHub (https://github.com/Jun-Nagai-Lab/Dewa-et-al).

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

**Acknowledgements** We thank N. Fujimori, R. Ando, N. Miyasaka and N. Mataga for technical assistance; M. Motoyoshi for assisting with imaging; Y. Li for sharing GRAB sensors; and T. J. McHugh, J. P. Johansen, T. Kitamura, K. Tanaka, T. Kitazawa, T. Kanaya, B. Wulaer, E. Shigetomi, S. Okabe, B. Deneen and B. S. Khakh for valuable comments and discussions. STPT, cryosectioning and image acquisition were performed with common use equipment at the Support Unit for Bio-Material Analysis, RRD of RIKEN CBS. We thank the Division of Gene Expression Dynamics, the MEXT Cooperative Research Project Program, the Medical Research Center Initiative for High Depth Omics, and acknowledge support from CURE:JPMXP1323015486 for the Medical Institute of Bioregulation, Kyushu University for sequencing. Collaborations between the J.N. and T.M. groups are supported by Japan Agency for Medical Research and Development (AMED) grant JP24gm1910004. This work was supported in part by JSPS KAKENHI grant numbers 24K02128 (to J.N.), 24H02180 (to J.N.), 24KJ2074 (to K.K.), 23H04179 (to J.N.), JP22K15625 (to K.D.), JP21H00220 (to J.N.), JP21H02588 (to J.N.), JP21H05640 (to J.N.), JP21K15207 (to T.Y.), JP24H00861 (to M.S.), JP23H02782 (to M.S.), JP21H02752 (to T.M.), JP22H05062 (to T.M.), JP21H05044 (to M.I.), JP24K02264 (to M.I.), JSPS KAKENHI 24K02441 (E.T.), 25K19410 (G.N.), 24K19053 (S.N.) by Japan Science and Technology Agency (JST) under grant numbers JPMJFR2249 (to J.N.), JPMJAX211K (to T.Y.), JPMJPR1906 (to M.S.), by AMED under grant numbers JP21km0908001 (to J.N.), JP22dk0207063 (to J.N.), JP23jf0126004 (to J.N. and T.M.), JP21wm0425001 (to T.M.), JP23gm6510022 (to M.S.), JP21zf0127003h (to M.I.), JP22wm0425011 (to M.I.), Multidisciplinary Frontier Brain and Neuroscience Discoveries (Brain/MINDS 2.0) by AMED JP23wm0625001 (to H.S.), by RIKEN-Osaka University Science and Technology Hub Collaborative Research Program (to J.N.), and by the Takeda Science Foundation (to J.N.), the Mitsubishi Foundation (to T.M.), Daiichi Sankyo Foundation of Life Science (to T.M.), Mochida Memorial Foundation for Medical and Pharmaceutical Research (to T.M.), Astellas Foundation for Research on Metabolic Disorders (to T.M.), Ono Pharmaceutical Foundation for Oncology (to T.M.), Immunology and Neurology (to T.M.), the Nakajima Foundation (to J.N.), the Chemo-Sero-Therapeutic Research

Institute (to M.I.) and the Uehara Memorial Foundation (to T.M.). K.K. and H.K. were partly supported by JSPS Research Fellowships (24KJ2074 and 22KJ3141, respectively).

**Author contributions** K.D. carried out vector constructions with help from A. Komori, performed mouse surgeries and behaviour tests, and with M.A.H. and J.N., analysed brain-wide imaging data. A. Kasai provided guidance on the analysis of brain-wide data. H.K. performed ex vivo astrocyte calcium imaging and analysis. H.S. performed 3D rendering and video creation. A. Kuwahara performed primary astrocyte experiments with help from K.D., who established the protocol. K.K. conducted in vivo fibre photometry and animal behaviour experiments, including associated surgeries with guidance from N.T. on coding for data analysis. T.Y. and M.S. provided guidance on astrocyte sensors. M.T., G.N., S.N. and E.T. provided key mouse lines and contributed intellectually to related analyses. Tissue preparation for single-cell transcriptomics was performed by K.D., K.K., H.K. and M.A.H., followed by cell sorting by A.Y. and analysis of sequencing data by K.K. with guidance from M.I. and T.M., who provided equipment. K.D., A. Komori and N.A. performed RNAscope analyses. K.D. and K.K. performed in vivo infusion of antibodies, for which K.K. performed mouse surgeries and histological analyses. J.N. conceptualized and directed the project, guided analyses and wrote the paper with help from K.D., K.K., A.K. and H.S. All authors commented on and approved the manuscript.

**Competing interests** The authors declare no competing interests.

**Additional information**
**Correspondence and requests for materials** should be addressed to Jun Nagai.

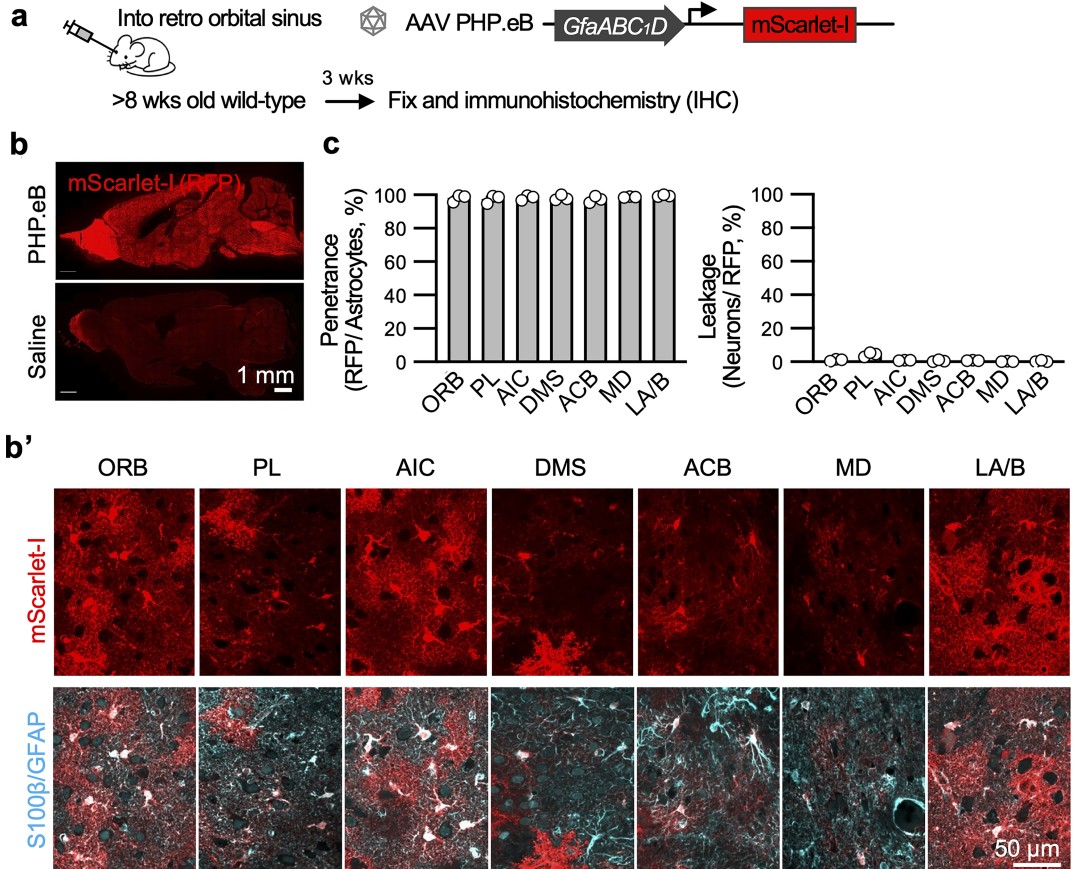

**Extended Data Fig. 1 | Validation of astrocyte *Fos* tagging approach: brain-wide astrocyte selectivity. a**, Schematic illustrating an AAV PHP.eB vector intravenous (i.v.) injection into retro orbital sinus of adult mice for expressing red fluorescence reporter mScarlet-I in astrocytes in whole brain. **b**, Representative images showing that whole-brain expression of mScarlet-I was observed in AAV injected mice, but not in saline injected mice. AAV PHP. eB-mediated mScarlet-I delivery resulted in expression within S100β/GFAP positive astrocytes. Subpanel b′ shows mScarlet-I expression in the orbitofrontal cortex (ORB), the prelimbic cortex (PL), the anterior insular cortex (AIC), the dorsomedial striatum (DMS), the nucleus accumbens (ACB), the mediodorsal thalamus (MD), and the lateral and basal amygdala (LA/B). Representative image from 3 mice. **c**, Bar plots showing that 97.3%–99.5% of astrocytes were positive for mScarlet-I, whose signals were enhanced with anti-RFP antibody, in the indicated 7 brain regions (left) and that those regions had few mScarlet-I-positive neurons (right). n = 3 mice. Data are mean ± SEM (error bars smaller than symbols in some cases). Sample sizes and replicates in Supplementary Table 1.

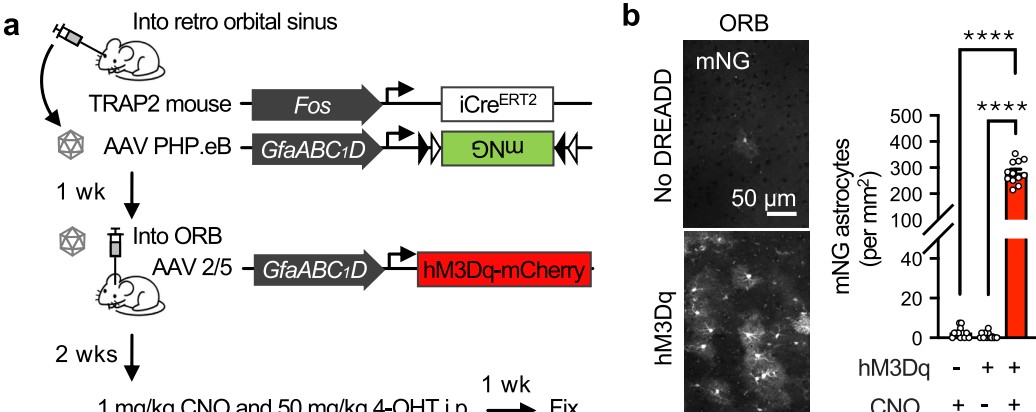

**Extended Data Fig. 2 | Validation of astrocyte *Fos* tagging approach: *Fos* upregulation sensitivity. a**, Schematic illustrating AAV-PHP.eB vector i.v. injection into *Fos*-iCre[ERT2] (TRAP2) mice for expressing green fluorescence reporter mNeonGreen (mNG) in *Fos*+ astrocytes. AAV2/5-GfaABC1D-hM3Dq-mCherry was stereotaxically delivered into the orbitofrontal cortex (ORB) to induce *Fos* in astrocytes by 1 mg/kg CNO i.p. injection and tag those *Fos*+ astrocytes with mNG by 50 mg/kg 4-hydroxytamoxifen (4-OHT) i.p. injection. **b**, hM3Dq activation with CNO administration increased *Fos* expression in ORB astrocytes. hM3Dq-, CNO+: n = 12 slices, hM3Dq+, CNO-: n = 12 slices, hM3Dq+, CNO+: n = 12 slices, hM3Dq-, 3 mice/group. ****p < 0.0001. One-way ANOVA, Tukey's test. Data are mean ± SEM. Sample sizes and replicates in Supplementary Table 1.

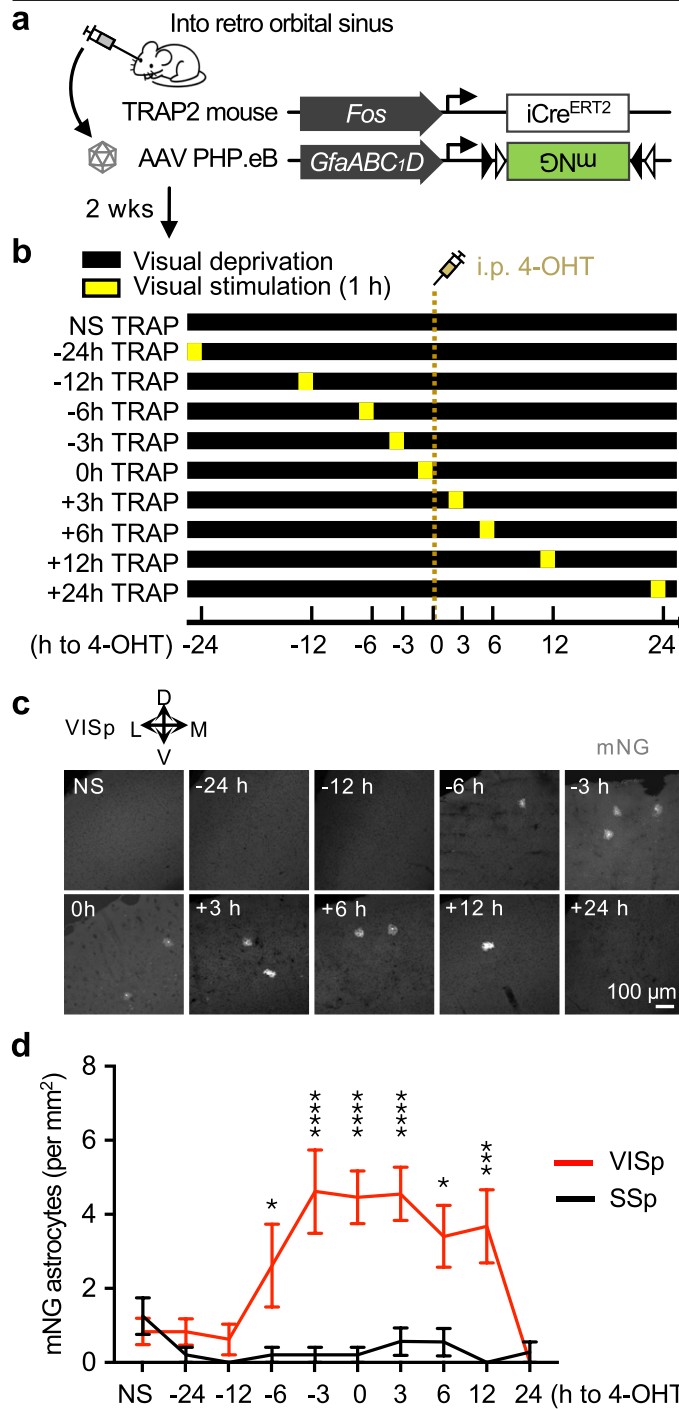

**a**, Into retro orbital sinus

TRAP2 mouse — Fos — iCre^ERT2

AAV PHP.eB — GfaABC1D — mNG

2 wks

**b**
- Visual deprivation
- Visual stimulation (1 h)

i.p. 4-OHT

NS TRAP
-24h TRAP
-12h TRAP
-6h TRAP
-3h TRAP
0h TRAP
+3h TRAP
+6h TRAP
+12h TRAP
+24h TRAP

(h to 4-OHT) -24  -12  -6  -3  0  3  6  12  24

**c**
D / L–M / V

VISp

mNG

NS | -24 h | -12 h | -6 h | -3 h
0h | +3 h | +6 h | +12 h | +24 h

100 µm

**d**

mNG astrocytes (per mm²)

VISp
SSp

NS -24 -12 -6 -3 0 3 6 12 24 (h to 4-OHT)

**Extended Data Fig. 3 | Validation of astrocyte *Fos* tagging approach: Temporal window of astrocyte *Fos* tagging with visual stimulation.**
**a**, Schematic illustrating AAV-PHP.eB vector i.v. injection into *Fos*-iCre^ERT2 (TRAP2) mice for expressing green fluorescence reporter mNeonGreen (mNG) in *Fos*+ astrocytes. **b**, Timeline of visual stimulation experiment to determine effective time window of astrocyte *Fos* tagging. **c**, Example images of *Fos*-tagged (mNG-positive) astrocytes in the primary visual cortex (VISp) of TRAP2 mice that underwent the visual stimulation experiment (representative of 9–12 slices from 3-4 mice/group). **d**, Quantification of mNG astrocyte density in TRAP2 mice VISp and primary somatosensory cortex (SSp): NS: n = 12 slices from 4 mice, −24 h: n = 12 slices from 4 mice, −12 h: n = 12 slices from 3 mice, −6 h: n = 12 slices from 4 mice, −3 h: n = 12 slides from 4 mice, 0 h: n = 12 slices from 4 mice, 3 h: n = 9 slices from 3 mice, 6 h: n = 9 slices from 3 mice, 12 h: n = 9 slices from 3 mice, 24 h: n = 9 slices from 3 mice. Two-way ANOVA, Šídák's multiple comparisons test. Data are mean ± SEM. *$p < 0.05$, ***$p < 0.001$, ****$p < 0.0001$. Sample sizes and replicates in Supplementary Table 1.

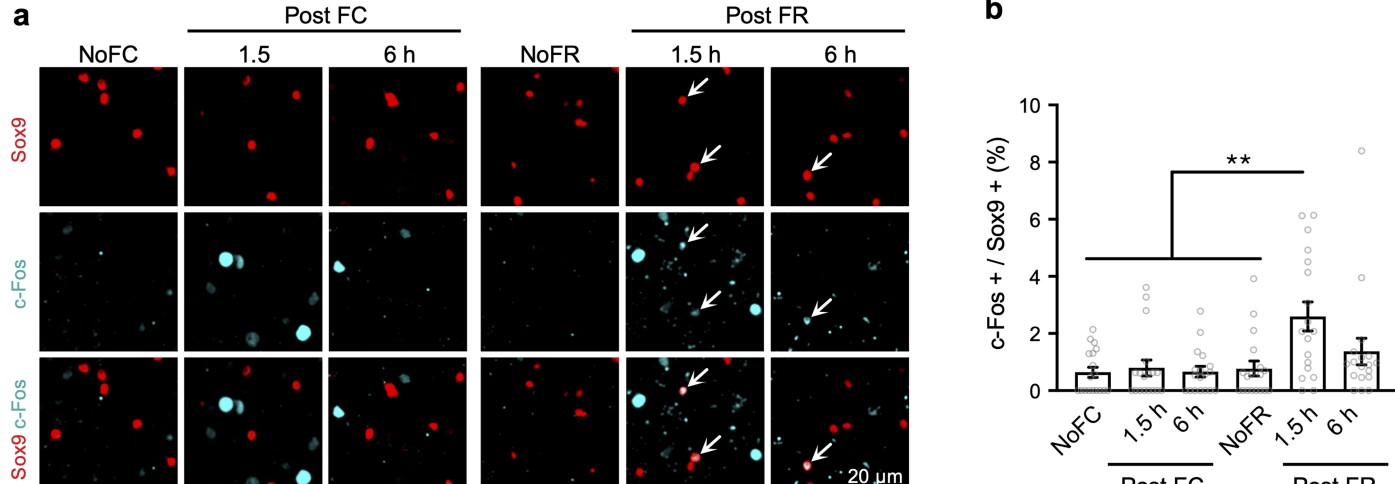

**Extended Data Fig. 4 | Astrocyte c-Fos protein levels following fear conditioning and recall. a**, Representative immunostaining of c-Fos expression in a subset of amygdalar astrocytes after fear conditioning (FC) and fear recall (FR). Arrowheads denote c-Fos+ Sox9+ cells (representative of 15 slices from 3 mice/group). **b**, Quantification of c-Fos+ astrocytes in the lateral and basolateral amygdala (LA/B) after FC and FR. n = 15 slices from 3 mice/group. One-way ANOVA, Tukey's test. Data are mean ± SEM. **p < 0.01. Sample sizes and replicates in Supplementary Table 1.

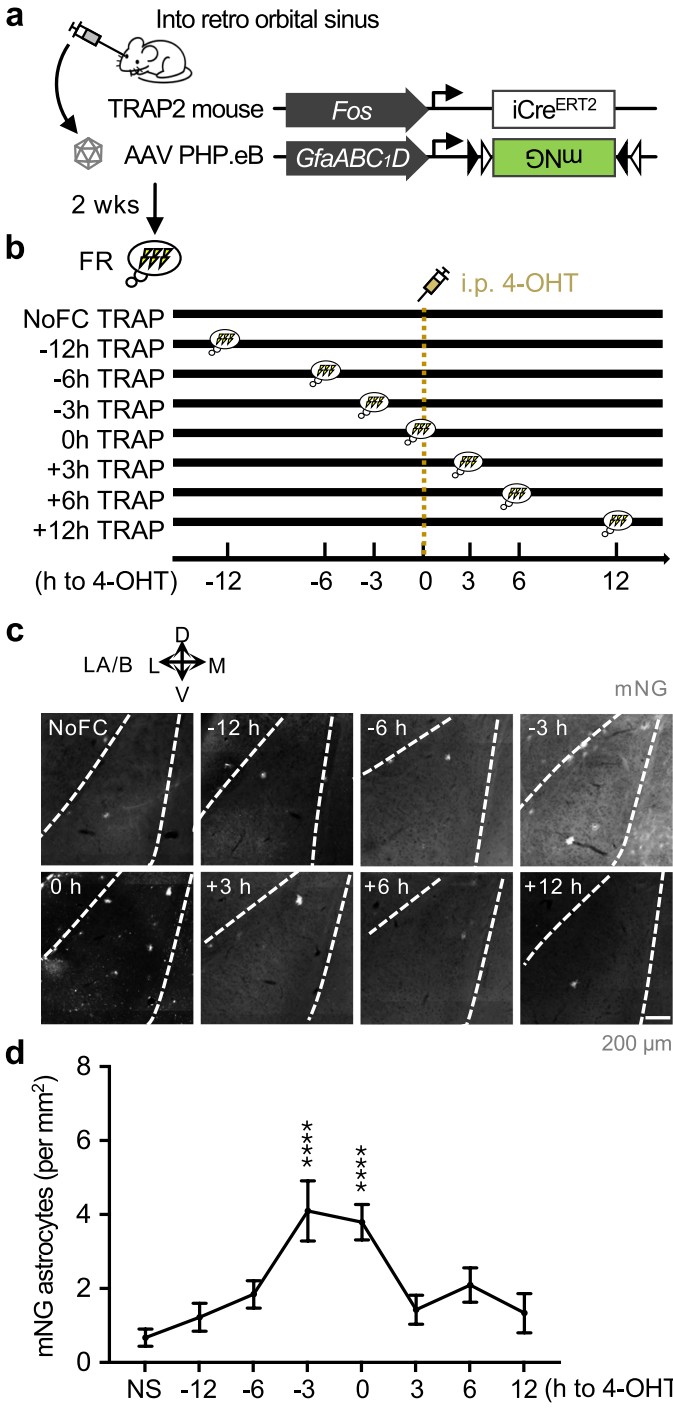

**a** Into retro orbital sinus

TRAP2 mouse — Fos → iCre^ERT2

AAV PHP.eB — GfaABC₁D → mNG

2 wks

**b** FR

**c** LA/B

mNG

NoFC | -12 h | -6 h | -3 h

0 h | +3 h | +6 h | +12 h

200 μm

**d**

**Extended Data Fig. 5 | Validation of astrocyte *Fos* tagging approach: Temporal window of astrocyte *Fos* tagging with fear recall. a**, Schematic illustrating AAV-PHP.eB vector i.v. injection into *Fos*-iCre^ERT2 (TRAP2) mice for expressing green fluorescence reporter mNeonGreen (mNG) in *Fos*+ astrocytes. **b**, Timeline of fear recall experiment to determine effective time window of astrocyte *Fos* tagging. **c**, Example images of *Fos*-tagged (mNG-positive) astrocytes in the lateral and basolateral amygdala (LA/B) of TRAP2 mice that underwent the fear recall experiment (representative of 17–48 slices from 3–8 mice/group). **d**, NS: n = 24 slices from 4 mice, −12 h: n = 30 slices from 5 mice, −6 h: n = 42 slices from 7 mice, −3 h: n = 23 slides from 4 mice, 0 h: n = 48 slices from 8 mice, 3 h: n = 30 slices from 5 mice, 6 h: n = 18 slices from 3 mice, 12 h: n = 17 slices from 3 mice. Two-way ANOVA, Šídák's multiple comparisons test. Data are mean ± SEM. ****p < 0.0001. Sample sizes and replicates in Supplementary Table 1.

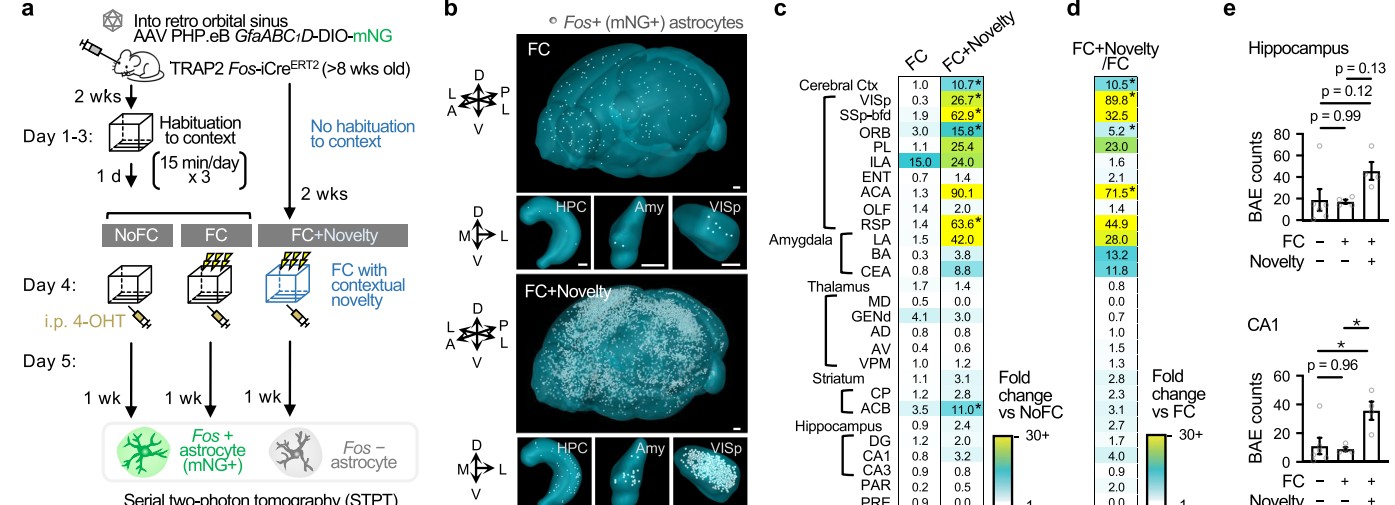

**Extended Data Fig. 6 | Fear conditioning with contextual novelty induces astrocyte *Fos* ensembles. a**, Left: Whole-brain mapping pipeline. Alongside NoFC and FC groups (Fig. 1a), an additional cohort received 4-OHT immediately after CtxA exposure with shocks but no habituation (FC+Novelty). Brains were fixed and processed for STPT, 3D reconstruction, regional segmentation, and single-cell counting as in Fig. 1a. **b**, 3D rendering of the whole brain and three brain regions - the hippocampus (HPC; CA1, CA2, CA3 and dentate gyrus), the amygdala (lateral, basolateral and basomedial amygdalar subregions), primary visual cortex (VISp) - showing that *Fos*-mNG astrocytes (white dots). D dorsal, V ventral, A anterior, P posterior, M medial, L lateral. Scale bars, 500 μm. **c-d**, Heatmap of BAE count fold changes (vs NoFC in **c**, vs FC in **d**, n = 4 mice/group). *FDR < 0.05. *FDR < 0.05; full stats in Supplementary Table 2. Levene's test followed by one-way ANOVA or Kruskal-Wallis; post-hoc Tukey or Dunn with Bonferroni. FDR controlled by Benjamini-Hochberg. Full data in Supplementary Table 2. Primary visual area (VISp), primary somatosensory area barrel field (SSp-bfd), orbital cortex (ORB), prelimbic cortex (PL), infralimbic cortex (ILA), entorhinal cortex (ENT), anterior cingulate cortex (ACA), olfactory cortex (OLF), retrosplenial cortex (RSP), lateral amygdala (LA), basal amygdala (BA), central amygdala (CEA), mediodorsal thalamus (MD), dorsal geniculate thalamus (GENd), anterodorsal thalamus (AD), anteroventral thalamus (AV), ventroposterior medial nucleus of thalamus (VPM), caudoputamen (CP), nucleus accumbens (ACB), dentate gyrus (DG), hippocampal CA1 (CA1), hippocampal CA3 (CA3), para-subiculum (PAR), and pre-subiculum (PRE). **e**, Hippocampus/ CA1 BAE counts higher in FC+Novelty. FC-, Novelty-: n = 6 mice, FC+, Novelty-: n = 4 mice, FC+, Novelty+: n = 4 mice. Data are mean ± SEM. *p < 0.05. Sample sizes and replicates in Supplementary Table 1.

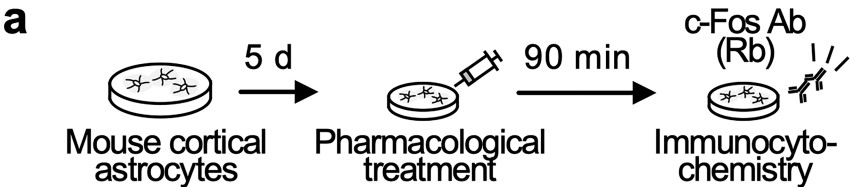

**a**

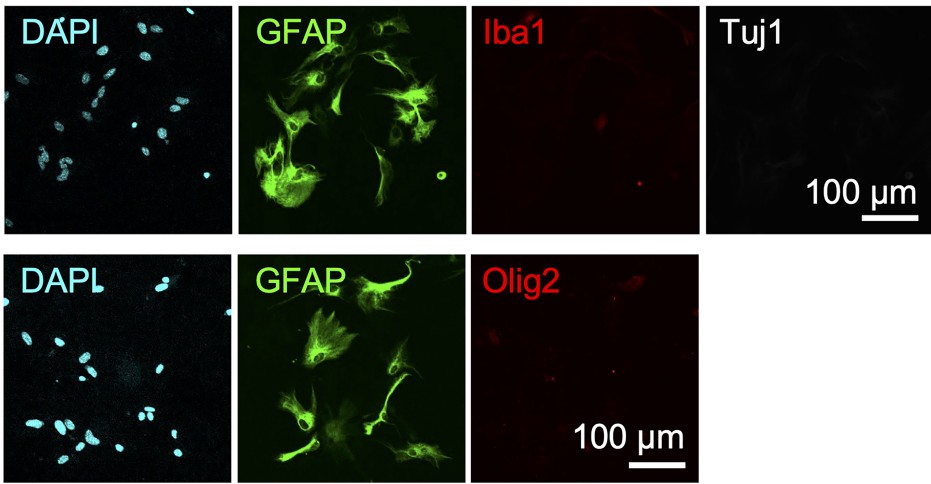

**b**

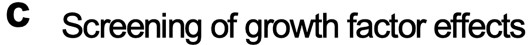

**c** Screening of growth factor effects

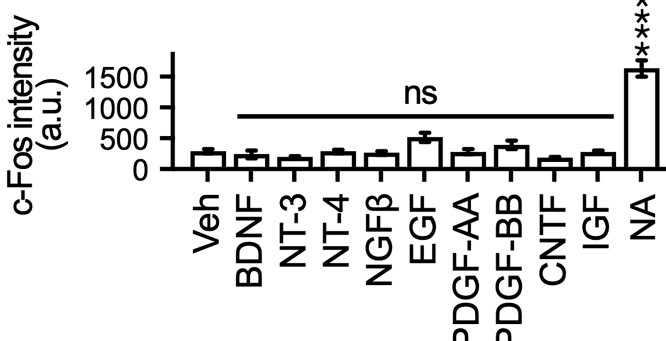

**Extended Data Fig. 7 | Growth factors known to induce neuronal c-Fos did not induce astrocyte c-Fos. a**, Schematics illustrating the design of pharmacological screening. **b**, Representative images from 9 biological replicates showing high purity of primary astrocyte culture. The immunoreactivities of Iba1, a microglia marker, Tuj1, a neuronal marker and Olig2, an oligodendrocyte marker, were not evident, and most of the DAPI signal overlapped with GFAP-positive primary astrocytes. **c**, Graph showing that c-Fos intensity in GFAP-positive astrocytes was increased with noradrenaline (NA), but not with growth factors that are known to induce c-Fos expression in primary neurons. TTX (1 μM) was used throughout the culture experiments to eliminate the potential effect of neuronal firing. Veh: n = 41 cells, NA: n = 45 cells, BDNF: n = 29 cells, NT-3: n = 33 cells, NT-4: n = 41 cells, NGFβ: n = 35 cells, EGF: n = 36 cells, PDGF-AA: n = 39 cells, PDGF-BB: n = 35 cells, CNTF: n = 40 cells, IGF: n = 46 cells. One-way ANOVA, Dunnett's test (vs Veh). Data are mean ± SEM. ****p < 0.0001; ns, not significantly different. Sample sizes and replicates in Supplementary Table 1.

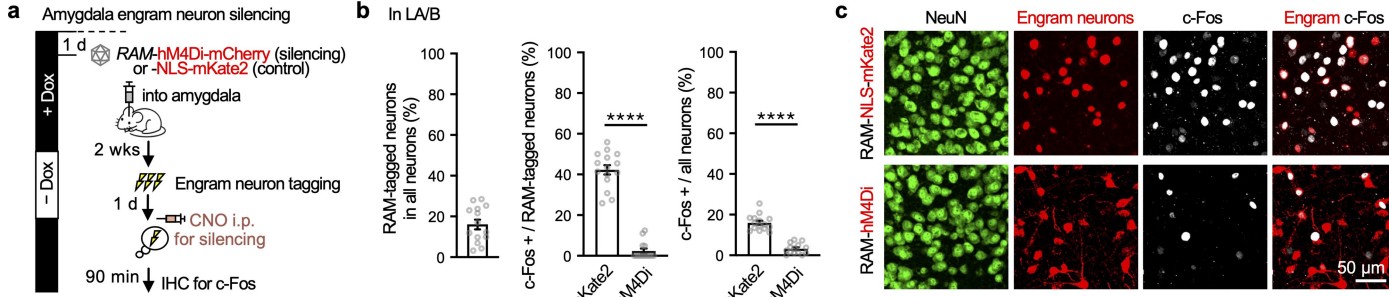

**Extended Data Fig. 8 | Validation of Robust Activity Marking (RAM) technique for silencing engram neurons. a**, Schematic showing the injection of RAM-hM4Di-mCherry or RAM-NLS-mKate2 into the amygdala. Mice underwent fear conditioning two weeks later followed by i.p. CNO (1 mg/kg) injections the day after. Brain samples were intracardially fixed for immunohistochemistry analysis. **b**, Graphs showing the percentage of RAM-expressed neurons in LA/B (left), co-expression of c-Fos and RAM (center), and overall c-Fos+ neurons (right). mKate2: n = 15 slices, hM4Di: n = 14 slices, 5 mice per group. Mann-Whitney test. ****p < 0.0001. **c**, Representative images (from mKate2: n = 15 slices, hM4Di: n = 14 slices, 5 mice/group) showing expression of neuronal marker NeuN and RAM-tagged engram neurons (indicated by mCherry signals) in **b**. Overlapping c-Fos-expressing engram neurons were fewer in the RAM-hM4Di group. Data are mean ± SEM. Sample sizes and replicates in Supplementary Table 1.

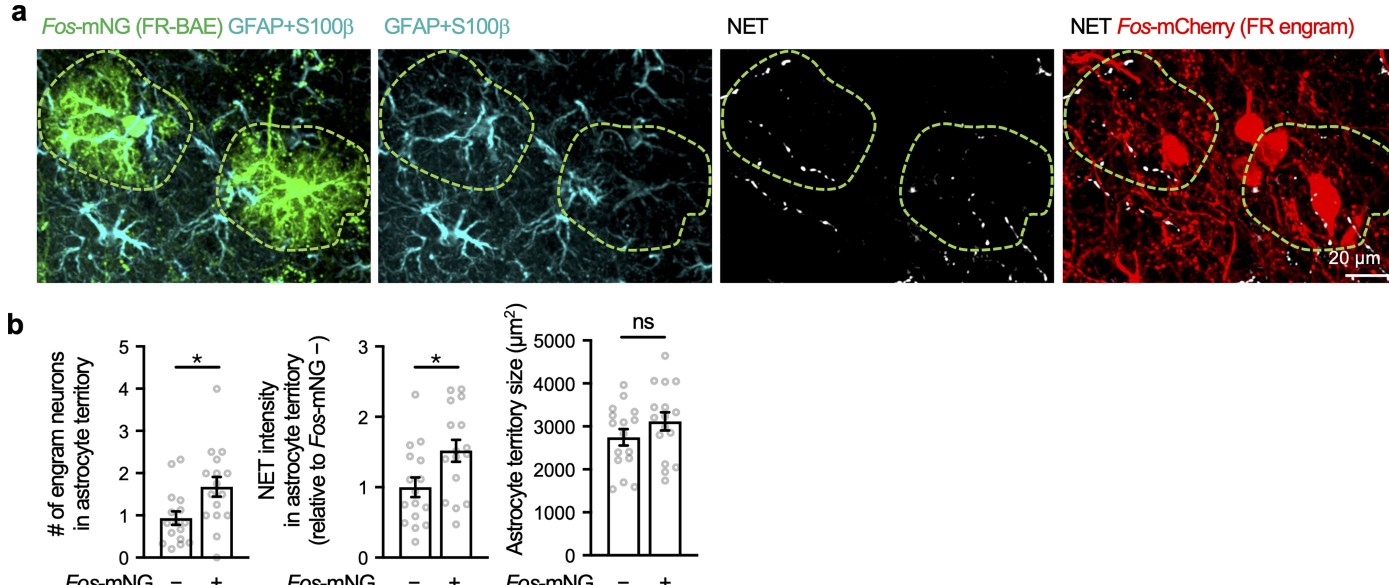

**Extended Data Fig. 9 | FR-BAE proximity to NA projections and engram neurons. a**, Images (representative of n = 16 slices from 4 mice) showing FR-BAE (green) in the amygdala overlapping with astrocyte markers GFAP and S100β (cyan) and proximal to NA projections (white) and engram neurons (red). **b**, Graphs showing that engram neurons (left) and NA projections (center) are statistically more colocalized with FR-BAE than non-FR-BAE astrocytes. Astrocyte territory size is not different between BAE and non-FR-BAE astrocytes (right). n = 16 slices from 4 mice/group. Mann-Whitney test. *p < 0.05; ns, not significantly different. Data are mean ± SEM. Sample sizes and replicates in Supplementary Table 1.

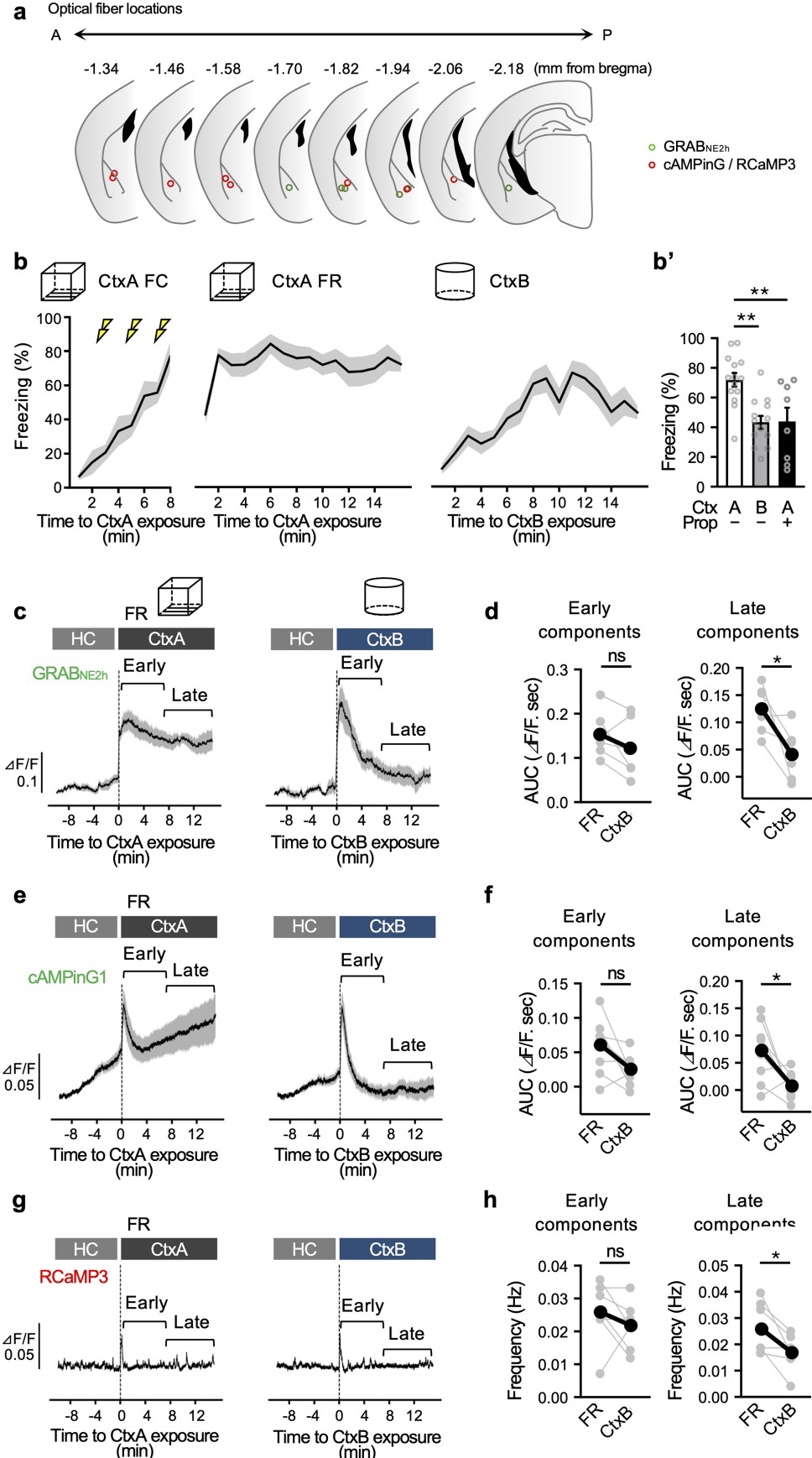

**Extended Data Fig. 10** | See next page for caption.

**Extended Data Fig. 10 | NA, astrocyte cAMP and Ca²⁺ signals in a context relevant or irrelevant to fear. a**, Histological analysis of optical fiber placement in LA/B for fiber photometry. Termination sites were mapped onto a standard atlas (adapted from Paxinos, G. & Franklin, K.B.J. *The Mouse Brain in Stereotaxic Coordinates, 2nd ed*. Academic Press, 2001). Circles show fiber tips from individual mice, color-coded by experimental group. n = 6 mice in GRAB$_{NE2h}$ and 8 mice in cAMPinG1/RCaMP3 experiments. **b**, Freezing before, during, after FC, during FR, and in CtxB (distinct from CtxA). Freezing was lower in CtxB and in CtxA with Prop vs FR (FR: n = 14; CtxB: n = 13; Prop: n = 8 mice). **c**, dF/F traces: NA signals evoked by CtxA or CtxB (early) persisted longer in CtxA (late). n = 6 mice. **d**, Scatter plots: integrated areas of early and late NA signals in CtxA vs CtxB. n = 6 mice. **e**, Astrocyte cAMP traces: gradual buildup in HC, strong early responses to context exposure. FR further enhanced late cAMP in CtxA, not CtxB. n = 8 mice. **f**, Scatter plots: late cAMP signal areas higher in CtxA vs CtxB, consistent with GRAB$_{NE2h}$. n = 8 mice. **g**, Astrocyte Ca²⁺ traces: evoked by context exposure (early). At 7–15 min, signal frequency remained elevated in CtxA, not CtxB (late). n = 7 mice. **h**, Scatter plots: late Ca²⁺ frequency higher in CtxA vs CtxB. n = 7 mice. Wilcoxon signed-rank test or paired t test was used based on normality of the data distribution. *p < 0.05, **p < 0.01, ***p < 0.001; ns, not significantly different. Data are mean ± SEM (error bars smaller than symbols in some cases). Sample sizes and replicates in Supplementary Table 1.

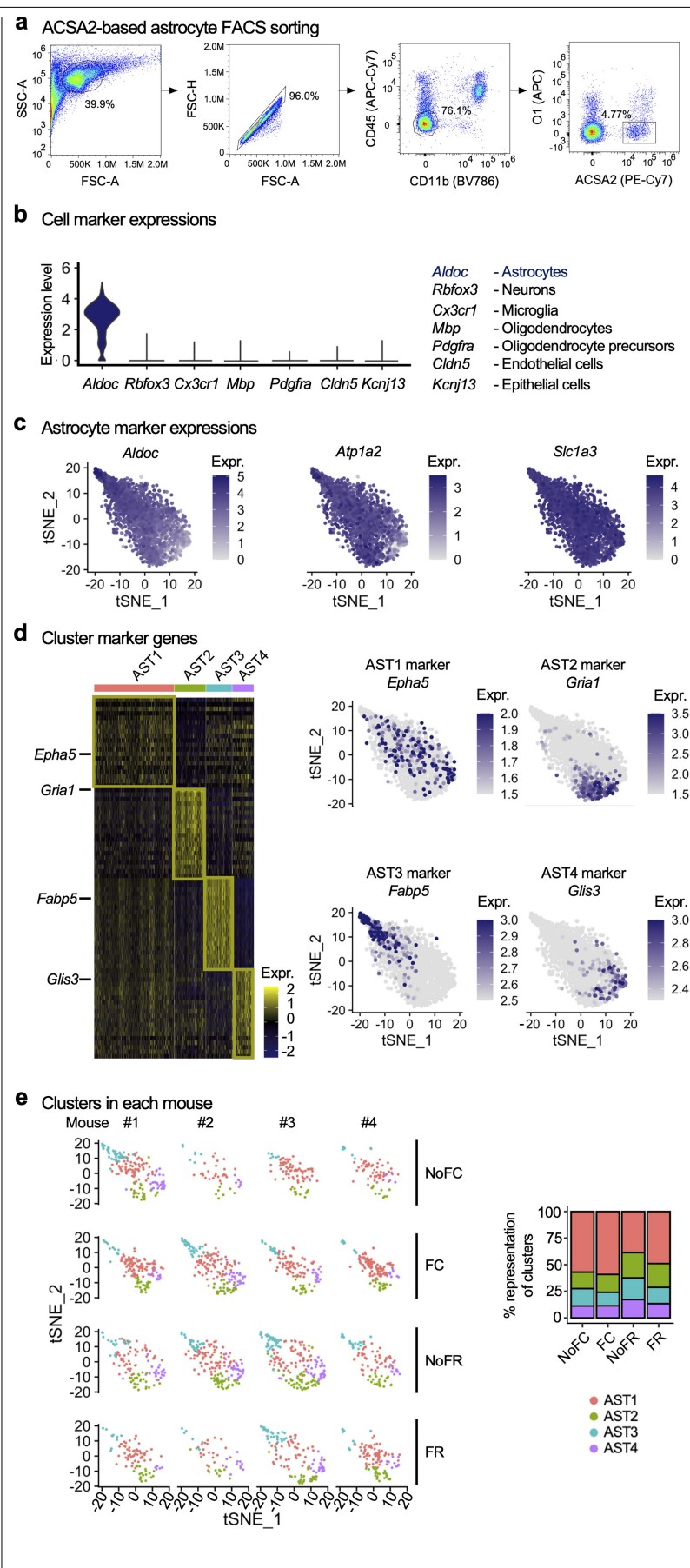

**a** ACSA2-based astrocyte FACS sorting

**b** Cell marker expressions

*Aldoc* - Astrocytes
*Rbfox3* - Neurons
*Cx3cr1* - Microglia
*Mbp* - Oligodendrocytes
*Pdgfra* - Oligodendrocyte precursors
*Cldn5* - Endothelial cells
*Kcnj13* - Epithelial cells

**c** Astrocyte marker expressions

**d** Cluster marker genes

**e** Clusters in each mouse

**Extended Data Fig. 11 | scRNA-seq of amygdalar astrocytes.**
**a**, Representative gating strategy of FACS-sorting of ACSA2+
astrocytes from the amygdala for scRNA-seq. **b**, Violin plot showing
relative expression levels of cell type marker genes in scRNA-seq.
*Aldoc*, an astrocyte marker, was the only gene showing high
expression. **c**, tSNE map showing astrocyte genes *Aldoc, Atp1a2*
(coding astrocyte cell surface antigen-2, ACSA2 with which we
aimed to FACS-sort astrocytes from the amygdala tissue), and
*Slc1a3* with which we identified high expression in astrocytes across
amygdalar astrocytes in our RNAscope experiments. **d**, Left: heatmap
showing the top 20 genes that were upregulated in each astrocyte
cluster relative to the others. Right: Example cluster marker genes
identified were plotted onto the tSNE map. See also Supplementary
Table 3. **e**, Left: tSNE plot of astrocyte clusters to illustrate that the 4
clusters are found in every single mouse examined. Right: Bar graph
showing the proportion of each astrocyte cluster in the four
experimental groups. n = 16 mice in **b-d**, 4 mice/group in **e**.

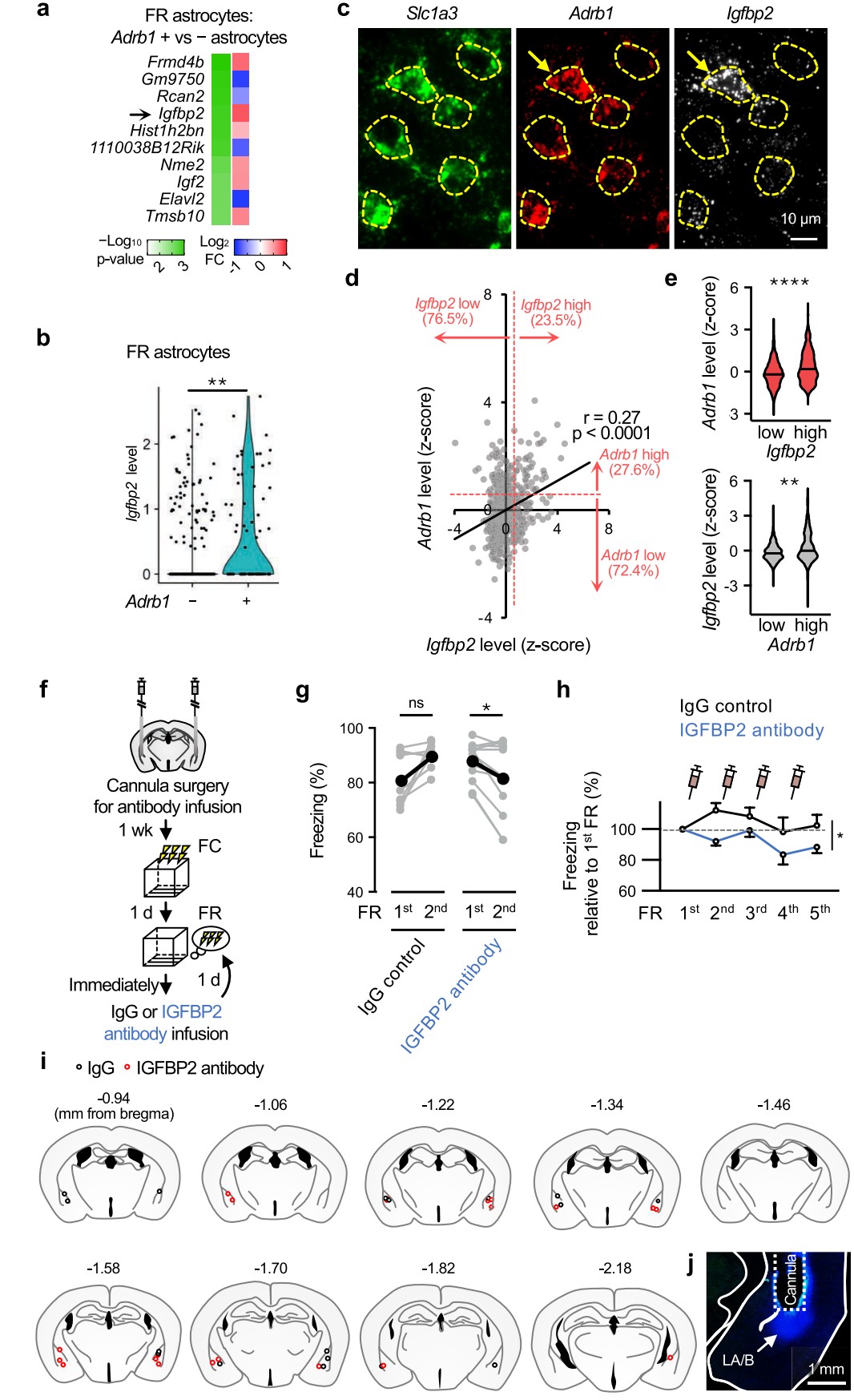

**Extended Data Fig. 12 |** See next page for caption.

**Extended Data Fig. 12 | Astrocytic Igfbp2 stabilizes memory during reconsolidation. a**, Top 10 differentially expressed gene candidates, ranked by p value comparing *Adrb1*-positive vs. -negative astrocytes. Heatmap shows decreased (blue) or increased (red) expression. **b**, Violin plot showing *Igfbp2* enrichment in *Adrb1*-positive astrocytes. n = 293 *Adrb1*−, 87 *Adrb1*+ astrocytes. Mann–Whitney test. **c**, RNAscope images (representative of 588 cells from 6 mice) of *Adrb1* and *Igfbp2* in *Slc1a3*-expressing astrocytes in LA/B from FR. Arrow: astrocyte with high *Adrb1* and *Igfbp2*. **d**, Correlation of *Igfbp2* and *Adrb1* levels. *Igfbp2*-high astrocytes (z > 0.5) = 24% of astrocytes. Rationale in Methods. **e**, *Adrb1* expression enriched in *Igfbp2*-high astrocytes (n = 451 low, 137 high cells). *Igfbp2* expression enriched in *Adrb1*-high astrocytes (n = 426 low, 162 high cells). Mann–Whitney test. Lines, median. **f**, Schematic of in vivo Igfbp2 perturbation. Mice implanted with bilateral cannulae into LA/B for Igfbp2-neutralizing antibody or control IgG infusion immediately after FR. **g**, Scatter graph: IgG infusion (n = 9 mice) showed no change in freezing between 1st and 2nd FR, whereas Igfbp2 antibody infusion (n = 12 mice) reduced freezing in 2nd FR, indicating destabilized memory. Wilcoxon signed-rank one-tail test. **h**, Freezing behaviour during 5-day Igfbp2 antibody infusions into LA/B within reconsolidation windows. n = 9 mice IgG, 12 mice Igfbp2 antibody. Two-way ANOVA with Šídák's test. **i**, Cannula placement in LA/B. Sites mapped on a standard atlas (adapted from Paxinos, G. & Franklin, K.B.J. *The Mouse Brain in Stereotaxic Coordinates, 2nd ed*. Academic Press, 2001). Each circle indicates cannula termination, coloured by group. n = 9 mice IgG, 12 mice Igfbp2 antibody. **j**, Representative LA/B image showing Alexa-labelled antibody infusion (representative of 36 sections from 21 mice). *p < 0.05, **p < 0.01, ****p < 0.0001; ns, not significant. Data are mean and SEM (error bars smaller than symbols in some cases). Full details in Supplementary Table 1.

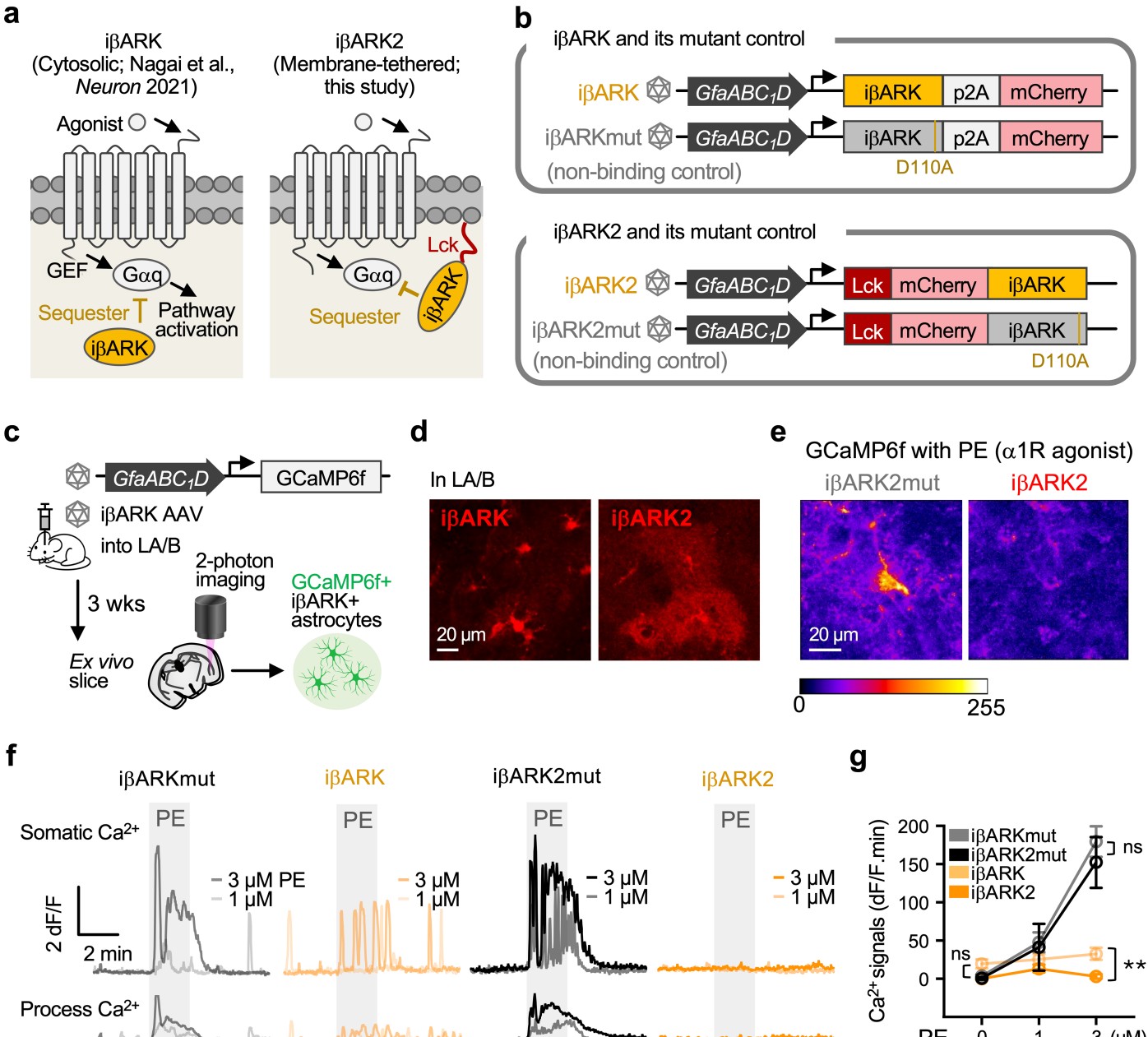

**Extended Data Fig. 13 | Astrocytic iβARK2 validation in the amygdala.**
**a**, Schematics showing the Gαq GPCR signalling cascade and the blocking of
which with iβARK. Left: iβARK, a 122-residue inhibitory peptide from β-adrenergic
receptor kinase[26], directly binds to Gαq subunits and thus sequesters them to
block subsequent Gαq-GPCR-induced signalling cascades. Right: iβARK2 with
Lck allows it to localize to the plasma membrane, thus being more proximal to
and binding Gαq subunits more efficiently. **b**, Top: AAV2/5 constructs for
astrocytic iβARK or iβARKmut. Bottom: AAV2/5 constructs for iβARK2 and
iβARK2mut. **c**, Schematic for Ca²⁺ imaging of iβARK-positive LA/B astrocytes.
**d**, Representative images showing expression of iβARK and iβARK2 in LA/B.
iβARK2 has improved expression near plasma membrane (iβARK: n = 5, iβARK2:

n = 5 biological replicates). **e**, Representative pseudo-coloured images showing
the high fluorescence of GCaMP6f in LA/B astrocytes (representative of n = 28
iβARK2mut and 13 iβARK2 astrocytes). **f**, dF/F traces of Ca²⁺ responses in a
iβARK(D110A)- or iβARK-expressing astrocyte evoked by PE (phenylephrine),
an agonist of Gαq-coupled α1A-adrenoceptor (coded by *Adra1a*) analyzed for
soma and processes separately. **g**, Scatter graph showing the average Ca²⁺
responses to 0, 1, and 3 μM PE, where iβARK significantly reduced the PE-induced
Ca²⁺ responses. iβARKmut: n = 29 cells, iβARK: n = 33 cells, iβARK2mut: n = 28
cells, iβARK2: n = 13 cells, 3-4 mice per group. Two-way ANOVA, Tukey's multiple
comparisons test. Data are mean ± SEM. **p < 0.01; ns, not significantly different.
Sample sizes and replicates in Supplementary Table 1.

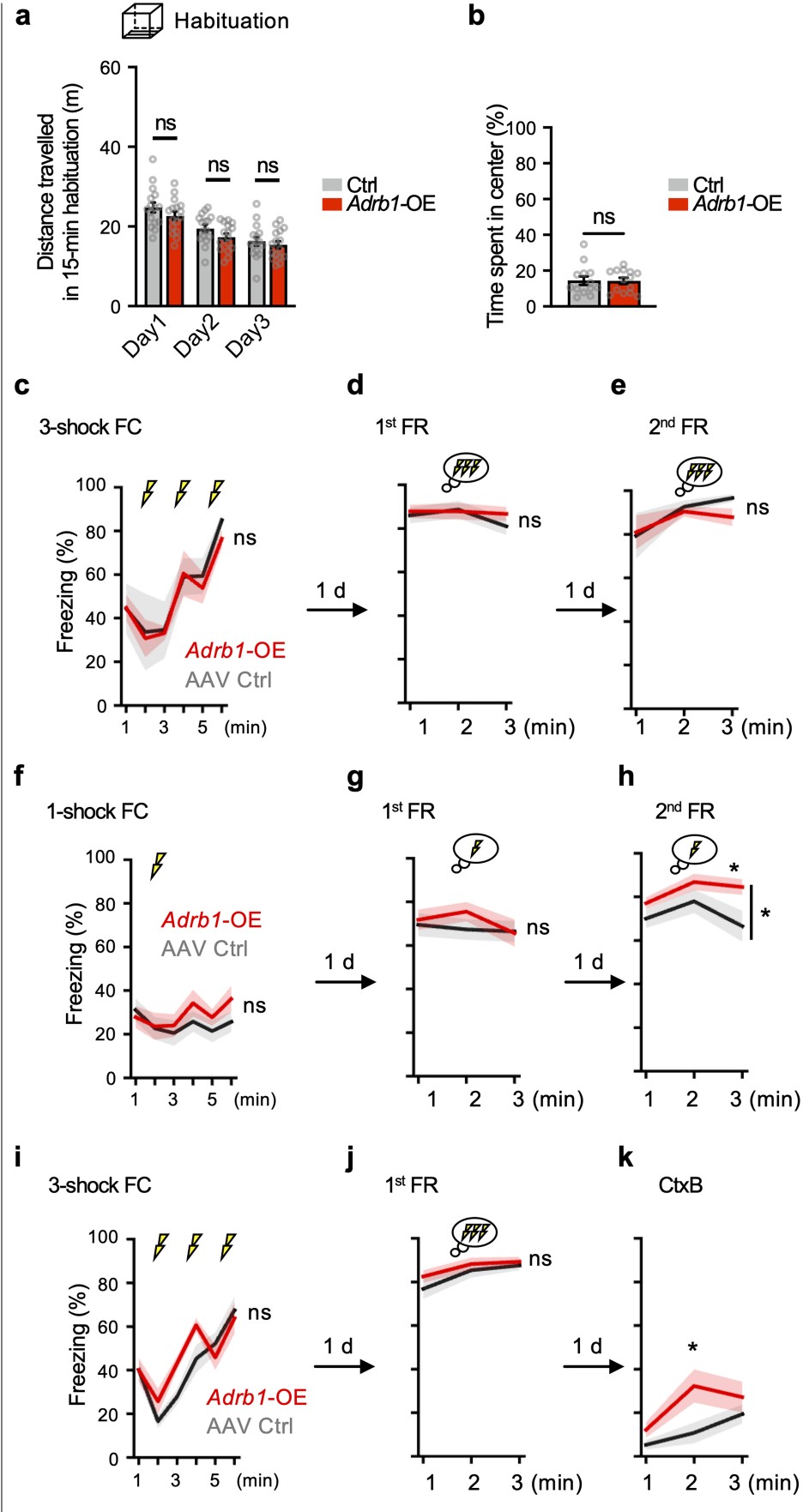

**Extended Data Fig. 14 | Additional assessments of the behavioural consequences of *Adrb1*-OE mice.**
**a**, Graphs showing no alteration in locomotion of *Adrb1*-OE mice during the open field test. Ctrl: n = 13 mice, *Adrb1*-OE: n = 14 mice. Two-way ANOVA and Šídák's multiple comparisons test were used. **b**, Graphs showing no alteration in time spent in center zone of *Adrb1*-OE mice during the open field test. Ctrl: n = 13 mice, Adrb1-OE: n = 14 mice. Mann-Whitney test. **c-k**, Minute-by-minute freezing behaviour of Ctrl and *Adrb1*-OE mice across conditioning and recall sessions. After three-shock FC (c), freezing was measured during the 1st (d) and 2nd FR (e). After one-shock FC (f), freezing was measured during the 1st (g) and 2nd FR (h), with *Adrb1*-OE mice showing significantly more freezing. In a separate three-shock FC cohort (i), freezing was measured during the 1st FR (j) and CtxB exposure (k), where *Adrb1*-OE mice also showed increased freezing. n = 5–7 mice per group (c–e), 13 per group (f–h), and 13–14 per group (i–k). Two-way ANOVA with Šídák's test. Data are mean ± SEM. *p < 0.05; ns, not significantly different. Sample sizes and replicates in Supplementary Table 1.

# Reporting Summary

## Statistics

For all statistical analyses, confirm that the following items are present in the figure legend, table legend, main text, or Methods section.

| n/a | Confirmed | |
|---|---|---|
| ☐ | ☒ | The exact sample size (*n*) for each experimental group/condition, given as a discrete number and unit of measurement |
| ☐ | ☒ | A statement on whether measurements were taken from distinct samples or whether the same sample was measured repeatedly |
| ☐ | ☒ | The statistical test(s) used AND whether they are one- or two-sided<br>*Only common tests should be described solely by name; describe more complex techniques in the Methods section.* |
| ☐ | ☒ | A description of all covariates tested |
| ☐ | ☒ | A description of any assumptions or corrections, such as tests of normality and adjustment for multiple comparisons |
| ☐ | ☒ | A full description of the statistical parameters including central tendency (e.g. means) or other basic estimates (e.g. regression coefficient) AND variation (e.g. standard deviation) or associated estimates of uncertainty (e.g. confidence intervals) |
| ☐ | ☒ | For null hypothesis testing, the test statistic (e.g. *F*, *t*, *r*) with confidence intervals, effect sizes, degrees of freedom and *P* value noted<br>*Give P values as exact values whenever suitable.* |
| ☒ | ☐ | For Bayesian analysis, information on the choice of priors and Markov chain Monte Carlo settings |
| ☒ | ☐ | For hierarchical and complex designs, identification of the appropriate level for tests and full reporting of outcomes |
| ☐ | ☒ | Estimates of effect sizes (e.g. Cohen's *d*, Pearson's *r*), indicating how they were calculated |

*Our web collection on statistics for biologists contains articles on many of the points above.*

## Software and code

Policy information about availability of computer code

| | |
|---|---|
| Data collection | QuantStudio (Applied Biosystems, 12K), CytExpert (Beckman Coulter, version 1.0.3), FlowJo (Becton Dickinson, version 10.9.0), Fluoview (FV3000, Olympus). Freezing was automatically measured throughout the testing trial by the TimeFZ4 software (version 2021_7_29, Ohara). |
| Data analysis | Softwares: NeuroInfo (MBL Bioscience, BNS-200), ANTs (Stnava, https://stnava.github.io/ANTs/), JupyterLab (3.6.7, Project Jupyter, https://jupyterlab.readthedocs.io/en/latest/), Python (v3.00, Python Software Foundation), Numpy (1.26.4, https://www.nature.com/articles/s41586-020-2649-2), Matplotlib (3.8.0, https://ieeexplore.ieee.org/document/4160265/authors#authors), Pandas (2.1.4, NumFOCUS, Inc., https://pandas.pydata.org/), Scipy (1.11.4, https://www.nature.com/articles/s41592-019-0686-2) , sklearn (1.2.2, RRID:SCR_002577), ImageJ (NIH), GraphPad Prism (10.1.2, GraphPad), RStudio (2023.12.1+402, Posit Software, PBC), R4.3.2 (R Core Team, RRID:SCR_001905; https://www.R-project.org/).<br>Codes: https://github.com/BrainImageAnalysis/Astrocyte3DMapping/, https://github.com/Jun-Nagai-Lab/Dewa-et-al/ |

For manuscripts utilizing custom algorithms or software that are central to the research but not yet described in published literature, software must be made available to editors and reviewers. We strongly encourage code deposition in a community repository (e.g. GitHub). See the Nature Portfolio guidelines for submitting code & software for further information.

## Data

Policy information about availability of data

All manuscripts must include a data availability statement. This statement should provide the following information, where applicable:
- Accession codes, unique identifiers, or web links for publicly available datasets
- A description of any restrictions on data availability
- For clinical datasets or third party data, please ensure that the statement adheres to our policy

All the single-cell transcriptomics data are available at Gene Expression Omnibus with accession identifier GSE272414. For the scRNA-seq analysis, we used the mouse reference genome mm10 (GENCODE vM23/Ensembl 98), downloaded from the 10x Genomics reference data site https://cf.10xgenomics.com/supp/cell-exp/refdata-gex-mm10-2020-A.tar.gz. All raw replicate data values used to generate the figures and the associated statistical tests are provided in Supplementary Table 1. Fos-mNG astrocyte counts in 677 brain regions in NoFC, FC, NoFR and FR groups are provided in Supplementary Table 2. All raw data supporting the findings of this study are available at RIKEN CBS Data Sharing Platform https://neurodata.riken.jp/id/20250831-001 (DOI: 10.60178/cbs.20250831-001).

## Research involving human participants, their data, or biological material

Policy information about studies with human participants or human data. See also policy information about sex, gender (identity/presentation), and sexual orientation and race, ethnicity and racism.

| | |
|---|---|
| Reporting on sex and gender | n/a |
| Reporting on race, ethnicity, or other socially relevant groupings | n/a |
| Population characteristics | n/a |
| Recruitment | n/a |
| Ethics oversight | n/a |

Note that full information on the approval of the study protocol must also be provided in the manuscript.

# Field-specific reporting

Please select the one below that is the best fit for your research. If you are not sure, read the appropriate sections before making your selection.

☒ Life sciences   ☐ Behavioural & social sciences   ☐ Ecological, evolutionary & environmental sciences

For a reference copy of the document with all sections, see nature.com/documents/nr-reporting-summary-flat.pdf

# Life sciences study design

All studies must disclose on these points even when the disclosure is negative.

| | |
|---|---|
| Sample size | Sample and group sizes were determined based on data from similar models used in the cited papers and the authors' previous publications. Specifically: brain-wide imaging (PMID: 35379803, 30692687), PHP.eB AAV validation (PMID: 34139149), Fos+ cell histological analyses (PMID: 35379803, 39549697, 30692687), c-Fos IHC (PMID: 34139149, 39506118), in vitro Fos screening assay (PMID: 32187527), fiber photometry (PMID: 38547869, 38514778), single-cell transcriptomics (PMID: 38326616), RNAscope (PMID: 31031006), behavioral tests (PMID: 35379803, 30692687, 39549697), and ex vivo astrocyte Ca²⁺ imaging (PMID: 34139149, 31031006). |
| Data exclusions | Tissue samples that yielded insufficient cell counts (<thirty cells) during scRNA-seq quality control were excluded from the study. |
| Replication | To verify the reproducibility of the experimental findings, all data collection was done in multiple batches, at least twice independently as stated in figure legends. All attempts at replication were successful. |
| Randomization | All samples were randomly allocated into treatment groups. |
| Blinding | Investigators were blinded to group allocation during data acquisition for imaging, sequencing, and in vivo recordings. For behavioral experiments, blinding was implemented by using numerical mouse IDs, and whenever possible, data collection and analysis were carried out by different individuals. For in vitro experiments, blinding was not performed because the experimenters needed to know the conditions of each well to conduct the assays appropriately. |

# Reporting for specific materials, systems and methods

We require information from authors about some types of materials, experimental systems and methods used in many studies. Here, indicate whether each material, system or method listed is relevant to your study. If you are not sure if a list item applies to your research, read the appropriate section before selecting a response.

## Materials & experimental systems

| n/a | Involved in the study |
|---|---|
| ☐ | ☒ Antibodies |
| ☐ | ☒ Eukaryotic cell lines |
| ☒ | ☐ Palaeontology and archaeology |
| ☐ | ☒ Animals and other organisms |
| ☒ | ☐ Clinical data |
| ☒ | ☐ Dual use research of concern |
| ☒ | ☐ Plants |

## Methods

| n/a | Involved in the study |
|---|---|
| ☒ | ☐ ChIP-seq |
| ☐ | ☒ Flow cytometry |
| ☒ | ☐ MRI-based neuroimaging |

# Antibodies

| Antibodies used | Primary antibodies: |
|---|---|
| | Rat anti-GFAP, Thermo Fisher Scientific, 13-0300, RRID: AB_2532994, 1:500 for IHC |
| | Mouse anti-S100β, Sigma-Aldrich, S2532, RRID: AB_477499, 1:500 for IHC |
| | Mouse anti-RFP, MBL, M155-3, RRID: AB_1278880, 1:500 for IHC |
| | Rabbit anti-RFP, Rockland, 600-401-379, RRID: AB_2209751, 1:500 for IHC |
| | Rabbit anti-c-Fos, Synaptic Systems, 226003, RRID: AB_2231974, 1:5000 for IHC, 1:1000 for ICC |
| | Mouse anti-NET, Mab Technologies, NET05-2, RRID: AB_2571639, 1:2000 for IHC |
| | Goat-Sox9 (1:100, R&D systems, AF3075), 1:100 for IHC |
| | Mouse anti-c-Fos, Santa Cruz, sc-166940, RRID: AB_10609634, 1:50 for ICC |
| | Rabbit anti-Iba1, Fujifilm, 019-19741, RRID: AB_839504, 1:1000 for ICC |
| | Chicken anti-Tuj1, Novus Biologicals, NB100-1612, RRID: AB_10000548, 1:1000 for ICC |
| | Mouse anti-Olig2, Millipore, MABN50, RRID: AB_10807410, 1:100 for ICC |
| | Rabbit anti-mGluR3, Abcam, ab166608, RRID: AB_2833092, 1:500 for ICC |
| | CD11b-BV786 (M1/70, Lot 4317109, BD Bioscience), 1:200 for FACS |
| | CD45-APC-Cy7 (30-F11, Lot B442084, BioLegend), 1:200 for FACS |
| | O1-eFluor660 (50-6506-82, Lot 2848347, eBioscience), 1:200 for FACS |
| | ASCA2-PE-Cy7 (130-116-246, Lot, 5231104462, Miltenyi Biotec), 1:200 for FACS |
| | |
| | Secondary antibodies (1:1000 dilution): |
| | Alexa Fluor 488 donkey anti-rat, Abcam, ab150153, RRID: AB_2737355 |
| | Alexa Fluor 488 donkey anti-chicken, ThermoFisher, A78948, RRID: AB_2921070 |
| | Alexa Fluor 568 donkey anti-mouse, Abcam, ab175700, RRID: AB_3083693 |
| | Alexa Fluor 568 donkey anti-goat, Abcam, ab150129, RRID: AB_2687506 |
| | Alexa Fluor 568 donkey anti-rabbit, Abcam, ab175470, RRID: AB_2783823 |
| | Alexa Fluor 647 donkey anti-rabbit, Abcam, ab150067, RRID: AB_2894821 |
| | Alexa Fluor 647 donkey anti-mouse, Abcam, ab150111, RRID: AB_2890625 |
| | Alexa Fluor 647 donkey anti-chicken, Jackson ImmunoResearch, 703-605-155, RRID: AB_234039 |
| Validation | All antibodies were either validated by the manufacturers for the specified species and applications or previously confirmed in the literature. Optimal dilutions were determined empirically on mouse brain tissues, and secondary-antibody-only controls confirmed specificity (no signal without primary). Details for each antibody are as follows: |
| | Rat anti-GFAP (Thermo Fisher Scientific #13-0300, clone 2.2B10, RRID: AB_2532994): Monoclonal IgG2a antibody raised in rat against enriched bovine GFAP filaments. The immunogen was enriched bovine glial filaments. The manufacturer's documentation states that it reacts with mouse, rat, human, and bovine GFAP and is validated for immunohistochemistry and related applications.(https://www.biocompare.com/9776-Antibodies/7012889-GFAP-Glial-Fibrillary-Acid-Protein-Monoclonal-Antibody-2-2B10/). Previously used in PMID: 38302444. |
| | Mouse anti-S100β (Sigma-Aldrich #S2532, clone SH-B1, RRID: AB_477499): Mouse IgG1 monoclonal antibody (clone SH-B1) against S100β. The immunogen was S100β purified from bovine brain. The manufacturer's documentation states that the antibody has been validated for immunohistochemistry and other assays, with broad species reactivity (reacts with human, rat, bovine, pig, goat, etc.) (https://www.sigmaaldrich.com/US/en/product/sigma/s2532?srsltid=AfmBOorJCzKHm_kMMaejg31phReliQl0VB2jbal0Tfv-qbF-cYqt5ahV) . Previously used in PMID: 34139149. |
| | Mouse anti-RFP (MBL International M155-3, clone 8D6, RRID: AB_1278880): Mouse IgG1 monoclonal antibody (clone 8D6) against red fluorescent protein (RFP). The manufacturer's documentation states that the antibody has been validated for immunocytochemistry and immunohistochemistry (https://www.mblbio.com/bio/g/dtl/A/index.html?pcd=M155-3). Previously used in PMID: 37323585. |
| | Rabbit anti-RFP (Rockland #600-401-379, polyclonal, RRID: AB_2209751): Rabbit polyclonal antibody affinity-purified against full-length Discosoma RFP. The immunogen was a recombinant RFP (234 amino acids from Discosoma sp.) The manufacturer's documentation states that the antiserum was immunoaffinity-purified and extensively adsorbed to remove cross-reactivity with mammalian proteins (https://www.rockland.com/categories/primary-antibodies/rfp-antibody-pre-adsorbed-600-401-379/?srsltid=AfmBOooSy5itbhp4CoqwicBqO7rJ2R_FvAMkOQWn17_MoyAj0Ylu6SFm). Previously used in PMID: 34139149. |
| | Rabbit anti-c-Fos (Synaptic Systems #226 003, polyclonal, RRID: AB_2231974): Rabbit polyclonal antibody against c-Fos. Immunogen was a synthetic peptide corresponding to amino acids 2–17 of rat c-Fos (UniProt P12841). The manufacturer's documentation states that the antibody has been validated it is affinity-purified and validated for immunohistochemistry on rodent brain tissue (https://fnkprddata.blob.core.windows.net/domestic/data/datasheet/SS2/226003.pdf). Previously used in PMID: 34139149. |
| | Mouse anti-NET (Norepinephrine Transporter; Mab Technologies #NET05-2, clone 2-3B2, RRID: AB_2571639): Mouse IgG2b |

monoclonal antibody (clone 2-3B2) raised against a peptide from the mouse norepinephrine transporter (NET) .The manufacturer's documentation states that it specifically recognizes mouse and rat NET on neuronal membranes but does not bind human NET (species specificity confirmed by sequence homology) (https://mabtechnologies.com/categories/product/7-norepinephrine-transporter-mouse-net05-2). Previously used in PMID: 29133432.

Goat anti-Sox9 (R&D Systems AF3075, polyclonal): Goat polyclonal IgG antibody against Sox9. Immunogen was recombinant human SOX9 (Met1–Lys151) expressed in E. coli. The manufacturer's documentation states that the antibody has been validated antigen affinity purification in immunocytochemistry and Western blot (primarily on human samples) (https://www.rndsystems.com/products/human-sox9-antibody_af3075). Previously used in PMID: 28336567.

Mouse anti-c-Fos (Santa Cruz Biotechnology sc-166940, clone E-8, RRID: AB_10609634): Mouse monoclonal IgG1 κ antibody (clone E-8) against c-Fos. The epitope recognized is mapped to amino acids 128–152 of human c-Fos. The manufacturer's documentation states that this clone is validated for immunofluorescence and immunohistochemistry (paraffin) in mouse, rat, human, and even zebrafish tissues (https://www.scbt.com/p/c-fos-antibody-e-8?srsltid=AfmBOoq6tOL6gXYJG5QLW-dcHYBvgg9iCpBg6OakoaDJTwh0HV3lwoQ9). Previously used in PMID: 37229433.

Rabbit anti-Iba1 (FUJIFILM Wako #019-19741, polyclonal, RRID: AB_839504): Rabbit polyclonal antibody against Iba1 (microglial calcium-binding protein). The immunogen is a synthetic peptide corresponding to the C-terminal amino acids of Iba1, a region conserved in mouse, rat, and human. The antibody was affinity-purified and is recommended for immunocytochemistry/immunohistochemistry (but not Western blot) by the manufacturer (https://labchem-wako.fujifilm.com/us/product/detail/W01W0101-1974.html). Previously used in PMID: 33414461.

Chicken anti-βIII Tubulin (TuJ1) (Novus Biologicals NB100-1612, polyclonal, RRID: AB_10000548): Chicken polyclonal IgY antibody against neuron-specific class III β-tubulin (TuJ1 antigen). This antibody was generated by immunizing chickens with synthetic peptides from human TUBB3, and the IgY was affinity-purified. The manufacturer's documentation states that the antibody has been validated for immunofluorescence and immunohistochemistry, and is widely used as a pan-neuronal marker. (https://www.novusbio.com/products/beta-iii-tubulin-antibody_nb100-1612?srsltid=AfmBOopjF61F7ST1-ktgtvSZnbk8thSNz3TsNcuO5_haFp-BGAt_Oczo). Previously used in PMID: 38843836.

Mouse anti-Olig2 (Millipore Sigma MABN50, clone 211F1.1, RRID: AB_10807410): Mouse monoclonal IgG2a antibody (clone 211F1.1) against Olig2 (oligodendrocyte lineage transcription factor). The immunogen was a recombinant human Olig2 protein. The manufacturer's documentation states that the antibody has been validated in Western blot, immunocytochemistry, and immunohistochemistry; it reacts with Olig2 in mouse, rat, and human samples (https://www.merckmillipore.com/INTL/en/product/Anti-Olig2-Antibody-clone-211F1.1,MM_NF-MABN50). Previously used in PMID: 17428828.

Rabbit anti-mGluR3 (Abcam ab166608, clone EPR9009(2), RRID: AB_2833092): Rabbit recombinant monoclonal antibody against mGluR3. The immunogen was a proprietary synthetic peptide from human mGluR3 (GRM3). The manufacturer's documentation states that this clone is validated for ICC/IF and flow cytometry and recognizes mGluR3 in human, mouse, and rat samples (https://www.labome.com/knockout-validated-antibodies/mGlu3-antibody-knockout-validation-Abcam-ab166608.html). Its specificity was further confirmed by knockout validation (showing loss of staining in Grm3–/– tissues). In our hands, this antibody labeled mGluR3 in transfected cell cultures (consistent with manufacturer's validation in HEK293 cells and brain tissue).

CD11b–BV786 (BD Biosciences, clone M1/70): A rat monoclonal IgG2b κ antibody (clone M1/70) against CD11b (integrin α_M) conjugated to Brilliant Violet 786. The manufacturer's documentation states that this clone is a well-characterized antibody that binds mouse CD11b (with demonstrated cross-reactivity to human CD11b) (https://www.bdbiosciences.com/en-us/products/reagents/flow-cytometry-reagents/research-reagents/single-color-antibodies-ruo/bv786-rat-anti-cd11b-integrin-m.569504). Previously used in  PMID: 30760929.

CD45–APC-Cy7 (BioLegend, clone 30-F11): A rat monoclonal IgG2b κ antibody (clone 30-F11) against pan-CD45, conjugated to APC/Cy7. Clone 30-F11 recognizes all isoforms of mouse CD45 (including both CD45.1 and CD45.2 alleles). The manufacturer's documentation states that the antibody has been validated for flow cytometry (BioLegend reports routine QC on mouse leukocytes) and does not distinguish between leukocyte subsets since it binds a common epitope on CD45 (https://www.biolegend.com/de-at/products/apc-cyanine7-anti-mouse-cd45-antibody-2530). Previously used in PMID: 30760929.

Oligodendrocyte Marker O1–eFluor660 (Invitrogen eBioscience, clone O1): A mouse IgM monoclonal antibody (clone O1) against galactocerebroside and sulfated lipids on late oligodendrocyte progenitors, conjugated to eFluor™ 660 (equivalent to Alexa 647). The manufacturer's documentation states that this antibody (originally O1 from Sommer and Schachner's work) reacts with oligodendrocytes of multiple species (mouse, rat, human, chicken) because it targets a conserved lipid antigen. (https://www.thermofisher.com/antibody/product/Oligodendrocyte-Marker-O1-Antibody-clone-O1-Monoclonal/50-6506-82). Previously used in PMID: 39009472.

ACSA-2–PE-Cy7 (Miltenyi Biotec #130-116-246, clone REA969): A recombinant human IgG1 monoclonal antibody (REAfinity clone REA969) specific for mouse Astrocyte Cell Surface Antigen-2 (ACSA-2), conjugated to a PE/Cy7-equivalent fluorochrome (PE–Vio770). The manufacturer's documentation states that the antibody has been validated for flow cytometry as a pan-astrocyte marker in mice (https://www.labome.com/product/Miltenyi-Biotec/130-116-246.html). Previously used in PMID: 37845031.

Mouse IGFBP-2 Antibody (R&D Systems #MAB797): Mouse recombinant monoclonal antibody against IGFBP-2. The manufacturer's documentation states that this clone is Measured by its ability to neutralize IGFBP-2 inhibition of IGF-II-dependent proliferation in the MCF-7 human breast cancer cell line (https://www.rndsystems.com/products/mouse-igfbp-2-antibody-150101_mab797). Previously used in PMID: 36042312.

Rat IgG control antibody (R&D Systems # 6-001-F): Polyclonal Rat IgG. The manufacturer's documentation states that it offers controls for flow cytometry, immunohistochemistry, and Western blotting.
 (https://www.rndsystems.com/products/mouse-igfbp-2-antibody-150101_mab797). Previously used in PMID: 36042312.

# Eukaryotic cell lines

Policy information about cell lines and Sex and Gender in Research

| | |
|---|---|
| Cell line source(s) | AAVpro 293T cells (Clontech, 632273). |
| Authentication | The cells were authenticated by the supplier using the STR-based method. |
| Mycoplasma contamination | The cell line tested negative for mycoplasma contamination. |
| Commonly misidentified lines (See ICLAC register) | No misidentified cell lines were used. |

# Animals and other research organisms

Policy information about studies involving animals; ARRIVE guidelines recommended for reporting animal research, and Sex and Gender in Research

| | |
|---|---|
| Laboratory animals | Young adult (8-12 weeks old) wild-type C57BL/6NCrSlc mice obtained from the Japan SLC, Adra1a flox mice obtained from the RIKEN (RBRC11837) and Adrb1 flox mice (Accession Number: CDB1068K generated by RIKEN Center for Biosystems Dynamics Research https://large.riken.jp/distribution/mutant-list.html) and Fos2A-iCre/ERT2 (TRAP2, JAX Stock 030323) mice obtained from the Jackson Laboratory were used for in vivo experiments. TRAP2 mice were bred and maintained in our animal facility, backcrossed with C57BL/6NCrSlc at least 8 times before use for experiments. Wild-type C57BL/6NCrSlc female mice were obtained from the Japan SLC to obtain postnatal day 0-1 (P0-1) pups. Experiments were done using both male and female mice. Number of mice used in each experiment is listed in the main text and/or figure legends accordingly. Mice were housed in groups of two to five in a temperature- and humidity-controlled room (23 ± 3 °C, 45 ± 5% humidity) under a 12-hour (h) light−dark cycle (lights on from 7 AM to 7 PM) and given ad libitum access to water and laboratory mouse diet at all times. After fiber implantation, the mice were single housed. |
| Wild animals | The study did not involve wild animals. |
| Reporting on sex | Animals of both sexes were included in the study and sex was not considered in random assignments to experimental groups, particularly due to the difficulty of distinguishing sexes in neonatal mice for in vitro experiments. |
| Field-collected samples | The study did not use field-collected samples. |
| Ethics oversight | All experiments were approved by the RIKEN Animal Care and Use Committee. |

Note that full information on the approval of the study protocol must also be provided in the manuscript.

# Plants

| | |
|---|---|
| Seed stocks | n/a |
| Novel plant genotypes | n/a |
| Authentication | n/a |

# Flow Cytometry

## Plots

Confirm that:

☒ The axis labels state the marker and fluorochrome used (e.g. CD4-FITC).

☒ The axis scales are clearly visible. Include numbers along axes only for bottom left plot of group (a 'group' is an analysis of identical markers).

☒ All plots are contour plots with outliers or pseudocolor plots.

☒ A numerical value for number of cells or percentage (with statistics) is provided.

## Methodology

| | |
|---|---|
| Sample preparation | Mice were anesthetized with isoflurane 90 min after the initiation of behavioral paradigms and transcardially perfused with 1x PBS (30 mL for 2 min). The amygdala was freshly dissected out using a Brain Matrix (Ted Pella, 15067) and was placed into cold Hanks' balanced salt solution (HBSS) (14025092, Gibco) containing 10 mM HEPES (17514-15, Naclai Tesque), 0.54 % glucose (16806-25, Naclai Tesque), 5.0 µg/mL Actinomycin D (A1410, Sigma-Aldrich), 10 µM Triptolide (T3652, Sigma-Aldrich) and 27.1 µg/mL Anisomycin (A9789, Sigma-Aldrich), as previously described (Yamasaki et al., Commun Biol 2024; Marsh et al., Nat Neurosci 2022). Then, the tissue block was incubated for 15 min at 37°C in a 3 mL enzyme solution (1x HBSS, 10 mM HEPES, 0.54 % glucose, 0.5 mM EDTA (15575020, Invitrogen), 1.0 mM L-Cys (10309-41, Naclai Tesque), 5.0 µg/mL Actinomycin D, 10 µM Triptolide and 27.1 µg/mL Anisomycin) containing 20 U of papain per mouse (LS003127, Worthington Biochemical). The tissue was then homogenized by passing through a 18G needle and was further incubated for 15 min at 37 °C. Following homogenization using a 20G needle, the resulting cell suspension was centrifuged at 500 x g for 5 min at 4 °C. The homogenate was separated by gradient centrifugation with 30% Percoll (P1644, Sigma-Aldrich) in 1x PBS at 500 x g for 25 min at 4°C (no brake). The pellet containing astrocytes at the bottom of the tube was then collected and washed once with PBS containing 2 % FBS and 10 mM EDTA before staining. Fc receptors were blocked with Fc block (2.4G2, BD Bioscience) for 10 min at 4°C before incubation with primary antibodies. Cells were stained with antibodies directed against CD11b-BV786 (M1/70, BD Bioscience), CD45-APC-Cy7 (30-F11, BioLegend), O1-eFluor660 (50-6506-82, eBioscience) and |

ASCA2-PE-Cy7 (130-116-246, Miltenyi Biotec) for 40 min at 4°C. Additionally, cells were treated with hashtag antibodies to label source samples (TotalSeqB0301-B0310, B0312 and B0314, BioLegend).

Instrument
Cells were sorted using a CytoFlex SRT (Beckman Coulter)

Software
Data were acqired with CytExpert software (Beckman Coulter). Post aquisition analysis was perfomed using FlowJo software, version 10.9.0.

Cell population abundance
The cell population abundances are provided in the plots depicting the representative gating strategies of each experiment.

Gating strategy
In all experiments, small debris was removed with the preliminary FSC/SSC gate. Single cells were obtained by doublet exclusion. Further, gating strategies for the respective experiments are provided in the figures.

☒ Tick this box to confirm that a figure exemplifying the gating strategy is provided in the Supplementary Information.

