## [Peer Review File · Nature]

The astrocytic ensemble is a multiday eligibility trace stabilizing memory

Corresponding Author: Dr Jun Nagai

Version 0:

Reviewer comments:

Referee #1

(Remarks to the Author)

Dewa and colleagues have investigated the role of the immediate early gene Fos in astrocytes in contextual fear conditioning. Using a new approach for brain wide tagging of c-Fos expressing astrocytes the authors report particularly high increases of such astrocytes in the amygdala during fear recall (FR). They also found that noradrenaline (NA) can induce c-Fos expression in astrocytes and that fear conditioning (FC) and FR differentially affect NA signals and astrocytic Ca²⁺ and cAMP levels in the amygdala. Lowering of NA signalling and of the activity of amygdalar neurons reduced the number of c-Fos positive astrocytes. Furthermore, FC and FR affected the expression profiles of astrocytes, including the expression of beta1-adrenoreceptors. Overexpressing the latter in astrocytes was sufficient to increase the number of tagged astrocytes, to increase and generalize fear behaviour and to increase neuronal c-Fos expression in the amygdala. These experiments directly and elegantly demonstrate that contextual fear conditioning profoundly alters expression profiles of astrocytes and the involvement of NA in signalling in that. They also reveal that astrocytic manipulation can increase freezing behaviour.

Comments:

1) Throughout the manuscript it is suggested that the c-Fos positive/tagged astrocyte have an important and specific role but this is not directly addressed. It seems that the author did not use any astrocytic manipulation that is specific to the c-Fos positive/tagged astrocytes, which would be necessary to demonstrate such a role. For instance, what happens to freezing behaviour when only tagged/c-Fos expressing astrocytes are inhibited or when beta1-adrenoreceptors are removed from those selectively? The effect of overexpression of beta1-adrenoreceptors on memory are very interesting but conceptually similar findings of memory improvement via astrocytes have been reported before (Adamsky et al., 2018; Iwai et al., 2021).

2) The assay for brain-wide tagging of c-Fos expressing astrocytes in behavioural experiments is novel and interesting, but there are some open questions. The authors compare their results on astrocytes to the well-known neuronal c-Fos expression increase in contextual fear conditioning, which is important because the lack of astrocytic c-Fos upregulation after FC is interesting and some reference is helpful. However, it is not clear how exactly neuronal c-Fos expression in Fig. 1e-f was determined and quantified. Also, did the authors compare their findings with tagging astrocytes systematically with immunohistochemistry? Is it possible that the tagging significantly underestimates the number of c-Fos expressing astrocytes because the number of cells in the illustration (e.g. Fig. 1b, 4) seem relatively low given the density of astrocytes in the tissue and the volume of the brain regions?

3) A related question is about Fig. 5b. There, the percentage of Fos-positive amygdalar astrocytes is about the same in the FC and FR condition and Fos expression levels are higher in FR compared to FC. However, in Fig. 1d there is a large difference (> 30 fold) in c-Fos expressing/tagged astrocytes between FC and FR. It is therefore currently unclear how variations in c-Fos expression in astrocytes are captured by tagging and counting.

4) In the imaging experiments in Fig. 3d and Ext. Fig. 5e there is variable and sometimes rapid and strong increase in the reported cAMP levels already in the home cage, which casts doubt on the reliability of the recordings and their analysis.

5) The quantification of freezing in Ext. Fig. 5b reveals that in the control context B (rightmost panel), animals freeze almost as much as during fear-conditioning (leftmost panel). What is the explanation for this unexpected behaviour? A similar illustration/quantification of freezing behaviour should accompany other key findings related to contextual fear conditioning (e.g. Fig. 1, Fig. 6).

6) The method and results for silencing of amygdalar 'engram' neurons and their counting is barely documented, which makes it hard to appreciate the experiment.

7) The authors show that NA increases c-Fos expression by cultured cortical astrocytes through convergent signalling of beta-adrenoreceptors via Gs and mGluR3. Astrocytes can substantially change their expression profiles in culture, as acknowledged by the authors. Therefore, it is uncertain whether NA signals in a similar manner in situ. In vivo and in more intact preparations, there is plenty of experimental evidence that Gq-coupled alpha1-adrenoreceptor play an important role and the authors demonstrate in their current study that Gq-activation in astrocytes is sufficient to increase the number c-Fos expressing/tagged astrocytes (Ext. Fig. 2). If the authors wish to demonstrate a role of beta1-adrenoreceptors in astrocytes they could, for instance, combine the experimental approach shown in Fig. 2l-m with manipulation/inhibition of beta-adrenoreceptors and Gs-signalling selectively in astrocytes in situ.

8) A minor comment regarding the term 'astrocytic engram', which is used throughout the manuscript: By many definitions, engrams are the physical/chemical changes in brain elicited by learning (e.g. representing a memory) and the engram cells are those that undergo these changes. At the moment, the study indicates that astrocytes can facilitate memory formation/recall but whether they encode the memory (e.g. a specific context) has not been addressed.

Adamsky A, Kol A, Kreisel T, Doron A, Ozeri-Engelhard N, Melcer T, Refaeli R, Horn H, Regev L, Groysman M, London M & Goshen I (2018). Astrocytic Activation Generates De Novo Neuronal Potentiation and Memory Enhancement. *Cell* 174, 59–71.

Iwai Y, Ozawa K, Yahagi K, Mishima T, Akther S, Vo CT, Lee AB, Tanaka M, Itohara S & Hirase H (2021). Transient Astrocytic Gq Signaling Underlies Remote Memory Enhancement. *Front Cell Neurosci* 15, 18.

Referee #2

(Remarks to the Author)

In this study, Dewa et al investigate a relevant topic in neuroscience, the role of astrocytes in memory stabilization. By using cutting-edge approaches based on brain-wide Fos tagging and imaging method, authors have shown brain-wide behaviourally relevant astrocyte ensembles (BAE) recruited by emotional experiences (fear conditioning), being enriched in regions with neuronal engrams previously described. The further characterization of those BAE showed the role of NA and Igfbp2 signaling, and the pharmacological and genetic perturbation of these signaling pathways modulate memory stability and precision.

Showing the astrocyte memory engrams is an important finding using the novel methodology, but to have a comprehensive idea of the magnitude and relevance it is needed to have the neuronal engrams too. Although based on published database (Roy DS et al *Nat Commun* 2022), the fos expression in neurons is not performed in this study. The analysis of the simultaneous temporal profile following dual viral approach to tag both cellular engrams is strongly required; without it, the study gets incomplete and miss the unique opportunity to bring a milestone in the fields of neuron-glia interaction and memory consolidation. Considering the critical role of amygdala in fear conditioning (FC), the labeling of both ensembles and the dynamic changes over time would show outstanding information to understand how memory is stored and recall by both neuron and astrocyte engrams.

Additionally, the study shows important gaps between different experimental dataset:

The body of the work is performed following the strategy described in Roy et al *Nat Commun* 2022, that is, one week later of i.p. 4-OHT brains are analyzed and distribution of fos expression detected in BAE. However, authors study the time window of astrocyte Fos tagging by analyzing visual responses and found that the majority of tagging in visual cortex occurred within a six-hour window centered around the time of 4-OHT injection (Extended Data Fig. 3c,d). Why this change in the behavioral paradigm? The BAE in these cases are completely different. It is hard to understand the relevance for the study if Fos tagging is not analyzed by FC following this temporal scheme.

It is found that visual stimuli can induce Fos expression in astrocytes but barely by FC. It is intriguing that such strong stimuli, foot shock, that boost the neuronal activity in different brain areas (Roy DS et al *Nat Commun* 2022) do not induce significant changes in astrocytes at the level of Fos expression (Fig 1), but they can respond reliably with cAMP and Ca signals to NA signaling (Fig 3d,e). Then, Fig 5b shows that amygdalar astrocytes show enhances fos expression after FC, which it seems significant vs NoFC. The message of the study is confusing.

Even more, the expression of Adra1a is extremely high after NoFR and FR, but there is no mention to such results in the discussion. There is a progressive enhancement of expression in the experimental groups analyzed. Then, how is possible that enhanced Adra1a expression in amygdalar astrocytes, significantly higher than Adrb1 in all conditions, has been ignored in this study?

Experiments of overexpression of Adrb1 are important and reveal the role of astrocytes for fear generalization, however, to understand the role of this receptors in the particular ensembles, that is Ast3 subpopulation (Fig 5d), instead of bulk overexpression in all cells a more precise approach is required to tackle only such subpopulation.

Referee #3

(Remarks to the Author)

The paper "The astrocytic engram is a multiday eligibility trace for memory stabilization" by Dewa et al. from the Nagai group attempts to recognize astrocytic engrams. They start by using the self-developed whole brain BAE (Behaviorally-relevant Astrocyte Ensembles) technique in TRAP mice, to tag cFos expressing astrocytes. They report an increase in the number of tagged astrocytes, especially in the amygdala, in the recall phase of the fear condition (FC), but not during acquisition. From here on they attempt to show the mechanisms involved: 1) they show that the astrocytic engram depends on local neuronal activity and on the secretion of NA from the LC. 2) They further show that amygdalar astrocytes express more NA receptors during recall, and the astrocytes with high cFos expression levels have high NA β 1Rs levels. 3) They show that overexpression of β 1Rs in amygdalar astrocytes leads to more cFos+ engram astrocytes.

The authors employ a variety of techniques (their very own BAE, other multiple virus injections, behavior, photometry, cultures, FACS, single-cell RNA-seq, IHC), but they always use them close to question, without ever nailing the main point – that the cFos+ astrocytes are important to memory. All the beautiful experiments prove without doubt is that fear conditioning affect NA, and that NA affect astrocytes – both things are already known for a long time. They DO NOT show that NA affects the astrocyte engram, they don't do scRNA-seq of the astrocyte engram alone, nor do they show that the astrocytes of the engram are important to memory (or to anything).

In the summery they say: "However, our findings suggest that the astrocytic engram may not serve as a direct memory trace encoding experience specific information like neurons because the augmentation of astrocytic engram did not alter initial fear recall. Instead, it likely functions as an eligibility trace for memory stabilization over days, creating a permissive and specialized microenvironment within circuits that supports the stabilization of memories related to repeated experiences". This is totally speculative. Nowhere in the paper do they show it directly. Nowhere do they manipulate the BAE astrocytes (it is doable! they can express *Adrb1* only in them, for example) and test the effect on behavior. Sadly, I think that this work is not suitable for Nature.

Major:

1) Fig. 1: the BAE numbers in the 4 conditions were compared to neuronal count in 3 conditions (there was no NoFR neurons) from Roy et al. (Nat. Commun, 2022). How? The number of engram astrocytes is an order of magnitude smaller than the neuronal one – what does it mean?

2) Fig 2, in-vivo (j-m): the authors write: "Thus, our data provide strong evidence that the convergence of local neural signals and long-range projecting NA-ergic modulatory signals is a cue for inducing astrocyte Fos in LA/B in vivo". Not at all! This is pure speculation. Your data shows in a non-behavioral study, that LC plus local neurons external stimulation affects the number of cFos astrocytes. Nothing less, nothing more.

3) Behavior, general: Very bold sayings are given in the paper based on behavioral studies, but it is hard to find two of them that are exactly the same:

- Fig 4: Propranolol is given before and after recall, whereas CNO for both LC, local neurons and neuronal engram is given just before recall – why?
- Fig 5 m-n: "We assessed freezing behavior during subsequent FR sessions in CtxA the next day, with repeated 24-hour cycle infusions and FR sessions over five days (Fig. 5m)". What? Why? Which FR session are showing in 5n? For statistical analysis you used the Wilcoxon matched-pairs signed rank test – but the correct test should be a two-way ANOVA: treatment(IgG/Igfbp2) X time (before/after), and not each one by t-test - is it still significant?
- Fig 6e-h: FC with one shock. The only one in the paper, compared to three shocks. The significant finding to 1-shock was in 2 days FR, and the is not done at all for 3-shocks, but a completely different exp. Why?

4) Fig 6 a-c: this is the closest you came to manipulating the number of engram-astrocytes, yet you don't show the behavioral result – why??

5) Rows 101-103: "While astrocyte Fos induction has been demonstrated in response to various stimuli such as G protein-coupled receptor (GPCR) activation 40–43, behaviourally relevant stimuli 42,44 and astrocyte-neuron signaling 25,26,33 ...". This is the basis of cFos tagging in astrocytes. But it is misrepresenting some of these papers:

- "...G protein-coupled receptor (GPCR) activation 40–43" All these papers are using DREADDs, meaning they are NOT responding a natural stimulus, and indeed, in all of them beyond 90% of the astrocytes express cFos, clearly, abnormal.
- "...behaviourally relevant stimuli 42,44". #42 indeed show that startle response elevates cFos in astrocytes. In #44 I couldn't find any mention of c-fos in astrocytes at all. However, #43, reports no change in cFos following fear conditioning (Figure s4A), and is most relevant to the current study.
- "...astrocyte-neuron signaling 25,26,33". #25 (Sardar et al, Science, 2023) states: "We evaluated Fos expression, a marker of neuronal activity ... and we observed a significant 88.8% increase in neurons, but not in astrocytes", #26 shows RNA seq, #33 has RNAseq and MERFISH data.

So all together, there are only two papers that show protein levels cFos (which is relevant to BAE) following behavior, one shows an increase after startle (#42), and one shows no response to FC (#43). It's ok to give opposite results. It's not ok to give results that seem unified, when in fact they're not.

Minor:

1) scRNA-seq: First, it's for all astrocytes, not just the BAE (why? They should be easy to sort). Second, all astrocytes were divided to 4 clusters, that are NOT different between groups (maybe the NoFC is different, but surely that's not what you're looking for). What am I missing? Yes, cFos and *Adrb1* are expressed in AST3, but in very low levels – so what?

2) Fig 3c-e: the photometry is for all astrocytes, not for BAE. BAE is a small fraction of the astrocytes. Can they affect alone the astrocytic response?

3) Row 275: " There were significant reductions in freezing when mice were treated with Prop before being placed in CtxA (Extended Data Fig. 4b'...)" it is Extended Data Fig. 5b'.

4) Fig 6k – Why cFos is given as "intensity, a.u."? you can count the cells in the amygdala and get non arbitrary units. A bonus, you can compare it to previous studies.

Version 2:

Reviewer comments:

Referee #1

(Remarks to the Author)

The authors have extensively revised their manuscript and added several new and important experimental data sets, including the development of new methods. Their convincing revisions and thoughtful rebuttal fully address my previous comments. In particular, the new experiments using *ibARK2* answer a previously open central question, and the additional analyses of the astrocytic Fos expression/response help to appreciate the new findings and to compare them to related published work. Their efforts have greatly improved the clarity and impact of the study. I have no further comments.

Referee #2

(Remarks to the Author)

Dear Dr. Nagai and cols,

I found the revised manuscript very well presented, with well-designed experiments and straight approaches not easy to performed to address the previous concerns.

The new experiments and analysis makes the study stronger and solid to state the critical role of astrocytes for memory consolidation.

I want to express my congratulations for the excellent work made, including very demanding experiments with new animal models.

I do not have further comments.

Referee #3

(Remarks to the Author)

In my initial review, the main problem I raised, was that despite of the beautiful and varied tools that the authors use, they fail to prove the point that the whole paper relies on -that the cFos+ astrocytes are important to memory. This issue is now solved with the addition of a new experiment using specific silencing of this population (and only it) with *iβARK2*, and the finding that it impairs memory stabilization after reconsolidation.

My major comments were all answered, by further experiment or by correction of textual omissions, satisfactorily.

As far as I am concerned the paper is ready for publication.

A general suggestion for the future that the authors may use or choose to ignore: There's really no need to make the dialogue with the reviewers personal. The reviewers want to improve the paper, and so do you. They are serious people, and so are you. Believe it or not - you are on the same side! It is really unnecessary to "understand and acknowledge the reviewer's frustration" no need to "sincerely apologize for not fully meeting these expectations", or even to write "The reviewer is correct"... Moreover, there is certainly no need to argue on general comments as if they are specific, and write "We would appreciate clarification on which specific publications the reviewer is referring to with this comment". It is strategically not a smart move, because some reviewers may take it personally, and turn against you. Be proud of the gorgeous work that you are doing, and make it the center of the response. Not the reviewers.

From: Jun Nagai <jun.nagai@riken.jp>
Sent: Monday, November 11, 2024 8:42 PM
To: Noah Gray <n.gray@us.nature.com>
Subject: Re: Decision on Nature manuscript 2024-08-17984

Dear Dr. Gray,

Thank you for considering our manuscript and the reviewers' feedback. We are, of course, disappointed with the decision, as we believe our work represents a significant advance in mechanistic understanding of how astrocytic engrams are recruited into engram circuits to stabilize memory. While we feel that our data strongly support the novel induction mechanisms of behaviorally relevant astrocyte ensembles, we appreciate the reviewers' thoughtful and constructive critiques, especially their shared point that we should directly examine the functional roles of astrocytic engrams (Fos+ astrocytes) in memory.

In response, we are excited to inform you that this specific point aligns closely with our own priorities, and we had already initiated experiments to address it. Utilizing a new genetic tool we developed, our latest pilot data on Fos+ astrocyte manipulation fully support the model proposed in our study (please see the attached figure, which I explain in detail below). We have also spent the past several weeks gathering and analyzing both new and existing data. With a detailed revision plan in place, we are confident that we can address most, if not all, of the reviewers' concerns. Key points are outlined below, and we would be pleased to provide our new data or a point-by-point response if that would be helpful for you.

Given our new data and proposed future studies, we hope you might reconsider the rejection decision and allow a formal appeal and resubmission at a future date if we address the reviewers' key issues. If you are open to this, we would also appreciate any guidance on specific areas in the manuscript that might benefit from further development. If it would be easier to discuss our new data and the manuscript's prospects over the phone or Zoom, I am more than willing to arrange a meeting.

Key criticisms and how we will address them:

1. All the reviewers suggested ***specifically perturbing Fos+ astrocyte ensembles to reveal their roles in memory***. To address this, we have utilized a new, improved astrocyte silencing tool, ibARK2. This is a modification of the original ibARK I reported previously (Nagai et al., *Neuron* 2021), which enhances its efficacy via the addition of a membrane-tethering domain, Lck. This improvement localizes ibARK2 near plasma membrane GPCRs and associated Gαq-GTP proteins, which it binds to and sequesters, effectively silencing astrocyte GPCR signaling. We characterized ibARK2 and its mutated negative control, ibARK2mut (**Figure a,b**). We then generated Cre-dependent versions of ibARK2 for use with Fos-Cre TRAP to target behaviorally relevant astrocyte ensembles during fear recall (FR-BAE), directly addressing the point raised by all

the reviewers. Strikingly, our pilot data show that silencing FR-BAE with iβARK2 destabilizes memory upon repeated recall, consistent with our study's findings (**Figure c,d**). We will conduct these experiments with a larger sample and provide the complete dataset. We will also deposit all new plasmids and viruses, adding four more to the eight that we have already made available at Addgene.

- Reviewers 1 and 2 recommended **testing the involvement of Gαq-coupled α1a-adrenoceptors in astrocyte engram induction**. We have already initiated experiments, as this is important and related to point #1, where we observed that silencing Gαq signaling pathway in FR-BAE reduced memory stability. Specifically, we will count Fos+ astrocyte densities while pharmacologically antagonizing α1a-adrenoceptors and see that the receptor signaling contributes to astrocyte Fos induction.
- Reviewer 2 strongly suggested **analyzing the temporal profiles of both neuronal and astrocytic Fos ensembles in the same mice**. Although a comprehensive dataset on neuronal engrams has already been produced by a leading group in neuronal engram research Dr. Susumu Tonegawa's lab (that we used in this study; Roy et al., *Nat Commun* 2022), I agree with the importance of this point, as Reviewer 2 noted, "considering the critical role of the amygdala in fear conditioning (FC), labeling both ensembles and observing dynamic changes over time would provide valuable insights into how memory is stored and recalled by neuron and astrocyte engrams." We have therefore started experiments to address this. We are labeling Fos+ neurons and astrocytes in different colors using dual viral tagging in the amygdala of Fos-Cre TRAP mice during context

exposure, FC, and fear recall. I agree with that this approach offers a "unique opportunity to achieve a milestone in neuron-glia interaction and memory consolidation" (Reviewer 2).

In addition to the points above, we will, of course, strive to address all the reviewers' concerns and provide additional data in our point-by-point response. Here, however, we have highlighted what we consider the most critical issues. Although Reviewer 3's comments were somewhat emotive, raising points that need clarification and questioning the novelty of our work without providing any references, the primary scientific issue centers on the need to **"specifically perturb Fos+ astrocyte ensembles to reveal their roles in memory" (point #1), which we are now confident we can fully address.**

With this new perturbation experiment and the mechanistic data we have gathered—and will continue to add—we believe this study provides robust, direct evidence showing how behaviorally relevant astrocyte ensembles are recruited by neuronal engrams and noradrenaline signals to stabilize memory—a functional role which is distinct from that of neuronal engrams. The challenge in studying astrocytes has been bridging the complex relationship between diverse inputs and their multiple circuit outputs. Our study offers **novel insights, not merely tagging and manipulating astrocyte subpopulations, but linking behaviorally relevant neural signals integrated into astrocytes to specific circuit outputs that modulate behavior.**

Completing the necessary work for this revision will take some time. For this reason, I would like to inquire whether you would be open to the possibility of an appeal. Please let me know your thoughts.

Best,
Jun

Jun Nagai, PhD
Team Leader
Lab for Glia-Neuron Circuit Dynamics
RIKEN Center for Brain Science
Email: jun.nagai@riken.jp
Web: <https://junnnagai-lab.com>

Reviewers' comments (in italic) and our responses (in blue):

We sincerely thank the reviewers for their thoughtful evaluation of our manuscript and greatly appreciate their insightful critiques. We are especially encouraged by their recognition of our approach as “new” and “cutting-edge,” and by their comment that we “directly and elegantly” address a relevant topic in neuroscience. We deeply value this feedback and the opportunity to refine our work in response. At the same time, we acknowledge the important experimental points raised by the reviewers, which we agree have the potential to enhance the quality, depth, and impact of our manuscript.

We have taken these comments seriously and, in response, conducted substantial new experiments and analyses that we believe significantly strengthen our study. We describe these major key additions below and present point-by-point responses to each reviewer comment. New or revised text in the manuscript has been highlighted in blue.

Key criticisms and how we have addressed them:

- 1) All the reviewers suggested specifically perturbing Fos+ astrocyte ensembles to reveal their roles in memory. To address this, we have utilized a new, improved astrocyte silencing tool, i β ARK2. This is a modification of the original i β ARK I reported previously (Nagai et al., *Neuron* 2021¹), which enhances its efficacy via the addition of a membrane-tethering domain, Lck. This improvement localizes i β ARK2 near plasma membrane GPCRs and associated G α q-GTP proteins, which it binds to and sequesters, effectively silencing astrocyte GPCR signalling.

We characterized i β ARK2 and its mutated negative control, i β ARK2mut (Fig. 6a-g). We then generated Cre-dependent versions of i β ARK2 for use with Fos-Cre TRAP to target behaviourally relevant astrocyte ensembles during fear recall (FR-BAE), directly addressing the point raised by all the reviewers. Strikingly, our data show that silencing FR-BAE with i β ARK2 destabilizes memory upon repeated recall, directly supporting our conclusions (Fig. 6h-k). We have deposited all new plasmids and made available at Addgene (Supplementary Table 4).

- 2) Reviewers 1 and 2 emphasized the importance of testing the involvement of G α q-coupled α 1A-adrenoceptors in astrocyte ensemble induction. This is important and closely related to point #1, as i β ARK2 acts on G α q signalling. We performed a detailed series of new experiments (Fig. 5) to track expression changes in G α q-coupled α 1A and G α s-coupled β 1 adrenoceptors (the two dominant adrenoceptors in amygdalar astrocytes) over days, assess their correlation with FR-BAE inducibility, and test their causal roles via astrocyte-specific conditional knockout.

Importantly, we found that co-expression of these adrenoceptors peaked at 1-day post-conditioning and gradually declined over the following days, closely mirroring the inducibility window of FR-BAE which also peaks during recall at

1-day post conditioning. Astrocyte-specific deletion of *Adra1a* or *Adrb1* abolished FR-BAE formation, directly linking receptor expression to ensemble induction. These findings clarify the molecular basis for noradrenergic gating of astrocytic memory functions and support our central hypothesis: astrocyte ensembles serve as multi-day traces—biochemical states initiated by learning that persist over days to selectively stabilize subsequent, repeated experiences into lasting memory.

- 3) Reviewer 2 strongly encouraged analyzing the temporal profiles of both neuronal and astrocytic Fos ensembles in the same mice. First, we note that Reviewers 1 and 3 also commented on ambiguity in our method for neuronal Fos counts, and we acknowledge that our original description lacked sufficient clarity regarding how the publicly available data were reanalyzed and compared. We have now revised the Methods to explicitly detail our reanalysis approach. Briefly, we reanalyzed a dataset from Dr. Susumu Tonegawa's lab (Roy et al., Nat. Commun., 2022 ²), which comprehensively profiled neuronal engrams. That said, we now agree that this point warrants direct experimental validation.

To address this, we added new experiments using dual-color Fos tagging in Fos-iCreERT2 TRAP2 mice, enabling simultaneous labeling of both neuronal and astrocytic ensembles in the amygdala (Fig. 1i-k). We agree with that this approach offers a "unique opportunity to achieve a milestone in neuron-glia interaction and memory consolidation" (Reviewer 2).

In addition to the major points above, we have addressed all other reviewer concerns and provided additional data and clarifications in our detailed responses. While some reviewers initially questioned the novelty or clarity of our work, we believe the revised manuscript more clearly articulates our conceptual advances and methodological rigor. We especially note that the core scientific issue across all reviews was the need to "specifically perturb Fos+ astrocyte ensembles to reveal their roles in memory" (point #1 above)—new experimental data we have now fully addressed.

With this new perturbation and mechanistic data we have gathered, we believe this study provides robust, direct evidence showing how behaviourally relevant astrocyte ensembles are recruited by neuronal engrams and noradrenaline signals to stabilize memory—a functional role which is distinct from that of neuronal engrams. The challenge in studying astrocytes has been bridging the complex relationship between diverse inputs and their multiple circuit outputs. Our study offers novel insights, not merely tagging and manipulating astrocyte subpopulations, but linking behaviourally relevant neural signals integrated into astrocytes to specific circuit outputs that modulate behaviour.

Reviewer #1

(Remarks to the Author): Dewa and colleagues have investigated the role of the immediate early gene Fos in astrocytes in contextual fear conditioning. Using a new approach for brain wide tagging of c-Fos expressing astrocytes the authors report particularly high increases of such astrocytes in the amygdala during fear recall (FR). They also found that noradrenaline (NA) can induce c-Fos expression in astrocytes and that fear conditioning (FC) and FR differentially affect NA signals and astrocytic Ca²⁺ and cAMP levels in the amygdala. Lowering of NA signalling and of the activity of amygdalar neurons reduced the number of c-Fos positive astrocytes. Furthermore, FC and FR affected the expression profiles of astrocytes, including the expression of beta1-adrenoreceptors. Overexpressing the latter in astrocytes was sufficient to increase the number of tagged astrocytes, to increase and generalize fear behaviour and to increase neuronal c-Fos expression in the amygdala. These experiments directly and elegantly demonstrate that contextual fear conditioning profoundly alters expression profiles of astrocytes and the involvement of NA in signalling in that. They also reveal that astrocytic manipulation can increase freezing behaviour. (underlining by us)

Thank you for these supportive comments on the tools we made and our findings.

Comment #1-1: *Throughout the manuscript it is suggested that the c-Fos positive/tagged astrocyte have an important and specific role but this is not directly addressed. It seems that the author did not use any astrocytic manipulation that is specific to the c-Fos positive/tagged astrocytes, which would be necessary to demonstrate such a role. For instance, what happens to freezing behaviour when only tagged/c-Fos expressing astrocytes are inhibited or when beta1-adrenoreceptors are removed from those selectively? The effect of overexpression of beta1-adrenoreceptors on memory are very interesting but conceptually similar findings of memory improvement via astrocytes have been reported before (Adamsky et al., 2018; Iwai et al., 2021).*

We sincerely thank the reviewer for their thoughtful evaluation of our findings. The first question raised is highly relevant, as it is a concern shared by all reviewers: directly testing the functional roles of Fos+ astrocytes induced by fear recall (FR-BAE) through population-specific manipulation. This aligns closely with our research priorities, and we have now included new data demonstrating that selective silencing of FR-BAE disrupts memory stabilization upon repeated recall, directly supporting the central claim of our study. To address this, we utilized a new, improved astrocyte silencing tool, iβARK2—a modified version of the original iβARK we reported previously (Nagai et al., *Neuron*, 2021¹). This improved construct includes a membrane-tethering Lck domain that enhances its efficacy by localizing iβARK2 near plasma membrane GPCRs and associated Gαq-GTP proteins, where it binds and sequesters these targets, effectively silencing astrocytic GPCR signalling. We characterized iβARK2 alongside its mutated negative control, iβARK2mut, and confirmed robust membrane-localized expression across astrocytic territories. Functional assays in acute amygdala slices showed that iβARK2 significantly suppresses phenylephrine-evoked Gq-GPCR calcium responses with greater efficacy than the original iβARK.

Fig. 6a-g

Using a Cre-dependent system in Fos-iCreER^{T2} TRAP2 mice, we next expressed iβARK2 specifically in FR-BAE and found that this manipulation impairs memory stabilization, as evidenced by reduced freezing upon repeated recall compared to controls. Furthermore, levels of Igfbp2—a neuromodulatory factor secreted by astrocytes and induced after recall to support memory stabilization—were significantly reduced in iβARK2-expressing astrocytes. These findings indicate that iβARK2 impairs astrocyte-dependent memory support by disrupting downstream effector expression and establish a causal role for Fos⁺ astrocyte ensembles in maintaining fear memory through Igfbp2-mediated mechanisms. All new characterization and behavioural data have been incorporated into the revised manuscript (Fig. 6), and the newly developed plasmids and viruses have been deposited to Addgene (Supplementary Table 4).

Fig. 6h-k

Further, we respectfully disagree with the reviewer's assertion that our finding—that reinforcing noradrenaline signalling specifically in amygdalar astrocytes promotes memory stabilization—is conceptually similar to the findings reported by Adamsky et al. and Iwai et al.

First, the studies by Adamsky et al. and Iwai et al. showed that chemogenetic or optogenetic stimulation of astrocytes during conditioning enhanced memory acquisition. In contrast, we demonstrated that *Adrb1* overexpression in amygdalar astrocytes enhanced memory stabilization during memory reconsolidation following, not during, the 1st recall. Notably, data presented in our original manuscript (now shown in Extended Data Fig. 12i,j), along with new results (Extended Data Fig. 12c,d), demonstrate that *Adrb1*-OE mice did not exhibit enhanced memory acquisition or increased freezing during the first recall. However, these mice did show significantly increased freezing during the 2nd recall, consistent with enhanced memory stabilization after reconsolidation.

Extended Fig. 12 c-k

Supporting this, our new experiment using direct silencing of FR-BAE with β ARK2 (Fig. 6j) revealed that FR-BAE β ARK2 mice displayed no change in freezing at the 1st recall with β ARK2 expression but showed reduced freezing at the following 2nd recall—indicating that memory was destabilized during reconsolidation in the presence of β ARK2. Together, these findings strongly support the interpretation that FR-BAE contribute specifically to memory stabilization, rather than initial acquisition. This distinction is central to our study and underscores a novel astrocyte function in memory reconsolidation.

Second, Adamsky et al. manipulated astrocytes in the hippocampal CA1 region, while Iwai et al. targeted the anterior cortex (though their actuators were expressed brain-wide, and they illuminated anterior cortical regions, making it challenging to pinpoint exact loci). In contrast, our manipulations were specifically localized to the amygdala. As these regions serve distinct functions in memory, this likely

explains the differing and, importantly, complementary findings between their studies and ours.

Third, while the studies by Adamsky et al. and Iwai et al. involved manipulation of the astrocytic Gq-GPCR pathway, they did not address the physiological relevance of this signalling in memory processing. This limitation is not unique to their work, as the majority of astrocyte manipulation studies published to date lack physiological context for the signalling mechanisms involved (as discussed in Nagai et al., *Neuron*, 2021³). In contrast, our study (as generously noted by the reviewer) systematically characterizes the behavioural relevance of neural signals that recruit Fos⁺ astrocytes in the amygdala. Using screening approaches and *in vivo* gain- and loss-of-function experiments (Fig. 2-5, including new data), we show that Fos expression in astrocytes requires convergent input from engram neurons and noradrenaline neuromodulatory signals. This induction depends on multiple GPCRs, including *Adra1a* and *Adrb1*, for which we now present direct functional evidence with new data (Fig. 5; see our responses to your Comment #1-7 for further details).

Therefore, we respectfully disagree with the reviewer's comment that our findings lack conceptual novelty. While the studies by Adamsky et al. and Iwai et al. are indeed landmark papers utilizing chemo- and optogenetic manipulation of astrocytes in memory tasks, we believe our study offers a unique and novel mechanistic link between memory-relevant neural cues integrated into astrocytes to specific circuit outputs that stabilizes memory.

Comment #1-2: *The assay for brain-wide tagging of c-Fos expressing astrocytes in behavioural experiments is novel and interesting, but there are some open questions. The authors compare their results on astrocytes to the well-known neuronal c-Fos expression increase in contextual fear conditioning, which is important because the lack of astrocytic c-Fos upregulation after FC is interesting and some reference is helpful. However, (1) it is not clear how exactly neuronal c-Fos expression in Fig. 1e-f was determined and quantified. Also, (2) did the authors compare their findings with tagging astrocytes systematically with immunohistochemistry? (3) Is it possibly that the tagging significantly underestimates the number of c-Fos expressing astrocytes because the number of cells in the illustration (e.g. Fig. 1b, 4) seem relatively low given the density of astrocytes in the tissue and the volume of the brain regions? (Numbers 1, 2 and 3 were inserted by us to track each specific point raised.)*

First of all, thank you again for the supportive comments on the tools we developed.

Regarding the first question about how we obtained data on Fos⁺ neurons, we apologize for the lack of clarity in our analysis. Both our study and Roy et al. (*Nat. Commun.*, 2022)² used TRAP2 (Fos-CreERT2) mice combined with Cre-dependent fluorescent reporters (tdTomato in neurons by Roy et al., and mNeonGreen in astrocytes in our study) to perform Fos-tagging and single-cell resolution whole-brain imaging, followed by cell counting in anatomically registered brain regions using the Allen Brain Atlas. This enabled direct region-by-region comparisons

across the two datasets. Roy et al. analyzed 247 brain regions and defined 117 "engram significant regions," where Fos⁺ neuronal counts were significantly increased in both FC and FR conditions relative to non-foot-shocked controls. We extracted neuronal Fos⁺ fold changes upon FC or FR relative to non-shocked controls from a publicly available supplementary table in Roy et al., which provided region-level Fos⁺ neuron counts across all three groups. In our study, we initially quantified Fos-tagged cells across all 839 regions in the Allen Brain Atlas, and then selected putative 75 "engram significant regions" (64.1% of their total 117 engram significant regions) for correlation analysis. Selection criteria were as follows: we excluded regions (as we have described in the main text) containing Fos-tagged mNeonGreen-positive neuron-like cells, which were observed near the ventricles and midline (areas consistent with known GFAP-positive neural progenitor zones^{4,5}) to ensure accurate counting of bona fide astrocytes. We also excluded high-level hierarchical regions (e.g., the Midbrain, Thalamus) that had extremely high cell counts and could act as outliers in correlation analysis. This yielded 75 engram significant regions for correlation analysis in Fig. 1. We have made a separate Methods section "Correlational analysis of Fos⁺ cell counts" to provide a detailed explanation of this methodology.

For the second question regarding the analysis of Fos-tagged versus c-Fos⁺ astrocytes, we thank the reviewer for raising this excellent point. In response, we performed systematic immunohistochemical analysis of c-Fos⁺ astrocytes across multiple timepoints (Extended Data Fig. 4): NoFC, NoFR, 1.5 h and 6 h after FC, and 1.5 h and 6 h after FR. In agreement with our Fos-tagging results, we observed no significant c-Fos induction in astrocytes following FC or in the NoFR group. In contrast, FR led to a robust increase in astrocytic c-Fos expression at 1.5 h (~4-fold increase; $p < 0.01$ vs NoFC, 1.5 h/6 h post-FC, and NoFR), which gradually declined but remained elevated at 6 h (~2-fold vs controls). These data confirm that astrocyte Fos-tagging reflects recall-induced transcriptional activation, and the temporal profile of c-Fos protein supports its use as a validation marker for BAE. We appreciate this insightful guidance.

Extended Data Fig. 4

Regarding the third point, we completely agree with the reviewer that this is an important limitation. More broadly, it reflects a common technical challenge when using a single genetic entry point, which may underestimate the population of cells, including astrocytes, and perhaps neurons, engaged by a given behaviour. This limitation has also been noted in recent studies, such as Williamson et al., *Nature* 2025 ⁶.

This was not directly asked, but we appreciate the reviewer's interest in our data—specifically the observation that “the lack of astrocytic c-Fos upregulation after FC is interesting.” You may wonder about the Williamson et al., *Nature* 2025 after your initial review, which appears to report conflicting findings: namely, an increase in astrocytic Fos expression in the hippocampal CA1 region following FC, whereas we observed no such increase. We were equally intrigued by this discrepancy and conducted a new set of experiments to test a hypothesis that emerged from comparing their behavioural protocol with ours.

Specifically, we suspected that contextual novelty—present in their paradigm but minimized in ours—could be a key driver of astrocytic Fos induction. In our FC paradigm, we included 15-minute habituation to the conditioning chamber for 3 consecutive days, following the protocol used in original TRAP2 paper (DeNardo et al. *Nat Neurosci* 2019 ⁷), to minimize contextual novelty-driven responses and better isolate memory-related signals. In contrast, Williamson et al. did not include habituation. Notably, in the published peer review file accompanying their paper, one reviewer raised this issue explicitly:

“Comment 1-2: The authors should compare between the with and without shock groups, not the home-cage and fear-conditioned groups. The mere exposure to a new space could be enough to increase hippocampal activity... Since electric shock is essential for learning, this comparison (with vs. without shock) should be applied throughout.”

This point was not directly addressed in the final version of their paper.

To test our hypothesis, we repeated the FC protocol without habituation—what we term “FC+Novelty”—and performed brain-wide astrocytic Fos tagging using TRAP2 mice and serial two-photon tomography. As shown in the revised manuscript (Extended Data Fig. 6), FC+Novelty induced a robust increase in astrocytic Fos+ cells across multiple regions, including primary sensory and prefrontal cortices, nucleus accumbens, and amygdala. Importantly, when we extracted Fos+ astrocyte counts in CA1, we found a significant increase in the FC+Novelty group compared to both the NoFC and FC groups (3.2-fold vs. NoFC, $p < 0.05$; 4.0-fold vs. FC, $p < 0.05$), closely resembling the CA1 pattern reported by Williamson et al.

Extended Data Fig. 6

These findings confirm that our tagging method is highly sensitive to astrocytic Fos responses and that the discrepancy with Williamson et al. stems from biological differences in behavioural protocols, not technical limitations. This new dataset emphasizes the importance of separating contextual novelty from associative learning when interpreting astrocyte ensemble dynamics.

Ultimately, while others have focused on the presence or absence of Fos+ astrocyte populations and tagging and manipulating subpopulations, our central aim goes beyond that. This study advances a more fundamental question: what specific, behaviourally relevant neural signals are integrated by astrocytes to shape circuit-specific outputs that modulate behaviour? We believe this mechanistic perspective is key to understanding the adaptive roles of astrocytes in circuit computation related to memory, which is uncovered in the current study in the context of fear conditioning learning and memory.

Comment #1-3: *A related question in about Fig. 5b. There, the percentage of Fos-positive amygdalar astrocytes is about the same in the FC and FR condition and Fos expression levels are higher in FR compared to FC. However, in Fig. 1d there is a large difference (> 30 fold) in c-Fos expressing/tagged astrocytes between FC and FR. It is therefore currently unclear how variations in c-Fos expression in astrocytes are captured by tagging and counting.*

This is an important issue that the reviewer and Reviewer#2 raise (Comment #2-3). We agree it deserves clarification. The apparent discrepancy stems from methodological differences: the Fos-TRAP system labels astrocytes that express Fos cumulatively over several hours after 4-OHT injection (Extended Data Fig. 3, 5), whereas scRNA-seq captures a snapshot at a single time point (1.5 h after behaviour). This likely explains why TRAP reveals a >30-fold increase in Fos+ astrocyte density during FR versus FC, while transcriptomic analysis shows only a ~2-fold elevation in Fos expression. To clarify this, we have included the following sentence in lines 368–372:

It is worth noting that scRNA-seq provides a snapshot of gene expression at a single time point, whereas the Fos-TRAP system integrates Fos promoter activity over several hours, potentially explaining the greater fold differences observed with Fos tagging (>~30-fold increase in FR vs FC in tagged astrocytes; Fig. 1d–e).

To clarify further, we observed no significant c-Fos protein induction in astrocytes after FC or NoFR, whereas FR robustly induced astrocytic c-Fos protein at 1.5 h, with a slower decay observed at 6 h. Consistent with this time course, TRAP-tagging was effective when 4-OHT was administered immediately after or 3 h post-recall, indicating a biologically plausible window of 0–3 h for astrocytic Fos activation during FR (Extended Data Fig. 5), which is now clearly described in the text (lines 192-200).

We also emphasize that transcriptomics is inherently limited to a momentary snapshot and may underestimate dynamics of gene expression. Our primary goal in Fig. 5 and Extended Data Fig. 11 was not to quantify all Fos+ astrocytes, but to identify molecular correlates of FR-BAE. Adrenoceptor upregulation identified by scRNA-seq was carefully validated by RNAscope and functionally confirmed to be necessary for FR-BAE induction (Fig. 5). Additionally, this approach led us to identify Igfbp2 as a key downstream target. We validated its astrocyte-specific

expression at single-cell resolution and demonstrated its causal role in memory stabilization (Extended Data Fig. 11). Notably, silencing FR-BAE reduced *Igfbp2* and destabilized memory after reconsolidation (Fig. 6), whereas astrocytic *Adrb1* overexpression enhanced FR-BAE and *Igfbp2* expression, leading to memory over-stabilization (Fig. 7).

Collectively, scRNA-seq served as a hypothesis-generating tool that successfully guided our experiments. More importantly, our findings were rigorously validated both transcriptionally and functionally *in vivo*.

Comment #1-4: *In the imaging experiments in Fig. 3d and Ext. Fig. 5e there is variable and sometimes rapid and strong increase in the reported cAMP levels already in the home cage, which casts doubt on the reliability of the recordings and their analysis.*

The reviewer raises an important question, and we appreciate the opportunity to clarify this point and elaborate on our interpretation of these observations regarding the increase in astrocyte cAMP signals in the amygdala not only after exposure to the foot-shock chamber but also before the mouse is placed into it. This is an important point that merits discussion.

The pre-chamber exposure period involves unavoidable experimental steps, including moving the mouse from its housing isolator to the experimental room and handling the mouse to gently attach the recording cable to the amygdala (~4 mm deep), a process that can take over a minute. While we habituate animals to these procedures over several days prior to fear conditioning to minimize handling stress or novelty, such careful habituation may itself contribute to the animals associating our handling as part of the task-specific context.

Of note, this ramping of signals occurs only after the mouse has undergone fear conditioning, as seen on the fear recall (FR) day (Extended Data Fig. 5e). Furthermore, during the context B session (the day after FR), similar ramping signals are observed in pre-chamber period (Extended Data Fig. 5e), further supporting the interpretation that this activity reflects task-specific anticipatory processing, which has been suggested in rodents^{8–11}. Importantly, although ramping signals occur before context B exposure, they dissipate after a few minutes of entering context B (Extended Data Fig. 5e), in contrast to their sustained elevation during FR in context A. Critically, propranolol treatment abolished this ramping response (Fig. 4d, +Prop), further supporting the conclusion that the signal reflects noradrenergic modulation associated with fear memory recall.

Given the reasons above we are fully confident in the reliability of our data and analysis. All recording data and analysis codes have been deposited (https://neurodata.riken.jp/r/Nagai/KD_KK_et_al_2024/, password: cbss602) and are publicly available on GitHub (<https://github.com/Jun-Nagai-Lab/Dewa-et-al>), allowing anyone to reproduce our findings using the same methods. We thank the reviewer for raising this point, and in the revised manuscript, we will explicitly discuss this point.

Comment #1-5: The quantification of freezing in Ext. Fig. 5b reveals that in the control context B (rightmost panel), animals freeze almost as much as during fear-conditioning (leftmost panel). What is the explanation for this unexpected behaviour? A similar illustration/quantification of freezing behaviour should accompany other key findings related to contextual fear conditioning (e.g. Fig. 1, Fig. 6).

We thank the reviewer for their careful evaluation and thoughtful suggestions regarding data representation.

Regarding the first point, this behavioral dataset was collected during fiber photometry recordings, which involve procedures that can subtly influence behavior. Although mice were well habituated to tethering across multiple days, the combination of fiber attachment and entry into a novel environment (context B) following fear learning may plausibly induce a modulated state vigilance, anticipation, or a mild stress-like response. Such behavioral modulation is not unexpected and reflects the complex nature of exploratory behavior, which is shaped by multiple interacting factors and may extend beyond the primary scope of our study. Importantly, closer inspection of the initial three minutes of the context B session—a time window commonly used in prior studies for assessing freezing rate in context B—reveals a relatively low freezing rate (~20%). This observation aligns well with standard behavioral benchmarks and is consistent with other datasets acquired under comparable conditions, including the rightmost panel in Fig. 7h and Extended Data Fig. 12k. Together, these findings suggest that the behavior observed falls within a physiologically reasonable range and supports the overall validity of our conclusions.

For the second point, we appreciate the reviewer’s suggestion. We have now analyzed the freezing data for Extended Data Fig. 12 (previously shown above in response to Comment #1-1) and Fig. 1b.

Fig. 1b

Comment #1-6: The method and results for silencing of amygdalar ‘engram’ neurons and their counting is barely documented, which makes it hard to appreciate the experiment.

We appreciate the reviewer's suggestion and have now included additional methodological and quantitative details in the revised manuscript. As shown in Extended Data Fig. 9, we found that $16.1 \pm 2.4\%$ of amygdala neurons were tagged using the RAM-based engram labeling method and targeted for silencing via hM4Di expression. To evaluate whether this silencing approach effectively reduced engram reactivation, we quantified the fraction of c-Fos-positive neurons 90 minutes after fear recall. We observed a significant reduction in reactivation: the c-Fos-positive fraction within RAM-tagged engram neurons decreased from $42.3 \pm 2.3\%$ in mKate2-expressing controls to $2.3 \pm 1.2\%$ in the hM4Di group. Similarly, the overall fraction of c-Fos-positive neurons in the amygdala dropped from $16.0 \pm 1.0\%$ to $3.3 \pm 0.7\%$. These data confirm the efficacy of our chemogenetic silencing approach.

Extended Data Fig. 9

Comment #1-7: The authors show that NA increases c-Fos expression by cultured cortical astrocytes through convergent signalling of beta-adrenoreceptors via Gs and mGluR3. Astrocytes can substantially change their expression profiles in culture, as acknowledged by the authors. Therefore, it is uncertain whether NA signals in a similar manner in situ. In vivo and in more intact preparations, there is plenty of experimental evidence that Gq-coupled alpha1-adrenoreceptor play an important role and the authors demonstrate in their current study that Gq-activation in astrocytes is sufficient to increase

the number *c-Fos* expressing/tagged astrocytes (Ext. Fig. 2). If the authors wish to demonstrate a role of beta1-adrenoreceptors in astrocytes they could, for instance, combine the experimental approach shown in Fig. 2l-m with manipulation/inhibition of beta-adrenoreceptors and Gs-signalling selectively in astrocytes *in situ*.

We thank the reviewer and Reviewer #2 (comment #2-3) for raising this fundamentally important point, which we acknowledge was not adequately addressed in the original manuscript. Below, we clarify the conceptual and experimental context in three parts:

(1) Our pharmacological screen (Fig. 2) was performed in primary cultured astrocytes, which are known to undergo significant molecular and functional changes—including downregulation of GPCRs such as Gi-coupled *Grm3* and, importantly, Gq-coupled *Adra1a* as well (Sardar et al., IJMS 2021¹²). We now explicitly discuss this caveat in the revised manuscript (lines 406-409) and complement it with new *in vivo* data (Fig. 5e,f).

(2) While our *in vitro* data implicated β 1 adrenergic (Gs-coupled) signalling in Fos induction, our working model *in vivo* (as described in our original submission) posits that α 1A-adrenergic (Gq-coupled) signalling primes astrocytes by activating PLC, which in turn enables calcium responses through mGluR3–Gi signalling (Fig. 2h). We proposed this model based on the well-established absence of Gq-coupled glutamate receptors (such as mGluR5) in mature astrocytes (Sun et al., Science 2013¹³). However, this priming mechanism was not directly tested *in vivo* at the time of initial submission.

(3) As the reviewer noted, Extended Data Fig. 2 shows that astrocytic Gq activation via DREADDs is sufficient to induce Fos expression. However, this artificial activation does not directly establish behavioural relevance.

To address this, we conducted new *in vivo* experiments involving astrocyte-specific knockdown of either *Adra1a* or *Adrb1* in the amygdala, using floxed alleles, AAV2/5-*GfaABC1D*-Cre-4x6T, and newly developed Flp-based Fos-tagging AAVs (AAV2/5-Fos-FloER^{T2} and AAV2/5-*GfaABC1D*-fDIO-mNeonGreen; Supplementary Table 4). These experiments demonstrated that knockdown of either receptor significantly reduced FR-BAE density (Fig. 5l-p), validating the *in vivo* roles of both Gq- and Gs-coupled receptors.

Fig. 5I-p

Furthermore, in another newly added data (Fig. 6), we show that selective inhibition of Gq signalling in FR-BAE astrocytes using iβARK2 impairs memory stabilization—thus directly demonstrating a causal role for astrocytic Gq signalling in both Fos induction and memory function.

Together, these results address the reviewers' concerns and support our model that astrocytes integrate multiple neuromodulatory GPCR signals—including both Gq and Gs pathways—to induce FR-BAE formation and enable memory stabilization.

Comment #1-8: A minor comment regarding the term 'astrocytic engram', which is used throughout the manuscript: By many definitions, engrams are the physical/chemical changes in brain elicited by learning (e.g. representing a memory) and the engram cells are those that undergo these changes. At the moment, the study indicates that astrocytes can facilitate memory formation/recall but whether they encode the memory (e.g. a specific context) has not been addressed.

We appreciate the reviewer's thoughtful comment. Based on current experimental definitions (e.g., Josselyn and Tonegawa *Science*, 2020¹⁴), we believe that the Fos+ astrocyte ensembles in the LA/B identified in our study meet key criteria for engram cells: they are recruited during both encoding and recall, undergo lasting molecular changes (such as adrenoceptor upregulation and Fos/Igfbp2 induction), and contribute causally to memory retrieval when perturbed. These are measurable

and testable features within the limits of current methodology, and our findings support their fulfillment.

That said, we fully agree that astrocytes do not encode experience-specific content in the same manner as neurons, as we discussed in the main text (lines 526-531).

our findings suggest that the astrocytic ensemble may not serve as a direct memory trace encoding experience-specific information like neurons because the perturbation of astrocytic engram did not alter initial fear recall. Instead, it likely functions as an eligibility trace for memory stabilization over days, creating a permissive and specialized microenvironment within circuits that supports the stabilization of memories related to repeated experiences.

This raises an important question: what level of informational granularity is necessary for a cell population to be classified as an engram? We raise this not as a point of contention, but as a constructive prompt, especially as the field begins to explore causal memory-related roles for non-neuronal cell types. In light of this discussion, and to maintain clarity, we are open to softening our terminology and referring to these populations as “astrocyte ensembles” in key instances throughout the manuscript.

Reviewer #2

In this study, Dewa et al investigate a relevant topic in neuroscience, the role of astrocytes in memory stabilization. By using cutting-edge approaches based on brain-wide Fos tagging and imaging method, authors have shown brain-wide behaviourally relevant astrocyte ensembles (BAE) recruited by emotional experiences (fear conditioning), being enriched in regions with neuronal engrams previously described. The further characterization of those BAE showed the role of NA and Igfbp2 signaling, and the pharmacological and genetic perturbation of these signaling pathways modulate memory stability and precision. Showing the astrocyte memory engrams is an important finding using the novel methodology, but to have a comprehensive idea of the magnitude and relevance it is needed to have the neuronal engrams too. (underlining by us)

Thank you for your encouraging comments on the tools we developed and the findings we presented.

Comment #2-1: *Although based on published database (Roy DS et al Nat Commun 2022), the fos expression in neurons is not performed in this study. The analysis of the simultaneous temporal profile following dual viral approach to tag both cellular engrams is strongly required; without it, the study gets incomplete and miss the unique opportunity to bring a milestone in the fields of neuron-glia interaction and memory consolidation. Considering the critical role of amygdala in fear conditioning (FC), the labeling of both ensembles and the dynamic changes over time would show outstanding information to understand how memory is stored and recall by both neuron and astrocyte engrams.*

We sincerely thank the reviewer for emphasizing this important point. As the reviewer rightly notes, simultaneous labeling of both neuronal and astrocytic Fos ensembles would offer key insights into neuron–glia coordination during memory processing. While Roy et al. (Nat Commun 2022 ²) have already provided a comprehensive time-course dataset of neuronal Fos expression after FC and FR, we recognized the unique opportunity to complement this work and strengthen our conclusions.

To directly address the reviewer’s suggestion, we performed dual-color Fos tagging experiments in the amygdala of TRAP2 mice. We injected a neuron-specific Cre-dependent mCherry virus (*hSyn1-DIO-mCherry*) and an astrocyte-specific Cre-dependent mNeonGreen virus (*GfaABC₁D-DIO-mNG*) into the same animals, and prepared four experimental groups: NoFC, FC, NoFR, and FR. This approach enabled simultaneous visualization of Fos+ neuronal and astrocytic ensembles in the same brain region.

We found that neuronal tagging (mCherry+, neuronal engram) increased after both FC and FR, consistent with previous reports. In contrast, astrocytic tagging (mNG+ astrocyte ensemble) was strongly induced only after FR, closely mirroring our brain-wide results (Fig. 1a-h). This new dataset that is now included in the revised manuscript (Fig. 1i–k) reinforces our conclusion that astrocytic Fos ensembles in the amygdala are preferentially engaged during memory recall rather than

conditioning, and display distinct temporal dynamics compared to neuronal engrams.

Fig. 1i,j,k

Comment #2-2: Additionally, the study shows important gaps between different experimental dataset: The body of the work is performed following the strategy described in Roy et al Nat Commun 2022, that is, one week later of i.p. 4-OHT brains are analyzed and distribution of fos expression detected in BAE. However, authors study the time window of astrocyte Fos tagging by analyzing visual responses and found that the majority of tagging in visual cortex occurred within a six-hour window centered around the time of 4-OHT injection (Extended Data Fig. 3c,d). Why this change in the behavioural paradigm? The BAE in these cases are completely different. It is hard to understand the relevance for the study if Fos tagging is not analyzed by FC following this temporal scheme.

Thank you for raising this excellent point. In retrospect, the characterization of the tagging time window was conducted during the initial phase of our project using a visual stimulation protocol identical to that described in the original Fos-TRAP mouse paper (DeNardo et al., *Nature Neuroscience* 2019 ⁷). This served as a necessary positive control experiment to verify that the tagging system was functioning correctly, as we initially did not know whether the new system would work or not. Chronologically, this means we did not “change” the behavioural paradigm but rather started with this control.

Nonetheless, we fully agree with the reviewer that BAEs may differ between distinct behaviours. To address the reviewer’s concern directly, we have now experimentally assessed the tagging time window during fear recall, collecting new data and reporting these results in the revised manuscript (see Extended Data Fig. 5). Our data show that FR-BAE density is significantly elevated when 4-OHT is administered either immediately after fear recall (0 h)—the timing used throughout our manuscript, or 3 hours later.

Extended Data Fig. 5

This time course aligns with the observed dynamics of astrocytic c-Fos protein expression: c-Fos levels peak at approximately 1.5 h post-recall and gradually decline by 6 h (Extended Data Fig. 4). These results suggest a biologically plausible tagging window of 0–3 hours following fear recall (Extended Data Fig. 5). Please see the relevant text in lines 192-200.

Extended Data Fig. 4

Comment #2-3: It is found that visual stimuli can induce Fos expression in astrocytes but barely by FC. It is intriguing that such strong stimuli, foot shock, that boost the neuronal activity in different brain areas (Roy DS et al Nat Commun 2022) do not induce significant changes in astrocytes at the level of Fos expression (Fig 1), but they can respond reliably with cAMP and Ca signals to NA signaling (Fig 3d,e). Then, Fig 5b shows that amygdalar astrocytes show enhances fos expression after FC, which it seems significant vs NoFC. The message of the study is confusing.

Even more, the expression of *Adra1a* is extremely high after NoFR and FR, but there is no mention to such results in the discussion. There is a progressive enhancement of expression in the experimental groups analyzed. Then, how is possible that enhanced *Adra1a* expression in amygdalar astrocytes, significantly higher than *Adrb1* in all conditions, has been ignored in this study?

This is an important issue raised by the reviewer regarding Fig. 5b and shared with the Reviewer #1 (Comment #1-3), which we agree it deserves clarification. The apparent discrepancy stems from methodological differences: the Fos-TRAP system labels astrocytes that express Fos cumulatively over several hours after 4-OHT injection (detailed above Comment #2-2, Extended Data Fig. 5, supported by Extended Data Fig. 4), whereas scRNA-seq captures a snapshot at a single time point (1.5 h after behaviour). This likely explains why TRAP reveals a >30-fold increase in Fos+ astrocyte density during FR versus FC, while transcriptomic analysis shows only a ~2-fold elevation in Fos expression. To clarify this, we have included the following sentence in lines 368–372:

It is worth noting that scRNA-seq provides a snapshot of gene expression at a single time point, whereas the Fos-TRAP system integrates Fos promoter activity over several hours, potentially explaining the greater fold differences observed with Fos tagging (>~30-fold increase in FR vs FC in tagged astrocytes; Fig. 1d–e).

We now elaborate on the broader conceptual point raised in Comment #2-3. The apparent paradox - that FC, a strong sensory stimulus, elicits cAMP and calcium responses and yet minimal astrocytic Fos, whereas FR induces robust astrocyte Fos - is a central finding of our study, and is mechanistically resolved in Fig. 5 with new data.

In brief, we propose that amygdalar astrocytes do not respond immediately to foot shock with Fos expression, but are primed over time by FC through progressive upregulation of adrenergic receptors (*Adra1a* and *Adrb1*). This multiday adrenoceptor remodeling enables astrocytes to respond more robustly during FR, but not during the initial FC when receptor levels remain low.

Supporting this:

In Fig. 3d, we show marked "ramping up" of astrocyte cAMP during FR at 24 h post-FC, which is abolished by the β -receptor antagonist propranolol—mirroring a reduction in FR-induced astrocyte Fos (Fig. 4a).

In new data (Fig. 5d–f), RNAscope reveals that *Adrb1* and *Adra1a* mRNA levels increase progressively over days post-FC, peaking at 1 day, forming a molecular trace that is absent at the time of conditioning.

Fig. 5d–f

In Fig. 5g–h, we show that astrocyte Fos inducibility upon FR peaks at 1 day post-FC, coinciding with maximal *Adra1a*–*Adrb1* co-expression, and declines at later time points as receptor levels fall. This temporal alignment suggests that astrocytic Fos induction during FR depends on prior adrenoceptor upregulation—a form of experience-dependent tuning rather than a mere reflection of stimulus intensity.

Fig. 5g-k

In Fig. 5l-p, we establish causality: conditional knockout of *Adra1a* or *Adrb1* in astrocytes abolishes Fos induction upon FR, confirming that both receptors are necessary.

Fig. 5l-p

Although FC is behaviourally salient and induces strong neuronal activity, amygdalar astrocytes may not interpret foot shock alone as sufficient input to trigger Fos due to low baseline expression of adrenergic receptors. In contrast, FR represents a repeated experience occurring after prior learning and in a neuromodulatory context rich in NA—conditions to which astrocytes have become transcriptionally sensitized. This explains why FR, but not FC, elicits robust astrocyte Fos.

Taken together, these results suggest that astrocytic ensemble formation is not a passive reflection of stimulus strength, but a gated, experience-dependent response shaped by slow, multiday transcriptional remodeling. This is a central conceptual advance of our study: astrocytes encode behaviourally relevant information over time and modulate future responses through molecular priming. In contrast to the faster dynamics of neuronal ensembles, astrocytic ensembles form via an intriguingly slow multiday trace, adding a novel dimension to the brain's adaptive coding mechanisms.

Comment #2-4: *Experiments of overexpression of *Adrb1* are important and reveal the role of astrocytes for fear generalization, however, to understand the role of this receptors in the particular ensembles, that is *Ast3* subpopulation (Fig 5d), instead of bulk overexpression in all cells a more precise approach is required to tackle only such subpopulation.*

This is a highly critical point raised by all the reviewers: directly testing the functional roles of Fos+ astrocytes induced by fear recall (FR-BAE) through population-specific manipulation. This aligns closely with our research priorities, and we have now included new data demonstrating that selective silencing of FR-BAE disrupts memory stabilization upon repeated recall, directly supporting the central claim of our study. To address this, we utilized a new, improved astrocyte silencing tool, i β ARK2—a modified version of the original i β ARK we reported previously (Nagai et al., *Neuron*, 2021¹). This improved construct includes a membrane-tethering Lck domain that enhances its efficacy by localizing i β ARK2 near plasma membrane GPCRs and associated G α q-GTP proteins, where it binds and sequesters these targets, effectively silencing astrocytic GPCR signalling. We characterized i β ARK2 alongside its mutated negative control, i β ARK2mut, and confirmed robust membrane-localized expression across astrocytic territories. Functional assays in acute amygdala slices showed that i β ARK2 significantly suppresses phenylephrine-evoked Gq-GPCR calcium responses with greater efficacy than the original i β ARK.

Fig. 6a-g

Using a Cre-dependent system in Fos-iCreERT2 mice, we expressed iβARK2 specifically in FR-BAE and found that this manipulation impairs memory stabilization, as evidenced by reduced freezing upon repeated recall compared to controls. Furthermore, levels of Igfbp2—a neuromodulatory factor secreted by astrocytes and induced after recall to support memory stabilization—were significantly reduced in iβARK2-expressing astrocytes. These findings indicate that iβARK2 impairs astrocyte-dependent memory support by disrupting downstream effector expression and establish a causal role for Fos+ astrocyte ensembles in maintaining fear memory through Igfbp2-mediated mechanisms. All new characterization and behavioural data have been incorporated into the revised manuscript (Fig. 6), and the newly developed plasmids and viruses have been deposited to Addgene (Supplementary Table 4).

Fig. 6h-k

Reviewer #3

(Remarks to the Author): The paper "The astrocytic engram is a multiday eligibility trace for memory stabilization" by Dewa et al. from the Nagai group attempts to recognize astrocytic engrams. They start by using the self-developed whole brain BAE (Behaviourally-relevant Astrocyte Ensembles) technique in TRAP mice, to tag cFos expressing astrocytes. They report an increase in the number of tagged astrocytes, especially in the amygdala, in the recall phase of the fear condition (FC), but not during acquisition. From here on they attempt to show the mechanisms involved: 1) they show that the astrocytic engram depends on local neuronal activity and on the secretion of NA from the LC. 2) They further show that amygdalar astrocytes express more NA receptors during recall, and the astrocytes with high cFos expression levels have high NA β 1Rs levels. 3) They show that overexpression of β 1Rs in amygdalar astrocytes leads to more cFos+ engram astrocytes.

The authors employ a variety of techniques (their very own BAE, other multiple virus injections, behaviour, photometry, cultures, FACS, single-cell RNA-seq, IHC), but they always use them close to question, without ever nailing the main point – that the cFos+ astrocytes are important to memory. All the beautiful experiments prove without doubt is that fear conditioning affect NA, and that NA affect astrocytes – both things are already known for a long time. They DO NOT show that NA affects the astrocyte engram, they don't do scRNA-seq of the astrocyte engram alone, nor do they show that the astrocytes of the engram are important to memory (or to anything).

*In the summery they say: "However, our findings suggest that the astrocytic engram may not serve as a direct memory trace encoding experience specific information like neurons because the augmentation of astrocytic engram did not alter initial fear recall. Instead, it likely functions as an eligibility trace for memory stabilization over days, creating a permissive and specialized microenvironment within circuits that supports the stabilization of memories related to repeated experiences". This is totally speculative. Nowhere in the paper do they show it directly. Nowhere do they manipulate the BAE astrocytes (it is doable! they can express *Adrb1* only in them, for example) and test the effect on behaviour. Sadly, I think that this work is not suitable for Nature.*

(underlining by us)

First of all, we sincerely thank the reviewer for their detailed and thoughtful review of our manuscript. We recognize that reviewing a study employing multi-faceted approaches requires significant time and effort, and we deeply appreciate the care taken in providing feedback.

We understand and acknowledge the reviewer's frustration, as well as their clear expectations regarding what should be addressed in the manuscript. We sincerely apologize for not fully meeting these expectations, particularly the "main" point, in our initial submission. Below, we aim to address the reviewer's comments directly, focusing on the central issue raised throughout the review—the functionality of Fos+ astrocytes in memory. As mentioned earlier, we are pleased to report that we

have directly tested the functional roles of Fos+ astrocytes induced by fear recall (FR-BAE) through population-specific manipulation. This aligns closely with our research priorities, and we have now included new data demonstrating that selective silencing of FR-BAE disrupts memory stabilization upon repeated recall, directly supporting the central claim of our study. We have been committed to ensuring that the revised manuscript meets these expectations and provides greater clarity on the novel aspects of our study.

To address the main point, we utilized a new, improved astrocyte silencing tool, iβARK2—a modified version of the original iβARK we reported previously (Nagai et al., *Neuron*, 2021¹). This updated construct includes a membrane-tethering Lck domain that enhances its efficacy by localizing iβARK2 near plasma membrane GPCRs and associated Gαq-GTP proteins, where it binds and sequesters these targets, effectively silencing astrocytic GPCR signalling. We characterized iβARK2 alongside its mutated negative control, iβARK2mut, and performed functional assays in acute amygdala slices that showed that iβARK2 significantly suppresses phenylephrine-evoked Gq-GPCR calcium responses with greater efficacy than the original iβARK. Please see the figure below and the relevant text in lines 442-454.

Fig. 6a-g

Using a Cre-dependent system in Fos-CreERT2 mice, we expressed $i\beta$ ARK2 specifically in FR-BAE and found that this manipulation impairs memory stabilization, as evidenced by reduced freezing upon repeated recall compared to controls. Furthermore, levels of *Igfbp2*—a neuromodulatory factor secreted by astrocytes and induced after recall to support memory stabilization—were significantly reduced in $i\beta$ ARK2-expressing astrocytes. These findings indicate that $i\beta$ ARK2 impairs astrocyte-dependent memory support by disrupting downstream effector expression and establish a causal role for Fos+ astrocyte ensembles in maintaining fear memory through *Igfbp2*-mediated mechanisms. All new characterization and behavioural data have been incorporated into the revised manuscript (Fig. 6), and the newly developed plasmids and viruses have been deposited to Addgene (Supplementary Table 4). The relevant texts are in lines 455-462.

Fig. 6h-k

While we agree with the reviewer on the main point regarding the functionality of Fos+ astrocytes, we respectfully disagree with the assertion that "All the beautiful experiments prove without doubt is that fear conditioning affects NA, and that NA affects astrocytes – both things are already known for a long time." We would appreciate clarification on which specific publications the reviewer is referring to with this comment.

First, we fully acknowledge the rich history of research on NA signalling in fear conditioning and memory reconsolidation, as highlighted in the outstanding papers we cited and discussed in the introduction and discussion sections. However, while the field is well-established, noradrenaline recording in the

amygdala using sophisticated GRAB sensors from behaving mice during fear conditioning, recall and in context B has never been reported. Our study presents this for the first time.

Second, while noradrenaline-induced astrocyte signalling has been extensively studied in many regions especially cortical areas, to our knowledge no prior research has specifically investigated this signalling in the amygdala in the context of memory.

Third, our findings go significantly beyond describing NA–astrocyte responses. Through molecular screening and *in vivo* manipulations, we show that astrocytic Fos induction requires convergence of noradrenergic signalling and engram neuron activity (Fig. 4 with new data; see response to Comment #3-2). We further demonstrate that fear learning enhances β 1- and α 1A-adrenoceptor expression in astrocytes (Fig. 5d-f, new data), enabling them to respond robustly to NA during memory recall. This learning-induced adaptation primes astrocytes to detect and respond to subsequent emotionally salient events—specifically those involving reactivated engram neurons—over days.

Fig. 5d–f (relevant texts in lines 386-393)

This integration results in the formation of astrocytic Fos ensembles during memory recall (Fig. 5g-h, l-p) that contribute to memory stabilization (Fig. 6).

In Fig. 5g–h, we show that astrocyte Fos inducibility upon FR peaks at 1 day post-FC, coinciding with maximal *Adra1a*–*Adrb1* co-expression, and declines at later time points as receptor levels fall. This temporal alignment suggests that astrocytic Fos induction during FR depends on prior adrenoceptor upregulation—a form of experience-dependent tuning rather than a mere reflection of stimulus intensity.

Fig. 5g–k (relevant texts in lines 393-397)

In Fig. 5l–p, we establish causal relationships between the adrenoceptors and Fos induction: conditional knockout of *Adra1a* or *Adrb1* in astrocytes abolishes Fos induction upon FR, confirming that both receptors are necessary.

Fig. 5 l–p (relevant texts in lines 400-406)

This mechanism represents a novel, multi-day form of astrocyte signal integration—distinct from previously described transient and global NA-triggered Ca^{2+} events—with circuit specificity and behavioral consequence.

To further illustrate the functional selectivity of astrocytic responses, we addressed a point that may have emerged from a recent paper by Deneen’s group (*Nature* 2025 ¹⁵) which has been published after your initial review. They reported astrocytic Fos induction in hippocampal CA1 following fear conditioning—seemingly contrasting our finding of minimal astrocytic tagging in that region. We hypothesized that this discrepancy was due to differences in behavioral protocol, particularly the presence or absence of contextual novelty. Unlike Deneen et al., our design incorporated three days of habituation to dissociate novelty-driven from memory-specific responses. To test this, we implemented a new “FC+Novelty” protocol without habituation. Using TRAP2-based astrocyte tagging and brain-wide imaging, we observed robust Fos+ astrocyte activation across multiple brain regions, including CA1—closely matching Deneen et al.’s findings. This result validates the sensitivity of our system and supports the interpretation that the reported astrocytic Fos activation can reflect novelty rather than memory *per se*.

Extended Data Fig. 6 (relevant texts in lines 224-235)

Crucially, our study goes beyond documenting activation: we dissect the upstream drivers and demonstrate that astrocytes integrate behavioral experience over days—via learning-induced receptor changes and convergent neural input—to support memory stabilization. This framing moves the field from “do astrocytes respond?” to “how do astrocytes compute meaningful behavioral signals?”—a mechanistic and conceptual advance.

We hope this mechanistic framework is now clearer. If the reviewer continues to question the novelty of our findings, we respectfully request references to prior studies that demonstrate:

(i) *in vivo* NA and astrocytic cAMP dynamics in the amygdala during fear learning and recall;

(ii) learning-induced multiday adrenoceptor upregulation in astrocytes that enables future memory-specific responses; or

(iii) causal manipulation of a convergent signalling pathway (NA + engram activity → astrocyte Fos → Igfbp2 output) that selectively affects memory stabilization, rather than acquisition or novelty responses.

To our knowledge, these findings have not been previously demonstrated. We apologize if aspects of our data presentation or discussion were unclear in the original submission, and we hope that the additional data and revised text now communicate our conclusions more clearly.

Major:

Comment #3-1: Fig. 1: the BAE numbers in the 4 conditions were compared to neuronal count in 3 conditions (there was no NoFR neurons) from Roy et al. (Nat. Commun, 2022). How? The number of engram astrocytes is an order of magnitude smaller than the neuronal one – what does it mean?

The first question is shared with Comment #1-2. We appreciate the opportunity to clarify this important point. We have made a separate Methods section “Correlational analysis of Fos+ cell counts” to provide a detailed explanation of this methodology.

Both our study and Roy et al. (Nat. Commun., 2022)² used TRAP2 (Fos-iCreERT2) mice combined with Cre-dependent fluorescent reporters (tdTomato in neurons by Roy et al., and mNeonGreen in astrocytes in our study) to perform Fos-tagging and single-cell resolution whole-brain imaging, followed by cell counting in anatomically registered brain regions using the Allen Brain Atlas. This enabled direct region-by-region comparisons across the two datasets. Roy et al. analyzed 247 brain regions and defined 117 “engram significant regions,” where Fos+ neuronal counts were significantly increased in both FC and FR conditions relative to non-foot-shocked controls. We extracted neuronal Fos+ fold changes upon FC or FR relative to non-shocked controls from a publicly available supplementary table in Roy et al., which provided region-level Fos+ neuron counts across all three groups. In our study, we initially quantified Fos-tagged cells across all 839 regions in the Allen Brain Atlas, and then selected putative 75 “engram significant regions” (64.1% of their total 117 engram significant regions) for correlation analysis. Selection criteria were as follows: we excluded regions (as we have described in the main text) containing Fos-tagged mNeonGreen-positive neuron-like cells, which were observed near the ventricles and midline (areas consistent with known GFAP-positive neural progenitor zones^{4,5}) to ensure accurate counting of bona fide astrocytes. We also excluded high-level hierarchical regions (e.g., the Midbrain, Thalamus) that had extremely high cell counts and could act as outliers in correlation analysis. This yielded 75 engram significant regions for correlation analysis in Fig. 1.

Importantly, we used only the NoFC group as the baseline reference for calculating fold changes in Fos+ cell counts for both neurons and astrocytes (e.g., FC–NoFC, FR–NoFC), consistent with Roy et al.². Although our study included an additional NoFR group (foot-shocked but not recalled), this group was used only for internal comparisons and not included in any fold-change analyses involving Roy et al.’s neuronal dataset.

Regarding the magnitude of fold changes, we agree that neuronal Fos+ counts showed relatively smaller fold increases than astrocytes. This likely reflects the higher baseline level of neuronal Fos+ cells in NoFC animals, which reduces the apparent dynamic range. While Fos+ neurons are more numerous overall, a single astrocyte is estimated to modulate tens to hundreds of thousands synapses¹⁶,

suggesting that even sparse astrocytic ensembles can exert broad influence over neuronal networks.

Comment #3-2: Fig 2, *in-vivo* (j-m): the authors write: "Thus, our data provide strong evidence that the convergence of local neural signals and long-range projecting NA-ergic modulatory signals is a cue for inducing astrocyte Fos in LA/B *in vivo*". Not at all! This is pure speculation. Your data shows in a non-behavioural study, that LC plus local neurons external stimulation affects the number of cFos astrocytes. Nothing less, nothing more.

We believe there may be a slight misunderstanding here, as we fully share the same interpretation as the reviewer. The data presented in Fig. 2 do not involve any behavioural data, which is why we subsequently conducted the *in vivo* experiments shown in Fig. 4 to investigate whether fear recall-induced astrocyte Fos expression depends on neural signals. Upon careful review of the sentence in question, we did not claim any behavioural relevance nor did we make any speculative assertions in relation to Fig. 2. In this regard, the reviewer and we are fully aligned—nothing more, nothing less.

To further test whether the convergence of NA and engram neurons occurs in the amygdala during fear recall, we performed additional anatomical analyses. Specifically, we employed dual-color Fos tagging in the amygdala of TRAP2 mice to label both FR engram neurons and FR-BAEs (Fig. 4f,g). We injected a neuron-specific Cre-dependent mCherry AAV (*hSyn1-DIO-mCherry*) and an astrocyte-specific Cre-dependent mNeonGreen AAV (*GfaABC1D-DIO-mNeonGreen*), and used immunostaining for NET to visualize NA axons. We then estimated the territory of each astrocyte using its marker (GFAP + S100 β) immunostaining and quantified the overlap between these territories and mCherry+ engram neuron somata and NET+ NA projections. Our analysis revealed that both engram neuron somata and NA axons were significantly enriched within the domains of mNeonGreen+ FR-BAEs compared to non-FR-BAEs (Fig. 4f,g), despite no difference in the astrocytic territory size. These data indicate that FR-BAEs are anatomically positioned to integrate inputs from both engram neurons and noradrenergic fibers, lending support to the idea that such convergence may act as a cue for astrocyte Fos induction during fear recall.

Fig. 4f,g (relevant texts in lines 339-352)

Comment #3-3: Behaviour, general: Very bold sayings are given in the paper based on behavioural studies, but it is hard to find two of them that are exactly the same:

We thank the reviewer for this important observation and apologize for any confusion caused by the earlier presentation of behavioral data. In response, we have newly analyzed an additional dataset with a consistent focus on memory destabilization and restabilization following reconsolidation. Specifically, we now directly compare 1st recall freezing with 2nd recall freezing across multiple experimental contexts, as shown in Fig. 6j, Fig. 7f,g, and Extended Data Fig. 11h. We hope this revised analysis more clearly illustrates our behavioral framework and strengthens the interpretation of memory-specific effects.

- Fig 4: Propranolol is given before and after recall, whereas CNO for both LC, local neurons and neuronal engram is given just before recall – why?

Thank you for highlighting this important point. The difference stems from the distinct pharmacokinetics of propranolol and CNO. In mice, propranolol has a half-life of ~1.5-2 hours¹⁷, while CNO effects reported to persist for several hours¹⁸. Given that the tagging time window for Fos+ astrocytes spans for several hours (Extended Data Fig. 3 and new data in Extended Data Fig. 5), we opted to administer propranolol before and after the tagging to ensure effective beta receptor blockade throughout this period. We have explicitly clarified this rationale in the revised Methods (lines 1440-1446).

- Fig 5 m-n: "We assessed freezing behaviour during subsequent FR sessions in CtxA the next day, with repeated 24-hour cycle infusions and FR sessions over five days (Fig.

5m)". What? Why? Which FR session are showing in 5n? For statistical analysis you used the Wilcoxon matched-pairs signed rank test – but the correct test should be a two-way ANOVA: treatment(IgG/Igfbp2) X time (before/after), and not each one by t-test - is it still significant?

We thank the reviewer for their careful examination of the details of our analysis design. In response to the suggestion, we conducted new analysis to examine the 5-day freezing data, and the results continue to show statistical significance (below; Extended Data Fig. 11h).

Extended Data Fig. 11h

- Fig 6e-h : FC with one shock. The only one in the paper, compared to three shocks. The significant finding to 1-shock was in 2 days FR, and the is not done at all for 3-shocks, but a completely different exp. Why?

Thank you for raising this important point. In retrospect, we initially conducted 3-shock experiments and examined freezing during the 1st recall, which already showed high freezing in context A (~>80%). The question we aimed to address was whether reinforcing Fos+ astrocyte ensemble signalling could overstabilize memory to further enhance freezing, thus needing to design an experiment with the parametric space to detect significant increases in freezing. However, after consulting with behavioural biologist colleagues, we anticipated that detecting even higher freezing during the 2nd recall might be challenging due to a potential ceiling effect. We then revisited past literature suggesting that overstabilization can lead to generalized fear memory (loss of specificity) as described in the main text, which inspired us to test this idea. The results were positive, showing enhanced freezing in context B (Fig. 7h). To circumvent the ceiling effect, we also performed weak fear conditioning with a single foot shock to test the initial hypothesis, and the data showed that overstabilization enhances freezing in context A (Fig. 7g).

As the reviewer correctly pointed out, this represents a slightly different question. We agree that including a recall test in context A under the 3-shock protocol would provide a more complete comparison. We have now conducted this experiment,

and the data show that both *Adrb1*-OE and control mice exhibited similarly high freezing levels upon recall one day after FC (Fig. 7f, *Adrb1*-OE: $89.6 \pm 4.3\%$; Control: $85.2 \pm 2.9\%$), with no significant difference between the groups. We interpret this result as a ceiling effect, which aligns with our rationale for using the single-shock paradigm to test the potential for enhanced freezing as shown in Fig. 7g. We are grateful to the reviewer for encouraging a more thorough and consistent comparison across paradigms. We hope these additional data clarify the logic of the single-shock experiments and strengthen the overall interpretation. The new results have been included in the revised manuscript (Fig. 7f).

Fig. 7e,f,g

Comment #3-4: Fig 6 a-c: this is the closest you came to manipulating the number of engram-astrocytes, yet you don't show the behavioural result – why??

We believe there may be a slight misunderstanding here. We present data on the number of engram astrocytes following *Adrb1* overexpression selectively in amygdalar astrocytes (Fig. 7a-c, previously Fig. 6a-c) and report the corresponding behavioural consequences in Fig. 7e-h (previously Fig. 6e-h). These results strongly support the idea that reinforcing the recruitment of astrocyte Fos ensembles and downstream signalling including *Igfbp2* (Fig. 7b, previously Figure 6d) in the amygdala leads to memory overstabilization.

Comment #3-5: Rows 101-103: "While astrocyte Fos induction has been demonstrated in response to various stimuli such as G protein-coupled receptor (GPCR) activation 40–43, behaviourally relevant stimuli 42,44 and astrocyte-neuron signaling 25,26,33 ...". This is the basis of cFos tagging in astrocytes. But it is misrepresenting some of these papers:

- "...G protein-coupled receptor (GPCR) activation 40–43" All these papers are using DREADDs, meaning they are NOT responding a natural stimulus, and indeed, in all of them beyond 90% of the astrocytes express cFos, clearly, abnormal.

- "...behaviourally relevant stimuli 42,44". #42 indeed show that startle response elevates cFos in astrocytes. In #44 I couldn't find any mention of c-fos in astrocytes at all. However, #43, reports no change in cFos following fear conditioning (Figure s4A), and is most relevant to the current study.

- "...astrocyte-neuron signaling 25,26,33". #25 (Sardar et al, Science, 2023) states: "We evaluated Fos expression, a marker of neuronal activity ... and we observed a significant 88.8% increase in neurons, but not in astrocytes", #26 shows RNA seq, #33 has RNAseq and MERFISH data.

So all together, there are only two papers that show protein levels cFos (which is relevant to BAE) following behaviour, one shows an increase after startle (#42), and one shows no response to FC (#43). It's ok to give opposite results. It's not ok to give results that seem unified, when in fact they're not.

Thank you for the careful assessment of our citations. First, as the reviewer correctly pointed out, references (original numbers #40–43) involve non-natural, strong GPCR stimulations using DREADDs, and we have removed these citations from the revised text. Second, we agree that citations #43 and #44 do not demonstrate Fos increases in astrocytes following fear conditioning. We sincerely apologize for this oversight. In the revised manuscript, we have replaced these citations with the recent study by Williamson et al. (*Nature* 2025; now reference #27), which reports increased c-Fos protein in hippocampal astrocytes after contextual fear conditioning with contextual novelty. Third, we now explicitly distinguish between studies showing Fos mRNA upregulation (via RNA-seq and MERFISH; ref. #34 in the revised text, originally ref. #26) and those showing c-Fos protein induction in astrocytes following behaviour (startle and fear conditioning, respectively; refs. #27 and #41 [previously #42] in the revised text). The revised sentence reads (lines 110-111):

"Previous studies have reported that behaviourally relevant stimuli increased astrocyte Fos mRNA⁽³⁴⁾ and c-Fos protein, the gene product of Fos,^(27,41) across selected brain regions. Nonetheless, astrocyte Fos probing has not been previously explored on a brain-wide scale."

We sincerely appreciate the reviewer's comments, which helped us improve the clarity and accuracy of our manuscript.

Minor:

Minor Comment #3-1: scRNA-seq: First, it's for all astrocytes, not just the BAE (why? They should be easy to sort). Second, all astrocytes were divided to 4 clusters, that are NOT different between groups (maybe the NoFC is different, but surely that's not what you're looking for). What am I missing? Yes, cFos and *Adrb1* are expressed in AST3, but in very low levels – so what?

This is an important point, and we are happy to clarify.

First, there seems to be a slight misunderstanding. Fos-TRAPed astrocytes express mNeonGreen, which takes approximately one week to become detectable. As the reviewer correctly pointed out, these mNeonGreen-positive cells can be sorted for single-cell transcriptomics. However, the resulting data represent the transcriptomic profiles of astrocytes that were Fos-positive one week earlier (remote time point). In contrast, the transcriptomic analysis from samples harvested 90 minutes after behavioural tasks captures the "ongoing" transient state changes at a more recent time point, which aligns with the purpose of our experiment. This provides insight into astrocytic responses immediately following behavioural tasks such as conditioning or recall. We of course do not claim that transcriptomics captures the entire landscape of molecular events underlying all astrocytic transcriptional responses after learning and memory recall. Instead, the primary goal of this experiment was to identify potential downstream molecules involved in the cellular outputs of FR-BAE. We successfully achieved this with *Igfbp2*, which was further validated in Extended Data Fig. 11.

Regarding the second point, this relates to the reviewer's concern about correlations between high levels of *Fos* and *Adrb1* in certain subclusters being insufficient to confirm the *Adrb1-Fos-Igfbp2* signalling pathway in the same astrocytes. This is exactly why we performed single-cell RNAscope analysis, which clearly demonstrates this pathway (Fig. 5i-k, Extended Data Fig. 11a-e), and we confirmed their functional roles in the context of memory stabilization (Extended Data Fig. 11f-j).

Minor Comment #3-2: *Fig 3c-e: the photometry is for all astrocytes, not for BAE. BAE is a small fraction of the astrocytes. Can they affect alone the astrocytic response?*

We appreciate the reviewer's thoughtful point. To fully address this, single-cell imaging in behaving mice would be required. This necessitates the installation of a two-photon microscopy system combined with fear conditioning. Beyond the significant financial and labor-intensive requirements of such an installation, there are also experimental challenges. Implementing fear conditioning in this setup would result in cued-fear conditioning rather than the contextual-fear conditioning used throughout the current manuscript, meaning the entire study would need to be repeated with a different paradigm. This would represent a fundamentally distinct project. While this is certainly a valuable avenue for future research, we believe such experiments fall beyond the scope of this manuscript. Considering the significant time, effort, and resources required, and given that we do not view them as essential for supporting the core conclusions of this study, we feel they are better suited for future investigations.

Minor Comment #3-3: *Row 275: " There were significant reductions in freezing when mice were treated with Prop before being placed in CtxA (Extended Data Fig. 4b'...)" it is Extended Data Fig. 5b'.*

The reviewer is correct. We have fixed this in the revised manuscript.

Minor Comment #3-4: Fig 6k – Why cFos is given as "intensity, a.u."? you can count the cells in the amygdala and get non arbitrary units. A bonus, you can compare it to previous studies.

Thank you for raising this important point. We recognize that the distinction between "intensity" and "positive/negative" thresholds is a broader issue in the field of single-cell expression analysis, particularly for genes that do not exhibit a clear binary pattern. In our analysis, c-Fos intensity (immunoreactivity from IHC) does not show a crystal clear binary pattern; while bi-peaks are present, some cells fall within an intermediate range, as shown by the smearing illustrated in the violin plot. To address this, we presented "intensity" to avoid potential bias introduced by arbitrary thresholding for positive/negative classification.

With this in mind, we also analyzed the data using a % positive cells approach, as suggested by the reviewer. Defining c-Fos positivity as intensity exceeding 15-fold over the background, we observed ~12% c-Fos+ cells in Fos-tagged neurons (neuronal engrams labeled during fear recall) that re-expressed c-Fos within the amygdala in the control group in context B (Fig. 7k). This percentage aligns well with previous reports (e.g. ¹⁹, Figure 2L). Using this threshold, we also observed a significant increase in c-Fos+ neuronal engrams with reinforcing astrocyte Fos ensemble recruitment and signalling via *Adrb1*-OE in the amygdala. Thank you for highlighting this opportunity to strengthen our analysis.

Fig. 7k

1. Nagai, J. *et al. Neuron* **109**, 2256-2274.e9 (2021).
2. Roy, D. S. *et al. Nat. Commun.* **13**, 1799 (2022).
3. Nagai, J. *et al. Neuron* **109**, 576–596 (2021).
4. Yao, Z. *et al. Nature* **624**, 317–332 (2023).
5. Furube, E. *et al. Sci. Rep.* **10**, 2826 (2020).
6. Williamson, M. R. *et al. Nature* (2024) doi:10.1038/s41586-024-08170-w.
7. DeNardo, L. A. *et al. Nat. Neurosci.* **22**, 460–469 (2019).
8. Zaki, Y. *et al. Nature* (2024) doi:10.1038/s41586-024-08168-4.
9. Bae, S. E., Holmes, N. M. & Westbrook, R. F. *Learn. Mem. Cold Spring Harb. N* **22**, 519–525 (2015).
10. Wong, F. S., Westbrook, R. F. & Holmes, N. M. *eLife* **8**, e47085 (2019).
11. Rudy, J. W., Barrientos, R. M. & O'Reilly, R. C. *Behav. Neurosci.* **116**, 530–538 (2002).
12. Sardar, D. *et al. Int. J. Mol. Sci.* **22**, 3975 (2021).
13. Sun, W. *et al. Science* **339**, 197–200 (2013).

14. Josselyn, S. A. & Tonegawa, S. *Science* **367**, eaaw4325 (2020).
15. Williamson, M. R. *et al. Nature* **637**, 478–486 (2025).
16. Chai, H. *et al. Neuron* **95**, 531-549.e9 (2017).
17. Levy, A., Ngai, S. H., Finck, A. D., Kawashima, K. & Spector, S. *Eur. J. Pharmacol.* **40**, 93–100 (1976).
18. Roth, B. L. *Neuron* **89**, 683–694 (2016).
19. Lesuis, S. L. *et al. Cell* S0092-8674(24)01216–9 (2024) doi:10.1016/j.cell.2024.10.034.